# TRACERx analysis identifies a role for *FAT1* in regulating chromosomal instability and whole-genome doubling via Hippo signalling

Chromosomal instability (CIN) is common in solid tumours and fuels evolutionary adaptation and poor prognosis by increasing intratumour heterogeneity. Systematic characterization of driver events in the TRACERx non-small-cell lung cancer (NSCLC) cohort identified that genetic alterations in six genes, including *FAT1*, result in homologous recombination (HR) repair deficiencies and CIN. Using orthogonal genetic and experimental approaches, we demonstrate that *FAT1* alterations are positively selected before genome doubling and associated with HR deficiency. *FAT1* ablation causes persistent replication stress, an elevated mitotic failure rate, nuclear deformation and elevated structural CIN, including chromosome translocations and radial chromosomes. *FAT1* loss contributes to whole-genome doubling (a form of numerical CIN) through the dysregulation of YAP1. Co-depletion of *YAP1* partially rescues numerical CIN caused by *FAT1* loss but does not relieve HR deficiencies, nor structural CIN. Importantly, overexpression of constitutively active YAP1$^{5SA}$ is sufficient to induce numerical CIN. Taken together, we show that *FAT1* loss in NSCLC attenuates HR and exacerbates CIN through two distinct downstream mechanisms, leading to increased tumour heterogeneity.

Chromosomal instability (CIN) is pervasive during cancer evolution, particularly in non-small-cell lung cancer (NSCLC), where it is associated with poor recurrence-free survival[1–4]. CIN results in loss of heterozygosity (LOH) events, the burden of which correlates with the frequency of whole-genome doubling (WGD) events in solid tumours[5,6]. WGD not only mitigates the effect of LOH resulting from CIN, but also fosters ongoing CIN[7–10]. By duplicating the complete set of chromosomes, WGD is a key event during cancer evolution and correlates with poor prognosis and targeted therapy resistance[7,11–13]. Despite the importance of CIN and WGD in accelerating cancer evolution by promoting intratumour heterogeneity[7], genetic events responsible for the initiation and maintenance of CIN and WGD in NSCLC have not been systematically investigated.

Tracking Cancer Evolution Through Therapy/Rx (TRACERx) is a longitudinal cancer study that utilizes multiregional sampling and whole-exome sequencing (WES)[3] to identify and time genetic alterations and their relationship with WGD and CIN[3]. Here, we identify mutations in cancer driver genes that co-occur with WGD and CIN and further characterize their respective involvement in the DNA damage response (DDR) and CIN. We demonstrate that genetic perturbation of six TRACERx driver genes identified in our screen (particularly *FAT1*) causes deficiencies in homologous recombination (HR) repair and elevated CIN. Using NSCLC cell line models, we further characterize the molecular mechanisms by which *FAT1* alterations drive WGD and CIN—known mediators of drug resistance[12,14–16]. These findings highlight the importance of *FAT1* gene alterations in lung cancer evolution.

## Results

### Identification of DDR and CIN drivers in lung TRACERx

To identify alterations that correlate with WGD and CIN in NSCLC, we analysed multiregional WES data from tumours obtained from the first 100 patients within the lung TRACERx study[3]. Here, we identified

✉e-mail: wei-ting.lu@crick.ac.uk; jb@cancer.dk; n.kanu@ucl.ac.uk; charles.swanton@crick.ac.uk

795 driver events in 91 genes, excluded known oncogenes that could not be modelled appropriately by genetic depletion approaches and focused on 37 tumour suppressor gene mutations that co-occurred with WGD (Fig. 1a). Pathway analysis revealed cellular processes related to genome maintenance, such as the DDR, transcription and chromatin remodelling (Supplementary Fig. 1)[17–19]. To assess how these genes contribute to genome integrity, we performed a multi-parametric RNA interference (RNAi) screen using high-content imaging in four different lung cancer cell lines harbouring mutations in *KRAS*, *EGFR* and *TP53* to reflect the mutational landscape of the TRACERx cohort (Fig. 1b (top) and Extended Data Fig. 1a). DNA double-strand breaks (DSBs) induced by ionizing radiation, as well as replication stress induced by hydroxyurea, were chosen to model genotoxic stress. In parallel, the impact on chromosome loss was investigated utilizing a human artificial chromosome (HAC) reporter system[20] (Fig. 1b (bottom) and Extended Data Fig. 1b). These combined approaches enabled the identification of six tumour suppressor driver genes—namely *BAP1*, *CREBBP*, *FAT1*, *NCOA6*, *RAD21* and *UBR5*—as regulators of the DDR and maintenance of chromosomal stability in NSCLC (Fig. 1c). In addition, we also confirmed previously reported roles of *WRN*, *FANCM*, *DICER1*, *SMARCA4/BRG1*, *ARID1B*, *ARID2*, *KDM5C* and *ATRX* in the DDR[21–26] (Extended Data Fig. 1a).

The effect of these genes on the DDR was initially validated using two orthogonal approaches; the DR-GFP HR reporter assay[27] and a site-specific DSB-generating endonuclease system (DIvA)[28]. Loss of *BAP1*, *FAT1*, *NCOA6*, *RAD21* or *UBR5*, but not *CREBBP*, was associated with HR repair deficiency and impaired single-stranded DNA (ssDNA) resection—a step required for accurate HR repair (Fig. 1d,e and Supplementary Fig. 2). Next, we assessed early HR repair signalling 1 h after 6 Gy ionizing radiation in both A549 and H1944 cells. A marked decrease in the formation of RAD51 ionizing radiation-induced foci (IRIFs) was observed following depletion of *BAP1*, *FAT1*, *NCOA6*, *RAD21* or *UBR5* (Fig. 1f, Extended Data Fig. 2a,b and Supplementary Figs. 3 and 4). These alterations in HR efficiency could not be fully explained by cell cycle changes (Extended Data Fig. 2c), which dictate the selection of activating HR or non-homologous end-joining (NHEJ) repair, suggesting that these genes may be involved in HR directly. *FAT1* was prioritized as a top candidate of clinical relevance, as inactivating mutations in *FAT1* were highly recurrent (~10%) in the TRACERx 421 cohort (comprising 421 patients and 1,644 tumour regions), with a notable proportion of *FAT1* mutations occurring early before WGD (Fig. 1g and Supplementary Fig. 5a–c). Notably, ~20% of the lung TRACERx cohort were found to harbour other inactivating mutations that impair HR efficiency (Fig. 1g and Extended Data Fig. 2d).

### FAT1 alterations are positively selected in lung cancer

To quantify whether mutations in *FAT1* were under positive selection, we measured the enrichment of *FAT1* mutations before and after WGD using the ratio of the observed number of non-synonymous substitutions per non-synonymous site to the number of synonymous substitutions per synonymous site (dN/dS)[29]. Estimates of dN/dS above or below 1 suggest positive or negative selection, respectively, whereas estimates overlapping 1 imply that there is no evidence of selection. *FAT1* mutations, which occurred more frequently in lung squamous cell carcinoma (LUSC) compared with lung adenocarcinoma (LUAD), were under greater positive selection before WGD occurrence in LUSC (Fig. 2a and Extended Data Fig. 3a). In the TRACERx 421 cohort, an enrichment of copy number deletion events was identified around the *FAT1* genomic locus *4q35.2* only in patients with clonal WGD, indicating positive selection of *4q35.2* loss (Fig. 2b and Extended Data Fig. 3b–d). In LUSC, *FAT1* promoter hypermethylation events reducing FAT1 expression levels and *FAT1* copy number loss events were observed in the same tumours (Extended Data Fig. 3e–g). Furthermore, we detected a significant occurrence of mirrored subclonal allelic imbalance (MSAI) at the *4q35.2* locus, suggesting parallel evolution (Fig. 2c). Among genes encoded at the *4q35.2* locus, *FAT1* has the lowest Genome Aggregation Database

loss-of-function score in germline samples, implying that *FAT1* loss is the least tolerated event within *4q35.2* (Fig. 2d). These results highlight the importance of *FAT1* alterations.

### FAT1 ablation reduces HR efficiency

Considering the frequency of *FAT1* alterations in NSCLC (Figs. 1g and 2, Extended Data Fig. 3 and Supplementary Fig. 5) and its potential role in genome maintenance (Fig. 1d–f), we further elucidated at which stage *FAT1* acts in the DSB repair pathway by systematically investigating which mediators were affected by *FAT1* depletion 1 h post-ionizing radiation. *FAT1* knockdown did not impact IRIFs of early DSB repair mediators, including phosphorylated ATM, γH2A.X and 53BP1 oligomerization, which are associated with NHEJ[22] (Fig. 3a and Supplementary Fig. 6a,b). However, IRIF formation of CtBP interacting protein (CtIP), which is responsible for initiating ssDNA resection[22], was significantly impaired by *FAT1* knockdown (Fig. 3a and Supplementary Fig. 6c). *FAT1* depletion also reduced IRIF formation of the key HR mediator breast cancer type 1 susceptibility protein (BRCA1) in G2/M cells, using centromere protein F-positive staining as a marker (Fig. 3a and Supplementary Fig. 6d). CRISPR knockout of *FAT1* in H1944 or A549 cells impaired RAD51 IRIF formation, but not γH2A.X (Extended Data Fig. 4a,b and Supplementary Fig. 6e). A time-course post-ionizing radiation demonstrated that *FAT1* depletion was associated with persistent DNA damage, as manifested by the increased frequency of 53BP1 nuclear bodies (Extended Data Fig. 4c).

Although the size of full-length FAT1 (4,588 amino acids) limits its ectopic expression, the reduction in RAD51 foci formation could be partially rescued by overexpression of the FAT1 carboxy (C)-terminal intracellular domain[30] (HA–FAT1[ICD]; amino acids 4,202–4,588), which exhibited nuclear localization, suggesting that nuclear FAT1 may promote efficient HR (Fig. 3b,c and Extended Data Fig. 4d,e). *FAT1*-knockout A549 cells were more sensitive to genotoxic stress induced by poly(ADP-ribose) polymerase inhibitors, cisplatin and hydroxyurea (Extended Data Fig. 5a,b). By analysing both the TRACERx and The Cancer Genome Atlas (TCGA) LUAD datasets, we observed a correlation between *FAT1* loss and HR deficiency (HRD)-related genomic signatures, including elevated telomeric allelic imbalance (TAI)[31], large-scale transitions (LST)[32] and loss of heterozygosity (LOH)[33] (Fig. 3d–f). Using established reporters for distal and alternative end-joining activities, respectively[34], we confirmed that transient *FAT1* siRNA depletion reduced the alternative end-joining efficiency without significantly reducing distal end joining (Fig. 3g,h). Utilizing WGS data from Genomics England, no significant difference was observed in ID6 and SBS3 mutational profiles (Extended Data Fig. 5c,d and Supplementary Fig. 7a,b), both of which are mutation signatures associated with NHEJ activity[35].

### FAT1 ablation leads to structural and numerical CIN

Indeed, *FAT1* depletion resulted in an increased fork collapse rate and HAC loss (Fig. 4a and Extended Data Fig. 1b). Our analysis of the TCGA and Genomics England datasets revealed that *FAT1* loss correlated with an increased weighted genome instability index score and total mutational burden, indicating elevated structural and numerical CIN (Fig. 4b,c). To validate this observation, we used U2OS and type 2 pneumocyte (T2P) cells to investigate the formation of micronuclei and 53BP1 nuclear bodies in G1 daughter cells, both established markers of unresolved replication stress and HRD[36–39]. *FAT1* ablation at baseline and under replication stress induced by either low-dose aphidicolin or a short pulse of hydroxyurea exacerbated the formation of cyclin A-negative (G1-specific) 53BP1 nuclear bodies and micronuclei (Fig. 4d–h and Extended Data Fig. 6a). Notably, acentric micronucleus formation was significantly elevated following *FAT1* depletion in A549 and U2OS cells following replication stress (Fig. 4f,g). Under these conditions, *FAT1* ablation also resulted in an increased mitotic error rate at baseline and under replication stress, which

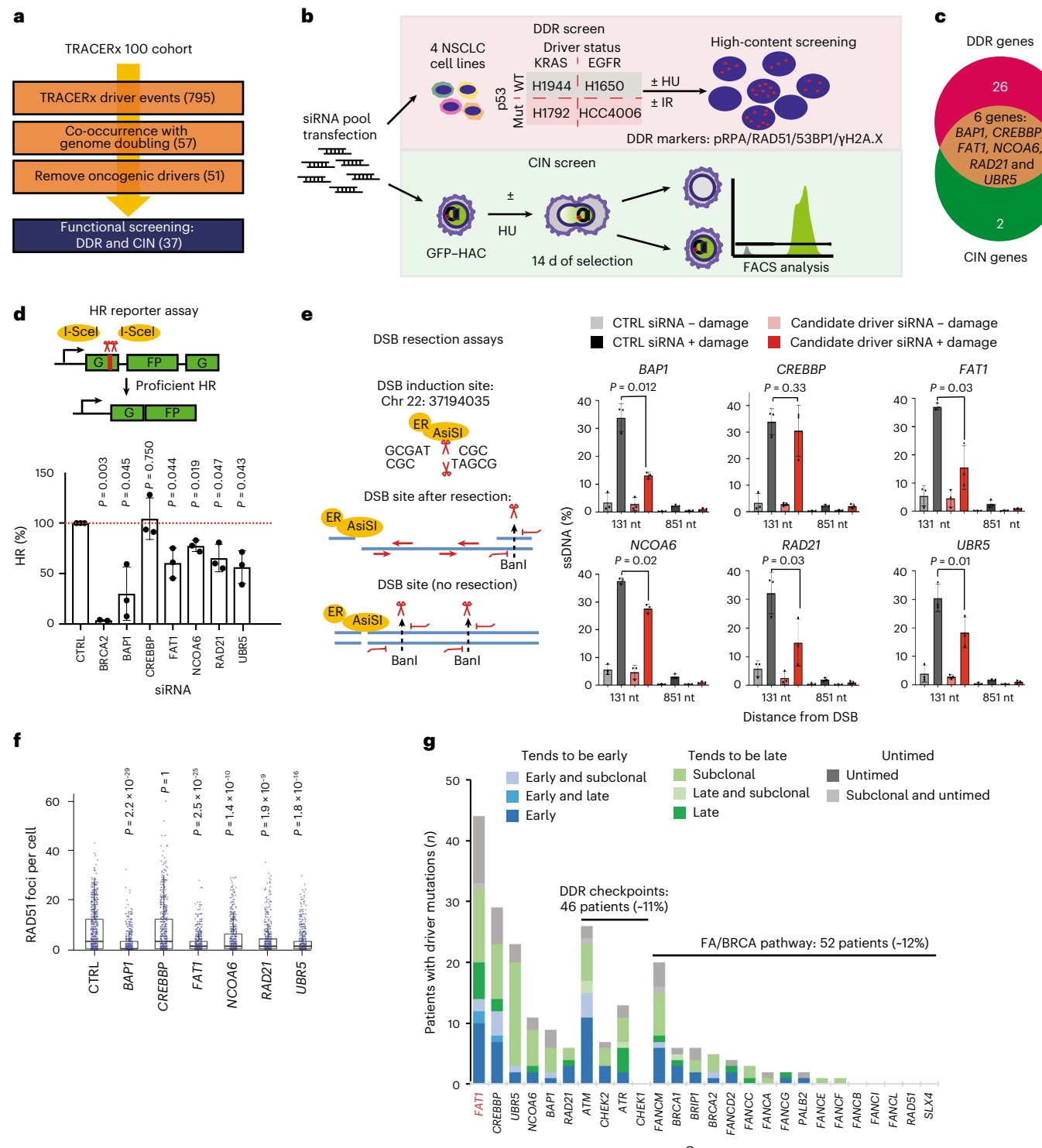

**Fig. 1 | DDR and CIN loss-of-function screen of genome doubling-associated drivers from the TRACERx 100 cohort. a,** Flow chart depicting candidate gene selection for the DDR and CIN screens. **b,** Schematic of the design of the DDR and CIN screens. **c,** Venn diagram showing the six driver genes contributing to DDR and CIN. **d,** Validation of the six candidate genes by DR-GFP homologous recombination reporter assay; BRCA2 serves as a positive control. HR efficiencies are normalized to those of control samples. Statistical significance was determined by two-sided, one-sample *t*-test. The data represent means ± s.d. (*n* = 3 biological replicates, except for BRCA2, for which *n* = 2). **e,** Validation of the six candidate genes by DIvA U2OS-AsiSI site-directed resection assay. Statistical significance was determined by two-sided paired *t*-test. The data represent means ± s.d. (*n* = 3 biological replicates). **f,** Box plots quantifying

RAD51 foci formation in A549 cells following depletion of the six candidate genes, following 6 Gy ionizing irradiation and 1 h of recovery. The box edges represent interquartile ranges, the horizontal lines represent median values and the ranges of the whiskers denote 1.5× the interquartile range (*n* = 3 biological replicates; >150 cells quantified per biological replicate). Statistical significance was determined by Kruskal–Wallis test with Dunn's multiple comparison test. **g,** Driver mutation distribution and mutational timing of the six candidate genes in the TRACERx 421 cohort. *ATM*, *CHEK2*, *ATR*, *CHEK1* and members of the Fanconi anaemia (FA)/BRCA pathway are included for comparison. *FAT1* is highlighted in red. CTRL, control; EGFR, epidermal growth factor receptor; FACS, fluorescence-activated cell sorting; HU, hydroxyurea; IR, ionizing radiation; mut, mutant; nt, nucleotides; WT, wild type.

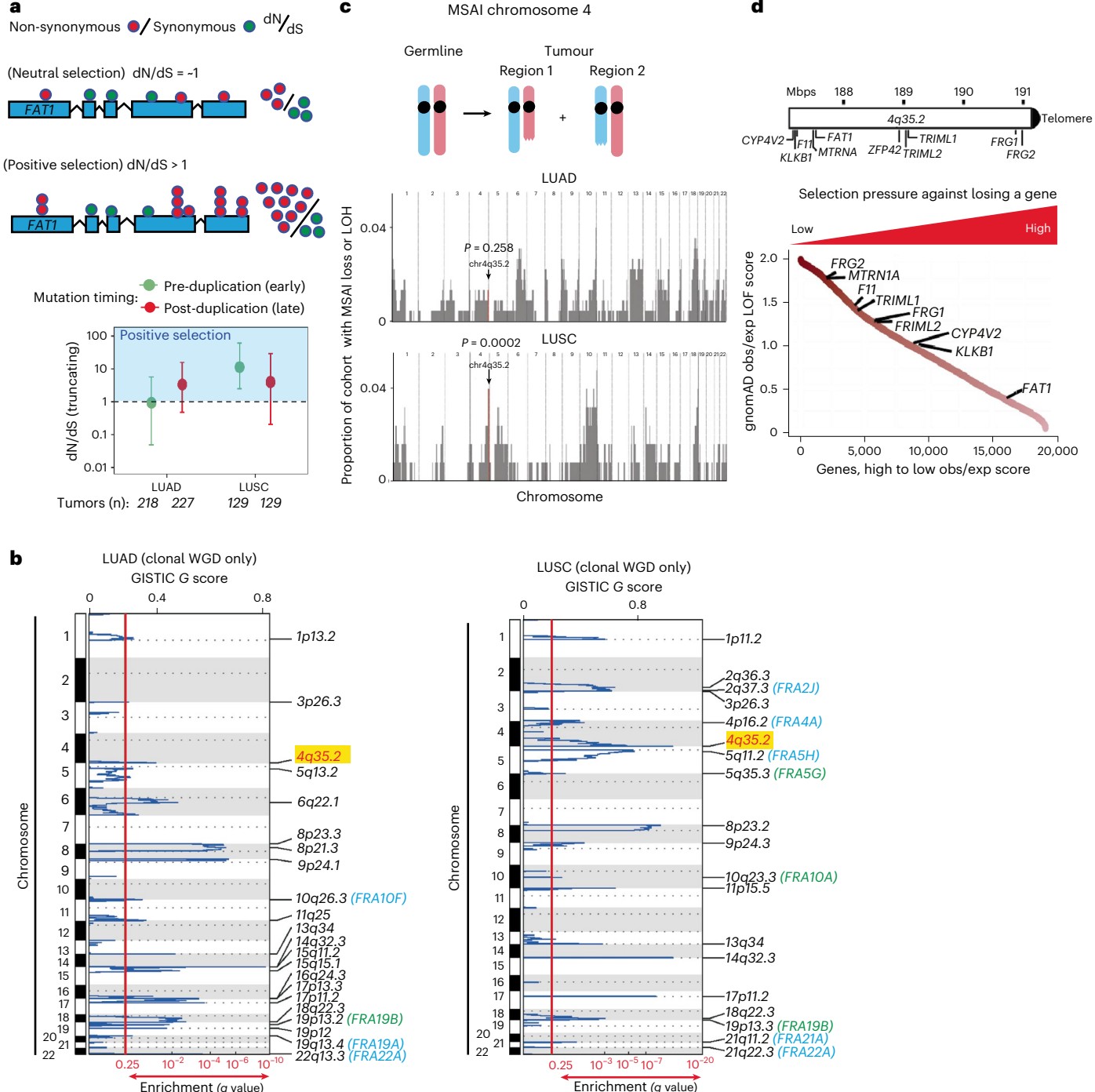

**Fig. 2 | *FAT1* loss of function is enriched in the TRACERx 421 cohort and leads to an elevated mitotic error rate and WGD. a**, Top, schematic of dN/dS ratio analysis. Bottom, results of dN/dS ratio analysis in the TRACERx 421 cohort, demonstrating that *FAT1* truncation mutations are selected early in LUSC tumour evolution. The data points represent estimated dN/dS ratios and the error bars represent 95% confidence intervals calculated using the genesetdnds function from the R package dNdScv. The TRACERx 421 cohort comprised 233 males and 188 females (421 patients total), corresponding to a 55:45 male:female ratio. 93% of the cohort was from a White ethnic background and the mean age of the patients was 69 years, ranging between 34 and 92 years. Written informed consent was obtained. None of the patients was compensated for their involvement in the study. **b**, Genomic identification of significant targets In cancer (GISTIC) analysis of LUAD (141 patients) and LUSC tumours (80 patients) in TRACERx with clonal WGD only, demonstrating that SCNA loss

at the *FAT1* genomic locus (*4q35.2*; red text and highlighted) is positively selected in tumours with clonal WGD only. SCNA loci overlapping with common or rare chromosome fragile sites[72,73] are annotated (in blue for common fragile sites and in green for rare fragile sites). **c**, MSAI analysis illustrating that the genomic region of chromosome 4 that harbours the *FAT1* gene (arrows) is frequently lost in LUSC. Statistical significance was determined by Fisher's exact test. In the schematic at the top, paternal and maternal chromosomes are indicated in blue and red, respectively. **d**, Top, schematic illustrating the location of the *FAT1* gene on chromosome 4, together with other *4q35.2* genes within the frequently lost *4q35.2* genomic region. Bottom, selection pressures against losing genes. The data are from the Genome Aggregation Database (gnomAD) and demonstrate high selective pressure against deletion of the *FAT1* genomic locus within *4q35.2*. exp, expected; LOF, loss of function; obs, observed.

manifested as an increased formation rate of chromatin bridges and lagging chromosomes (Fig. 5a and Extended Data Fig. 6b). Utilizing the DIvA site-directed DSB system[28,40], we further demonstrated that *FAT1* depletion resulted in increased illegitimate translocation of two DSBs induced on chromosome 17 (Fig. 5b). Concurrently, higher rates of structural chromosomal aberrations, including radial chromosomes and chromatid gaps, were observed upon *FAT1* loss (Fig. 5c and Extended Data Fig. 6c). *FAT1* silencing also reduced mitotic fidelity, as evidenced by deviations in the modal chromosome number (Fig. 5d–f and Extended Data Fig. 6d).

Since *FAT1* mutations are both common in lung cancer and evolutionarily selected before WGD (Fig. 2a–c), we experimentally validated the mitotic defect associated with *FAT1* loss using the near-triploid (3N) LUAD cell line PC9 (which harbours an in-frame deletion at exon 19 of the epidermal growth factor receptor-encoding gene) and its isogenic WGD hexaploid (6N) clone[12]. Transient FAT1 depletion in PC9 cells significantly elevated the rate of stalled replication forks and mitotic errors, further confirming the involvement of *FAT1* in genome maintenance (Extended Data Fig. 6e–g). Despite resulting in an elevated replication fork collapse rate, reduced formation of interphase and mitotic Fanconi anaemia complementation group D2 (FancD2) foci was also observed in *FAT1*-depleted WGD PC9 cells, suggesting a failure to recover from replication stress, leading to structural CIN (Extended Data Fig. 6e,g–j).

## FAT1 depletion leads to mitotic errors and WGD

WGD events were highly prevalent in the lung TRACERx 421 cohort (84% of LUSC and 77% of LUAD) and *FAT1* driver mutations were selected before WGD in LUSC tumours (Fig. 2a). However, *FAT1* mutations and WGD did not significantly co-occur in the TRACERx 421 cohort (Fisher's exact test; $P = 0.179$) (Extended Data Fig. 7a). This was probably due to the presence of other HR-related gene alterations (~20% of patients; Fig. 1g and Extended Data Fig. 2d), which also can contribute to WGD. To investigate whether *FAT1* alterations drive WGD, we used the PC9 lung cancer model to quantify the proportion of actively replicating cells with >6N genome content (the basal ploidy of PC9 is 3N). Increased 5-ethynyl-2′-deoxyuridine (EdU) incorporation rates (Fig. 6a,b), as well as a significant increase in loading of the replicative helicase MCM7 beyond 6N, were observed in *FAT1*-knockout PC9 cells (Extended Data Fig. 7b), both indicating a second replication event post-6N and suggesting that *FAT1* loss is associated with WGD. These observations were independent of p53 mutational status, as similar results were obtained in *TP53* wild-type, near diploid, untransformed retinal pigment epithelial-1 (RPE-1) cells immortalized with the human telomerase reverse transcriptase subunit (hTERT) (Extended Data Fig. 7c).

To identify the cause of WGD following *FAT1* depletion, we monitored cells using live-cell microscopy, tracking at single-cell resolution. Reported causes of WGD include cytokinesis defects, endoreplication, mitotic bypass and cyclin B1 dysregulation during G2 (refs. 41–44). No change in cyclin B1 level was observed upon *FAT1* knockdown in G2, ruling out cyclin B1 dysregulation as the cause of WGD in the absence of *FAT1* (Extended Data Fig. 7d,e). DNA synthesis, measured by EdU incorporation, was also unaltered in control versus *FAT1*-knockout WGD cells transiently blocked in mitosis using nocodazole (Extended Data Fig. 7f), thereby ruling out a role for *FAT1* loss in driving WGD through endoreplication in a manner similar to that of cyclin E amplification reported recently[44]. To investigate mitotic bypass, we used the hTERT RPE-1 cell line expressing both H2B-mTurquoise and fluorescent, ubiquitination-based cell cycle indicator (FUCCI) (hereafter FUCCI–RPE-1 cells) for live-cell imaging (Supplementary Video 1). *FAT1* depletion, irrespective of the induction of aphidicolin-induced replication stress, did not cause a significant increase in the rate of mitotic bypass (Fig. 6c and Supplementary Fig. 8). In contrast, *FAT1*-depleted cells demonstrated an elevated rate of cytokinesis failure, suggesting defects in this final step of cell division as the cause of the WGD associated with *FAT1* deficiency (Fig. 6d, Extended Data Fig. 7g and Supplementary Video 2). *FAT1* depletion was also associated with an increased rate of nuclear shape abnormalities in daughter cells after normal mitoses (Fig. 6e and Supplementary Video 3). Similarly, after *FAT1* depletion, increases in multinucleation and nuclear morphology alterations were observed in fixed U2OS and RPE-1 cells (Extended Data Fig. 7h,i) and WGD PC9 cells, respectively, after replication stress exposure (Extended Data Fig. 8a).

To delineate whether structural CIN precedes nuclear shape abnormalities, we performed live-cell spinning-disk confocal microscopy on FUCCI–RPE-1 cells in G2, allowing us to determine the timing and outcome of the pending mitosis and the fate of respective daughter cells at high resolution with low phototoxicity. We observed that *FAT1* depletion not only increases chromatin bridge formation rates (Extended Data Fig. 8b, left) but reduces the maintenance of normal nuclear morphology following mitotic chromosomal bridge formation (Extended Data Fig. 8b (right) and Supplementary Videos 4 and 5). Next, we investigated the long-term outcome of daughter cells with deformed nuclear morphology, by observing the heritability and recurrence of nuclear deformities and mitotic errors in respective daughter cells following mitosis (Extended Data Fig. 8c). In addition, the number of normal second mitoses was significantly reduced following an initial nuclear deformation event (Extended Data Fig. 8c). Furthermore, *FAT1*-depleted cells had an accelerated rate of metaphase entry (Extended Data Fig. 8d), suggestive of a potential role for *FAT1* in

**Fig. 3 | FAT1 loss attenuates HR repair. a**, Box plots demonstrating the impact of *FAT1* siRNA knockdown on early DNA damage signalling and 53BP1 binding in A549 cells. The boxes represent interquartile ranges, the black and red bars represent median and mean values, respectively, and the ranges of the whiskers denote 1.5× the interquartile range. Statistical significance was determined by two-sided Wilcoxon rank-sum test ($n = 3$ biological replicates). The total numbers of cells quantified per condition were as follows: $n \geq 370$ (pATM), $n \geq 438$ (γH2A.X), $n \geq 448$ (53PB1), $n \geq 560$ (CtIP) and $n \geq 218$ (BRCA1). **b**, Schematic of the FAT1 functional domains. The full-length FAT1 protein is 4,588 amino acids. **c**, RAD51 IRIF formation following 6 Gy ionizing radiation and 1 h of recovery in *FAT1* CRISPR knockout (sgFAT1) versus control A549 cells with overexpression of HA–FAT1[ICD] versus pcDNA3.1. The boxes represent interquartile ranges, the black and red bars represent median and mean values, respectively, and the ranges of the whiskers denote 1.5× the interquartile range. Statistical significance was determined by two-sided Kruskal–Wallis test followed by Dunn's test with Bonferroni correction ($n = 3$ biological replicates). **d–f**, Top, cartoons depicting examples of HRD-related large-scale transition (LST; **d**), telomeric allelic imbalance (TAI; **e**) and LOH (**f**). Bottom left, Permutation analysis showing a correlation between *FAT1* CNA and HRD-related genomic signatures based on TCGA LUAD data. Red lines indicates 90 and 95% confidence intervals, blue line indicates observed correlation value. Bottom right, *FAT1* driver mutation scores for these respective genetic alterations, based on TRACERx LUAD data. For the TRACERx LUAD data, tumour numbers were as follows: $n = 212$ (WT) and $n = 17$ (mut). In the box and whisker plots, the boxes represent interquartile ranges, the lines represent median values and the ranges of the whiskers denote 1.5× the interquartile range. Statistical significance was determined by two-sided mixed-effects linear model with purity as a fixed covariate and tumour ID as a random variable. **g**, Top, cartoon showing the design of the EJ5–GFP distal end-joining reporter integrated in U2OS cells. Bottom, *53BP1* siRNA knockdown, but not *FAT1* knockdown, affects the distal end-joining rate. The data represent means ± s.d. Statistical significance was determined by two-sided repeated measures one-way analysis of variance (ANOVA) with Holm–Šidák correction ($n = 5$ biological repeats). **h**, Top, cartoon showing the design of the EJ2–GFP alternative end-joining reporter integrated into U2OS cells. Bottom, *FAT1* siRNA knockdown significantly reduces the alternative end-joining efficiency. The data represent means ± s.d. Statistical significance was determined by two-sided paired *t*-test ($n = 4$ biological repeats). EGF, epidermal growth factor-like domain; LAMG, laminin G-like domain; NLS, nuclear localization signal.

regulating mitotic timing. Taken together, these results demonstrate that *FAT1* loss contributes to both an increased rate of structural CIN and elevated mitotic defects in daughter cells following CIN, contributing to WGD.

Since elevated CIN is known to synergize with WGD to generate increased intratumour heterogeneity[3,45] and escape from targeted therapeutic pressure[12,14,46,47], we also investigated whether *FAT1* loss might enable WGD PC9 cells to accelerate cancer evolution and bypass targeted therapy treatment. PC9 cells were treated with a sensitizing

concentration of the epidermal growth factor receptor inhibitor osimertinib (>90% inhibitory concentration) for 5 weeks (Extended Data Fig. 8e,f). An increase in clonal survival and clonal derivation rate was observed for *FAT1*-knockout cells over 3 months (Extended Data Fig. 8g,h). Osimertinib-resistant *FAT1*-knockout clones exhibited elevated ploidies compared with control clones (Extended Data Fig. 8i). Taken together, these results suggest that *FAT1* loss might fuel WGD and elevated CIN, which in turn exacerbates cancer evolution and targeted therapy resistance, as previously demonstrated[12,14,46,47].

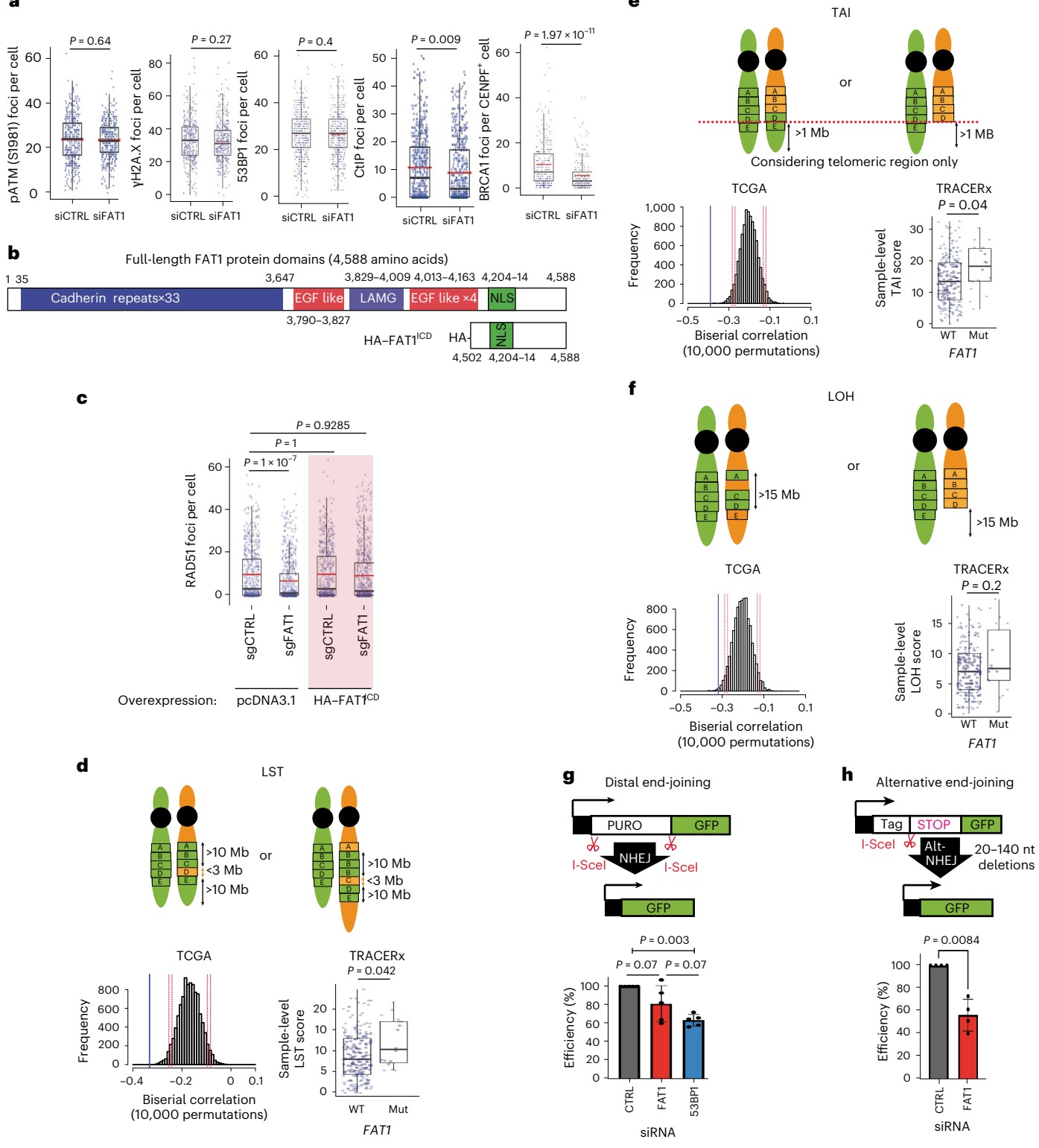

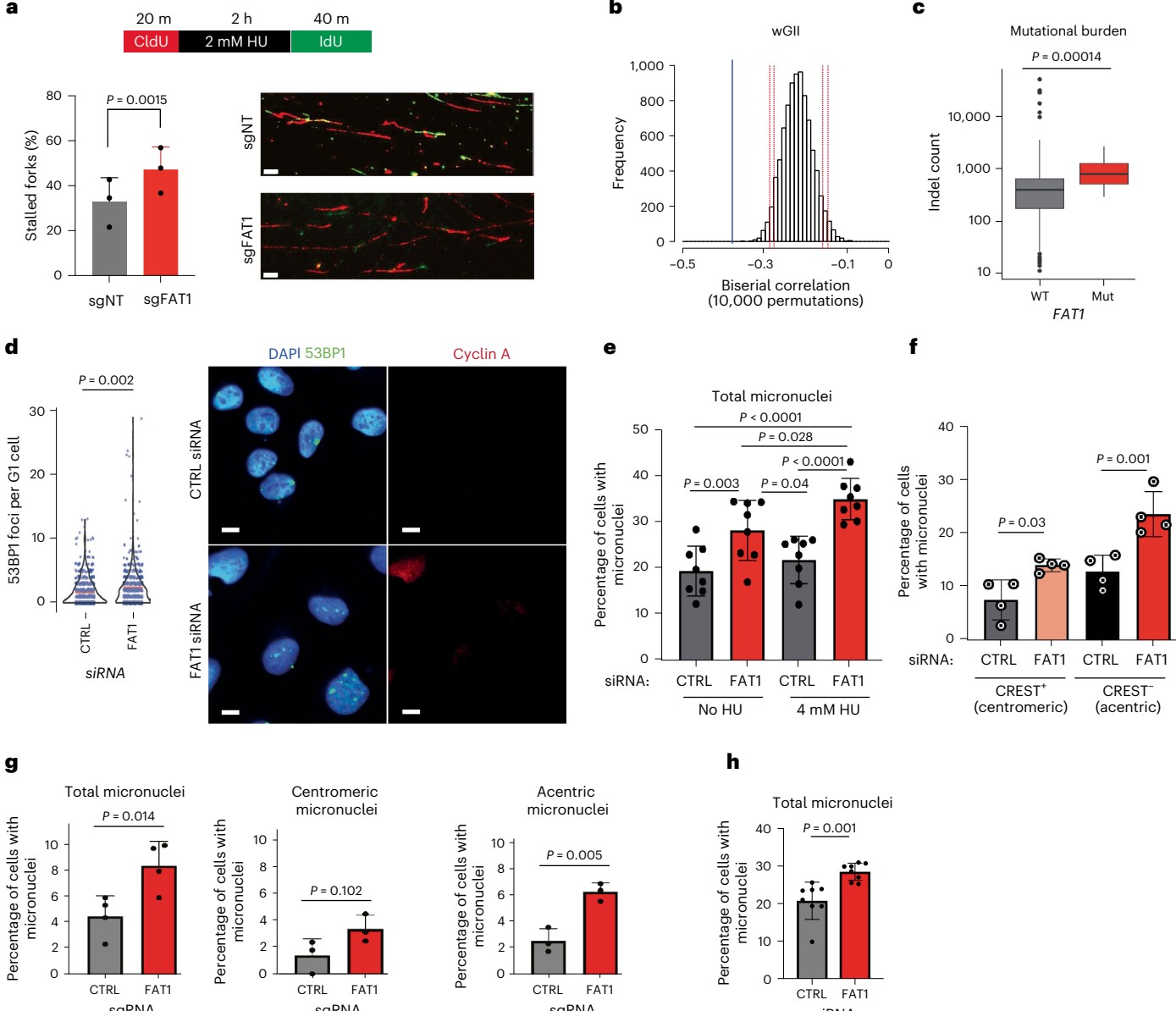

**Fig. 4 | FAT1 loss elevates replication stress and micronuclei. a**, *FAT1* knockout exacerbates replication fork stalling in A549 cells. Top, scheme of the nucleotide labeling used to measure replication fork stalling. Bottom (left) quantification; (right), representative image for the DNA fibre experiments. The data represent means ± s.d. Statistical significance was determined by two-sided paired *t*-test (*n* = 3 biological replicates; >600 forks counted in total). Scale bars, 20 μm. **b**, TCGA LUAD analysis showing that *FAT1* copy number loss is significantly correlated with weighted genome instability index measurements. The blue lines indicate *FAT1* loss and the red dotted lines indicate the 90 and 95% confidence intervals. Confidence intervals were generated using computational permutation analyses. **c**, Box plot comparing the numbers of indels in *FAT1* WT versus mutated tumours in the Genomics England LUAD and LUSC cohorts. The boxes represent interquartile ranges, the lines represent median values and the ranges of the whiskers denote 1.5× the interquartile range. Statistical significance was determined by two-sided Wilcoxon rank-sum test. *n* = 818 (WT) and *n* = 16 (mut). **d**, Transient *FAT1* siRNA knockdown induces the formation of

53BP1 bodies in cyclin A-negative U2OS cells following 4 mM hydroxyurea for 5 h and recovery for 24 h. Statistical significance was determined by two-sided Wilcoxon rank-sum test. Scale bars, 10 μm. The red bars in the graph to the left represent mean values (*n* = 5 biological replicates). **e,f**, Transient *FAT1* siRNA knockdown in U2OS cells induces the formation of total micronuclei with or without replication stress induced by 5 h of 4 mM hydroxyurea followed by 24 h recovery (**e**), as well as the formation of both acentric and centromeric micronuclei following the hydroxyurea treatment (**f**). The data represent means ± s.d. Statistical significance was determined by one-way ANOVA with Bonferroni correction. Biological repeats: *n* = 8 (**e**) and *n* = 4 (**f**). **g,h**, *FAT1* loss elevates the rate of micronuclei formation in response to replication stress induced by 0.2 μM aphidicolin treatment (24 h) following *FAT1* CRISPR knockout in A549 cells (**g**) or transient siRNA knockdown in T2P cells (**h**). The data represent means ± s.d. Statistical significance was determined by two-sided Student's *t*-test. Biological repeats: *n* = 4 (total micronuclei in **g**), *n* = 3 (centromeric and acentric micronuclei in **g**) and *n* = 8 (**h**).

## FAT1 depletion leads to dysregulated Yes-associated protein 1 signalling

To determine the molecular mechanism by which *FAT1* alterations lead to CIN, we investigated components of potential signalling pathways in which FAT1 has been implicated. *FAT1* loss has been connected to

dysregulation of the Hippo pathway, RAS–RAF–MEK–MAPK signalling and regulation of the level of Yes tyrosine kinase, which mediates hybrid–epithelial–mesenchymal transition[48,49]. Under our experimental conditions, no consistent *FAT1* siRNA-mediated alterations of MEK–ERK phosphorylation or Yes tyrosine kinase levels (Fig. 7a)

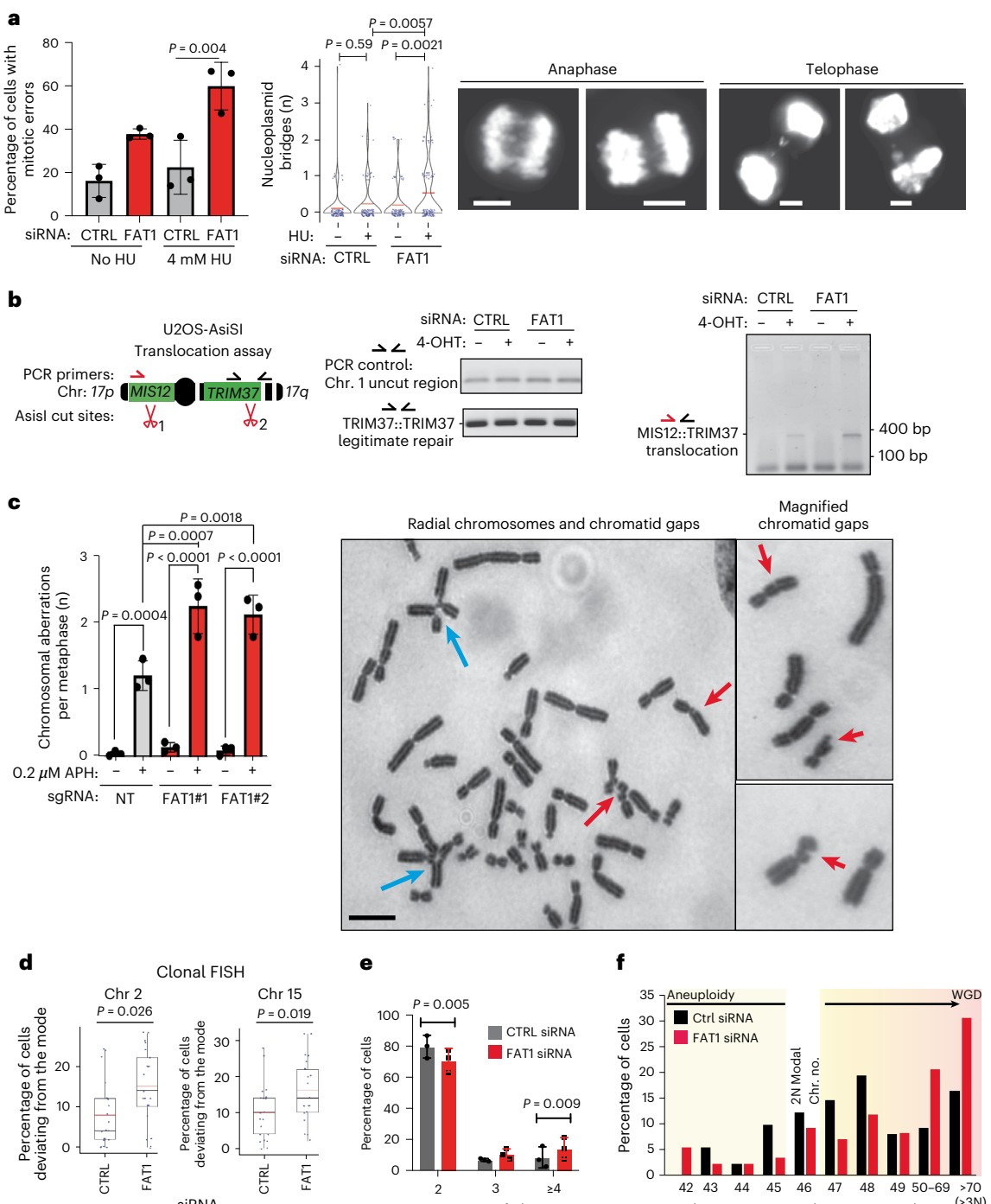

**Fig. 5 | *FAT1* loss increases structural CIN and chromosome numbers.**
**a**, Transient *FAT1* knockdown significantly increases the mitotic error rate (lagging chromosomes plus DAPI bridges; left; data represent means ± s.d.) and the occurrence of nucleoplasmid bridges (middle; red bars represent mean values) in U2OS cells after 5 h treatment with 4 mM hydroxyurea and 24 h recovery. Statistical significance was determined by one-way ANOVA with Bonferroni correction (left) or Dunn's test (middle). Right, selected maximum projection images following *FAT1* knockdown, showing DAPI-stained mitotic U2OS cells following treatment with 4 mM hydroxyurea and 24 h recovery. Scale bars, 5 μm. Over 100 mitotic cells were scored across three biological replicates. **b**, Representative PCR-based semi-quantitative DIvA U2OS-AsiSI translocation assay. Transient *FAT1* siRNA knockdown increases illegitimate repair products. PCR products generated from the uncut region and the legitimate repair product were used as the loading control. *n* = 3 biological replicates. **c**, Histogram (left) and representative images (right) showing that A549 cells with *FAT1* loss exhibit

a significantly increased number of chromosomal aberrations upon challenge with replication stress induced by 0.2 μM aphidicolin (APH) treatment. Scale bar, 5 μm. The data represent means ± s.d. Statistical significance was determined by one-way ANOVA with Holm–Šidák correction. A total of 60 metaphases were scored across three biological replicates per condition. Blue and red arrows indicate radial chromosomes and chromatid gaps, respectively. **d**–**f**, Transient *FAT1* siRNA knockdown causes a significant numerical deviation in chromosome number in H1944 cells, as determined by multiple methodologies, including clonal fluorescence in situ hybridization (**d**), ImageStream high-throughput flow cytometry (**e**) and metaphase spreads (**f**). The histogram data represent means ± s.d. For the box plots, the boxes represent interquartile ranges, the black and red lines represent median and mean values, respectively, and the ranges of the whiskers denote 1.5× the interquartile range. Statistical significance was determined by two-sided Wilcoxon rank-sum test (**d**) or two-sided paired *t*-test (**e**). *n* = 3 biological replicates for all cases. bp, base pairs. NT, non-targeting.

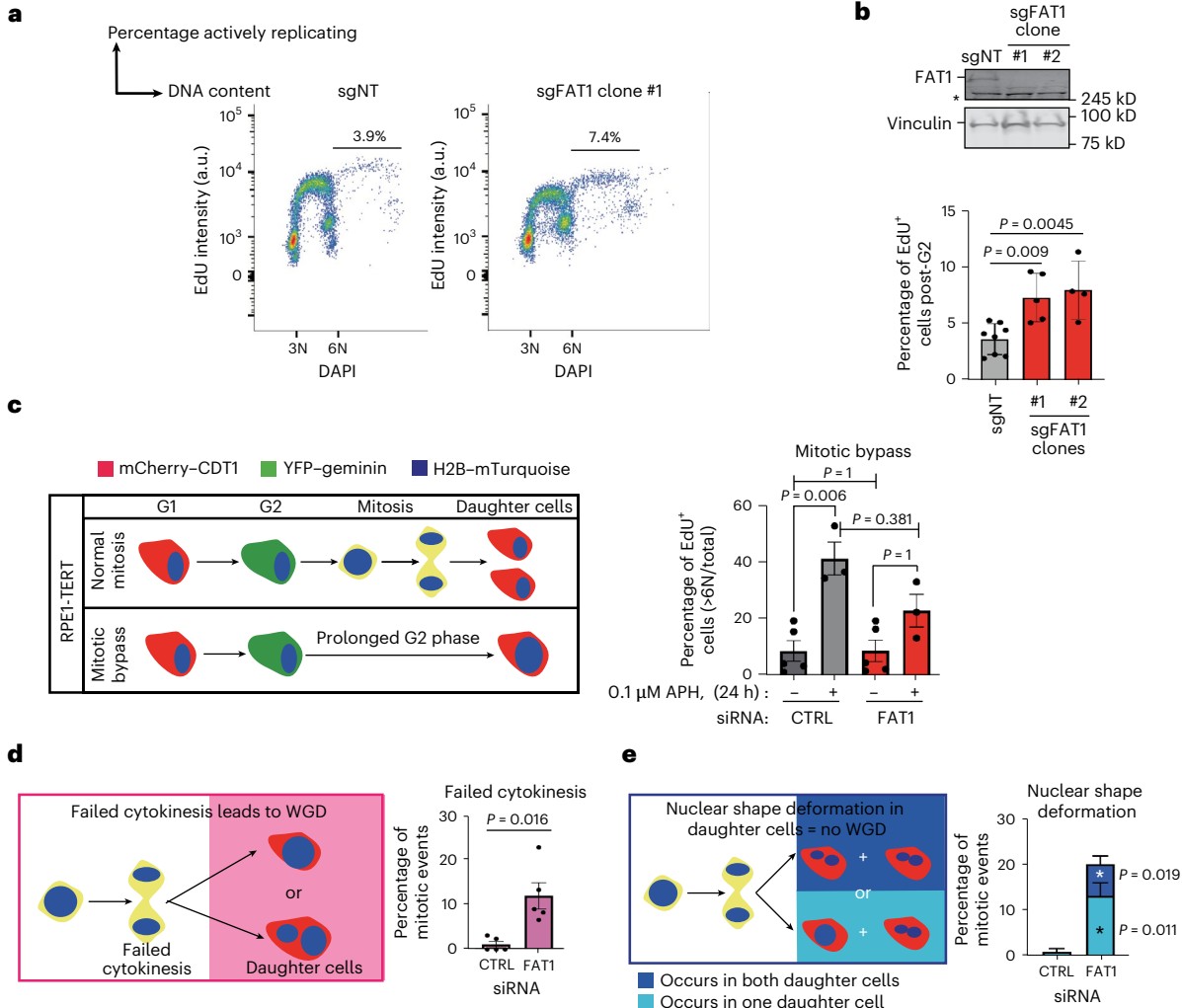

**Fig. 6 | *FAT1* loss leads to an elevated mitotic error rate and results in WGD.**
**a**, Representative dot plots demonstrating *FAT1* ablation in PC9 cells and assessment of EdU incorporation beyond the normal G2 phase, to visualize WGD. **b**, Top, representative western blot validating *FAT1* knockout. Bottom, quantification of EdU incorporation beyond the normal G2 population showing that *FAT1* knockout significantly increases the WGD population in PC9 cells. The data represent means ± s.d. Statistical significance was determined by one-way ANOVA with Bonferroni correction. Biological repeats: *n* = 7 (sgNT), *n* = 5 (sgFAT1 clone 1) and *n* = 4 (sgFAT1 clone 2). **c**, Schematic (left) and histogram (right) showing the impact of transient *FAT1* knockdown in TERT RPE-1 cells on the promotion of WGD through mitotic bypass, as determined by live-cell

imaging. The data represent means ± s.e.m. Statistical significance was determined by one-way ANOVA with Bonferroni correction. Biological repeats: *n* = 3 (with aphidicolin treatment) and *n* = 6 (without aphidicolin treatment). **d**,**e**, Schematics (left) and histograms (right) showing that transient *FAT1* siRNA knockdown in TERT RPE-1 cells increases the rates of cytokinesis failure (**d**) and nuclear shape deformation (**e**), as determined by 30× magnification live-cell microscopy imaging at 20 min intervals. The data represent means ± s.e.m. Statistical significance was determined by two-sided paired *t*-test, At least 200 mitotic events were tracked per condition over five biological replicates. YFP, yellow fluorescent protein.

were observed. However, *FAT1* depletion resulted in elevated nuclear localization of the key Hippo pathway activator YAP/TAZ (Fig. 7b and Extended Data Fig. 9a,b). Despite increased Yes-associated protein 1 (YAP1) nuclear localization, no significant alteration in YAP1 phosphorylation could be consistently observed following LATS1 or FAT1 depletion (Extended Data Fig. 9c) in T2P or FUCCI–RPE-1 cells.

Orthogonally, we designed a neonGreen reporter with 14xTEAD binding sites to elucidate how FAT1 regulates the ultimate output of the Hippo pathway, TEAD transcriptional activity. This is particularly important since *YAP1* and *TAZ/WWTR1* are highly analogous, functionally overlapping[50] and both amplified in lung cancer[3,51] (Extended Data Fig. 9d). By scoring HA-tagged construct and neonGreen reporter double-positive cells, we confirmed that *FAT1* loss elevated TEAD transcriptional activity, despite the presence of multiple copies of *YAP/TAZ* in hexaploid WGD PC9 cells (Fig. 7c, condition 1 versus condition 2). Notably, overexpression of the constitutively active YAP1 mutant

construct[52] further stimulated TEAD activity in *FAT1*-knockout cells (HA-YAP1^SSA; Fig. 7c, condition 3).

Next, we investigated which FAT1 domains are required to re-suppress TEAD transcriptional activity. In addition to the C-terminal HA–FAT1^ICD fragment capable of rescuing RAD51 IRIF formation (Fig. 3b,c), a FAT1 construct encompassing the extracellular and transmembrane domain was generated (mScarlet–HA–FAT1^WT in Fig. 7c), consisting of 3,735–4,588 amino acids of FAT1. Since FAT1 is known to cooperate with the E3 ligase MIB2 to regulate YAP/TAZ signalling[53], we also generated a FAT1 mutant lacking the MIB2 interaction domain (mScarlet–HA–FAT1^MIBΔ in Fig. 7c). Indeed, the mScarlet–HA–FAT1^WT fragment successfully re-repressed TEAD transcription, whereas the FAT1 construct lacking the MIB2 interaction domain failed to do so (Fig. 7c (conditions 4 and 5) and Extended Data Fig. 9e), highlighting the role of FAT1–MIB2 interaction in the modulation of TEAD transcription. Notably, the HA–FAT1^ICD construct, which lacks a transmembrane

domain anchor compared with HA–FAT1[WT] (Supplementary Fig. 9), failed to re-repress TEAD transcription (Fig. 7c, condition 6), despite successfully rescuing RAD51 IRIF formation in A549 cells (Fig. 3b,c).

Next, we investigated whether depletion of *LATS1/2*–crucial modulators of YAP1 nuclear localization[54]–might lead to phenotypes resembling those of DDR and CIN observed following *FAT1* depletion. Indeed, a marked reduction of the DNA end-resection rate, 53BP1 nuclear body formation and an increase in micronuclei formation were all observed following depletion of either LATS1 or LATS2 (Fig. 7d–f and Extended Data Fig. 10a). Using the site-directed endonuclease DIvA system, depletion of LATS2 caused illegitimate DSB repair, culminating in a translocation (Extended Data Fig. 10b). Notably, somatic copy number alteration (SCNA) loss of *LATS2* genomic loci at *13q12.11* was positively selected for in LUSC (Extended Data Fig. 10c). Despite causing HRD, neither *LATS1* nor *LATS2* depletion disrupted the activation of early DDR signalling, such as phosphorylation of KAP1/TRIM28 or γH2A.X phosphorylation (Extended Data Fig. 10a). Similar to *FAT1* depletion, *LATS1* knockdown led to an elevated mitotic error rate (Fig. 7g).

### FAT1 and YAP1 co-depletion reverses WGD but not HR defects

Given the involvement of *FAT1* loss in YAP/TAZ activity (Fig. 7b,c), we systematically investigated whether WGD and numerical and structural CIN associated with *FAT1* loss could be reversed by co-depleting *YAP1*. Despite *FAT1/YAP1* co-depletion reversing the cell cycle arrest associated with *YAP1* single knockdown (Extended Data Fig. 10d,e), co-depletion of *FAT1* and *YAP1* did not rescue HR activation defects after DSB formation (Fig. 8a–c). This result was orthogonally validated using the DR-GFP reporter assay, ssDNA formation after endonuclease-induced DSBs and RAD51 foci formation after ionizing irradiation (Fig. 8a–c). Next, since unresolved recombination intermediates due to HR repair deficiency can cause mitotic errors[55], we quantified the rate of mitotic errors. *YAP1/FAT1* co-depletion failed to rescue the elevated mitotic error and chromosomal bridges observed in *FAT1*-knockout A549 cells (Fig. 8d,e and Extended Data Fig. 10f).

Next, we investigated whether *FAT1/YAP1* co-depletion might rescue the CIN phenotypes associated with *FAT1* loss. Live-cell imaging experiments demonstrated that co-depletion of *FAT1* and *YAP1* reversed the cytokinesis failure phenotypes (Fig. 8f (left) and Extended Data Fig. 10g). However, the nuclear shape deformation observed following *FAT1* depletion was only partially rescued (Fig. 8f, right).

Taken together, our observations suggest that *FAT1* possesses dual roles. One outcome of *FAT1* loss is HRD leading to unresolved recombination intermediates, replication stress, structural CIN and nuclear deformation (Figs. 3–5 and 8a–d); these events appear to be YAP1 independent. In contrast, hyperactive YAP1 leads to increased cytokinesis failure and WGD (Fig. 8f). To visualize the impact of constitutively active mScarlet–YAP1[5SA] overexpression on WGD, we performed an EdU incorporation assay. Complementary to the literature[56], mScarlet–YAP1[5SA] overexpression promoted WGD in RPE-1 cells and this was independent of *TP53* status (Fig. 8g and Supplementary Fig. 10). Concurrently, we observed an emerging WGD population in *FAT1* wild-type and *FAT1*-knockout PC9 cells (Fig. 8h). These results suggest that constitutively active YAP1 is sufficient to promote WGD. We postulate that *FAT1* loss might drive genome instability through two different routes–one through WGD, dependent on YAP1, and the second through HRD, driving structural CIN–in a TEAD/YAP1-independent manner.

## Discussion

We systematically analysed 37 NSCLC drivers previously reported in the lung TRACERx study[3] that are associated with CIN and WGD. Particularly, we show that depletion of either *BAP1*, *FAT1*, *NCOA6*, *RAD21* or *UBR5* is associated with downregulation of multiple key steps required for HR repair and elevated loss of a HAC, thereby confirming previous reports of the roles of *BAP1*, *RAD21* and *UBR5* in genome maintenance[57–59]. In addition, our DDR screen confirms previous reports characterizing the role of *WRN*, *FANCM*, *DICER1*, *SMARCA4/BRG1*, *ARID1B*, *ARID2*, *KDM5C* and *ATRX* in the DDR[21–26]. Despite *FAT1* being frequently altered in normal somatic skin cells[60] and in multiple cancers[61–63], as well as being associated with NSCLC mortality[2], the mechanistic links between *FAT1* alterations, DDR and both numerical and structural CIN have remained elusive.

Through our comprehensive analyses, we have demonstrated that *FAT1* ablation impacts HR efficiency, but not NHEJ efficiency, nor the early signalling events of the DDR. *FAT1* loss results in hallmarks of HRD, such as increased sensitivity to replication stress inducers, elevated chromosome translocation, elevated fork collapse rates and increased HRD-predictive genetic scars in NSCLC. Our data suggest that *FAT1* loss leads to end-resection deficiency after the chromatin remodelling step to initiate HR. The inability to accurately resolve replication stress or clastogenic breaks may lead to chromosome-level alterations[36], especially structural CIN.

*FAT1* alterations have been detected in normal somatic tissue[60] and we hypothesize that *FAT1* inactivation occurs early in tumorigenesis. Using the multiregional sequencing of the TRACERx study to time the occurrence of *FAT1* alterations, we observed evidence of positive selection for tumour subclones with *FAT1* alterations before WGD. Indeed,

**Fig. 7 | *FAT1* loss leads to dysregulation of the Hippo pathway. a**, Western blot demonstrating the impact of *FAT1* knockdown on the Src–Mek–Erk signalling axis in hTERT RPE-1 and T2P cells. The results are representative of three repeats. **b**, Scatter plot showing the impact of transient siRNA depletion of *FAT1* on nuclear YAP1 localization using the stringent PTEMF fixation buffer (Methods) in TERT RPE-1 cells. The red bars represent median values. Statistical significance was determined by two-sided Kruskal–Wallis test followed by Dunn's multiple comparisons test. Over 170 cells were scored per condition over three biological replicates. **c**, Top, schematic illustrating the predicted domains of the FAT1 protein and respective regions cloned into a pCMV expression plasmid with an HA epitope tag. Bottom, TEAD activity in *FAT1* knockout PC9 hexaploid WGD cells. The normalized TEAD activity was measured as an enrichment of the neonGreen signal over the untransfected background signal in each experiment. The HA signal was used to identify successful *FAT1* rescue construct co-transfection at the single-cell level. TEAD activity was elevated in *FAT1*-knockout cells but could be further increased by overexpressing the constitutively active HA–YAP1[5SA] mutant. Overexpression of the *FAT1* wild-type construct repressed TEAD activity. However, overexpression of *FAT1* mutants devoid of the MIB2 binding region (mScarlet–HA–FAT1[MIBΔ]) and HA–FAT1[ICD] did not repress TEAD activity. The edges of the histograms represent mean values. Statistical significance was determined by two-sided Kruskal–Wallis test followed by Dunn's multiple comparisons test. More than 95 cells were scored over three biological replicates. **d**, DSB resection assay, showing that transient knockdown of *LATS1* and *LATS2*–both negative regulators of YAP1–represses ssDNA formation at DSB break sites (chr22:37194035, ssDNA measured 131 nucleotides from the DSB) in U2OS-AsiSI cells. The data represent means ± s.d. (*n* = 4 biological replicates). Statistical significance was determined by two-sided paired *t*-test. **e**, Both *LATS1* and *LATS2* siRNA knockdown in U2OS cells elevate rates of 53BP1 nuclear body formation when challenged with replication stress (5 h of 4 mM hydroxyurea followed by 24 h recovery). The boxes represent interquartile ranges, the black and red lines represent median and mean values, respectively, and the ranges of the whiskers denote 1.5× the interquartile range. Statistical significance was determined by two-sided Wilcoxon rank-sum test. More than 340 cells were scored over three biological replicates. **f**, Both *LATS1* and *LATS2* siRNA knockdown in U2OS cells induce centromeric and acentric micronuclei formation following challenge with replication stress (5 h of 4 mM hydroxyurea followed by 24 h recovery), suggestive of a mitotic segregation deficiency. Statistical significance was determined by repeated measures one-way ANOVA. The data represent means ± s.d. (*n* = 3 biological replicates). **g**, Transient siRNA knockdown of *LATS1* in U2OS cells elevates the mitotic error rate. The data represent means ± s.d. Statistical significance was determined by repeated measures one-way ANOVA (*n* = 3 biological replicates). MIB2, MindBomb2-interacting domain; mS, mScarlet; TM, transmembrane domain.

*FAT1* depletion attenuates HR and causes unresolved replication stress, both of which contribute to mitotic errors and micronucleus formation[39]. Using live-cell microscopy, we demonstrated that *FAT1* depletion not only elevates structural CIN, such as mitotic bridges, but also increases subsequent nuclear shape deformation.

*FAT1* has been proposed to be tumour suppressive and involved in various signalling pathways, including the CaMK2, WNT and Hippo signalling pathways[48–50,62]. Components of the Hippo pathway have been reported to impact various aspects of genome mainte- nance, such as mitotic control, stabilization of replication forks, and

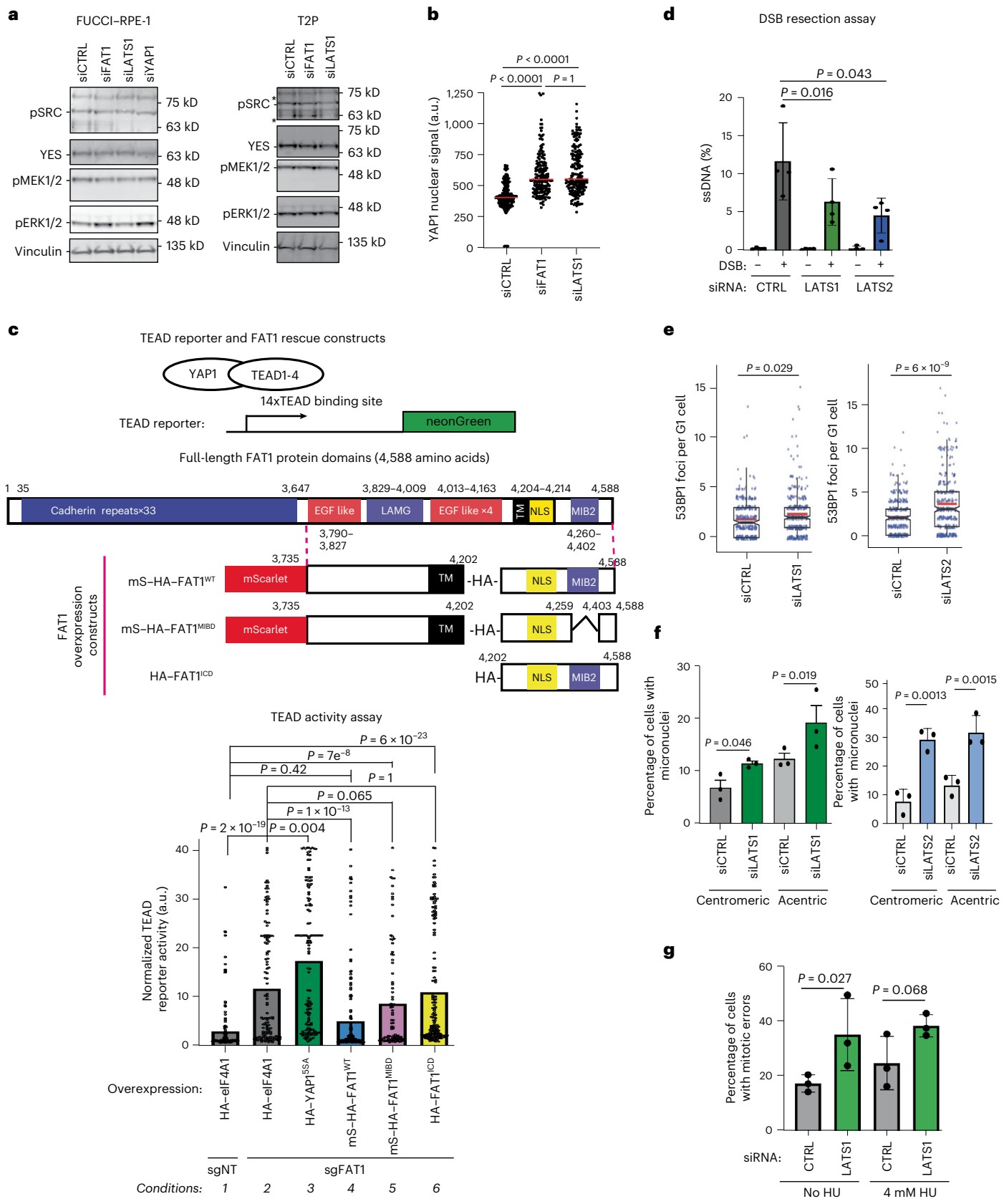

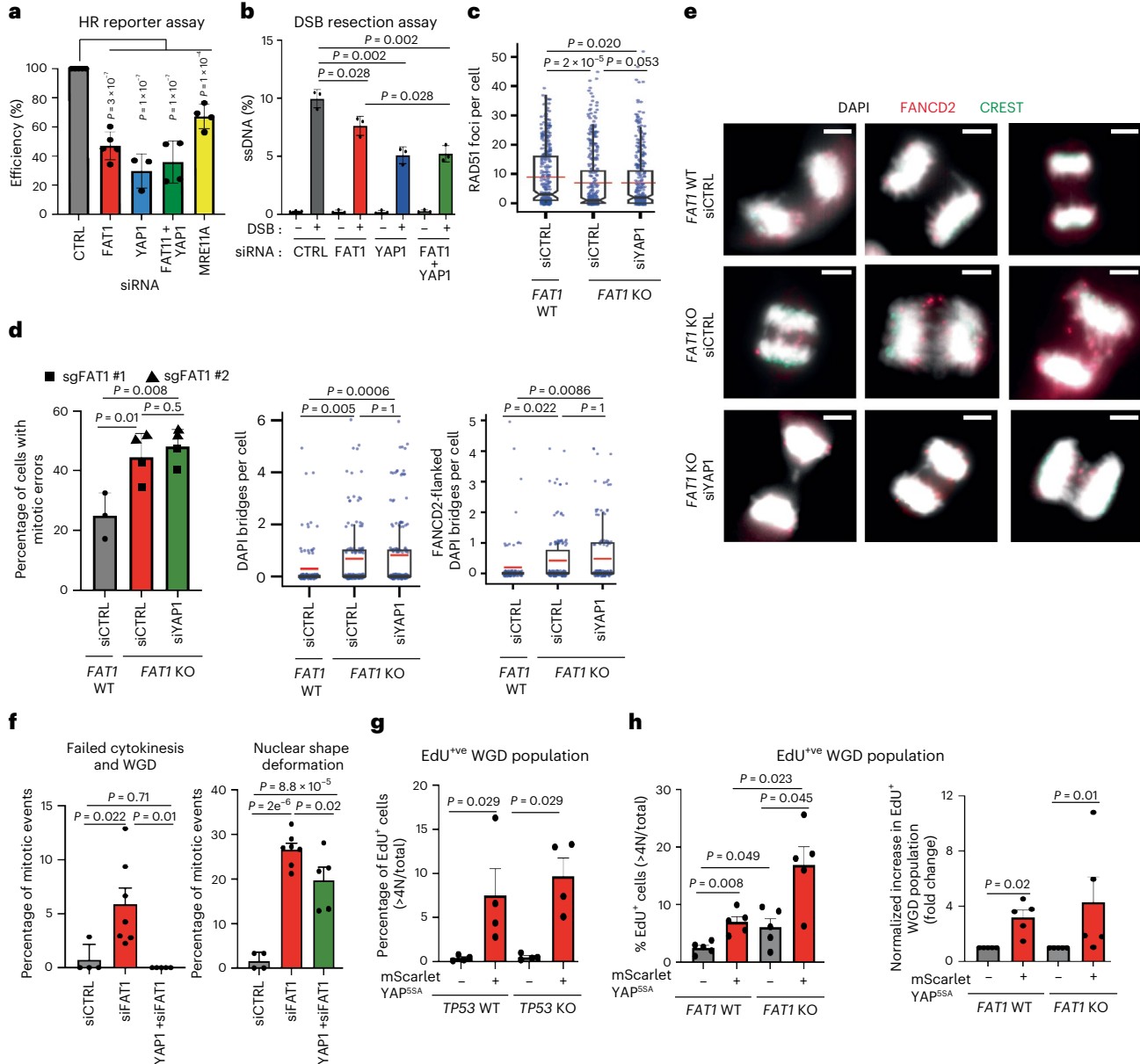

**Fig. 8 | Co-depletion of FAT1 and YAP1 reverses cytokinesis failure but not HR deficiencies. a**, Impact of *FAT1/YAP1* siRNA co-depletion in U2OS cells, as determined by DR-GFP HR reporter assay. *MRE11A* siRNA served as a positive control. The HR efficiencies are normalized to those of the control samples. Statistical significance was determined by one-way ANOVA with Holm–Šidák correction. The data represent means ± s.d. Biological replicates: $n = 5$ (siCTRL and siFAT1), $n = 3$ (siYAP1) and $n = 4$ (siFAT1 + siYAP1 and siMRE11A). **b**, ssDNA resection rates for *FAT1/YAP1* siRNA co-depletion, as determined by DIvA U2OS-AsiSI site-directed resection assay with the DSB site located at chr22:37194035, ssDNA measured 131 nucleotides from the DSB. Statistical significance was determined by one-way ANOVA with Holm–Šidák correction. The data represent means ± s.d. ($n = 3$ biological replicates). **c**, Box plots quantifying RAD51 foci formation in A549 cells following the loss of *FAT1*, or the combined loss of both *FAT1* and *YAP1*, after 6 Gy ionizing irradiation and 1 h recovery. The boxes represent interquartile ranges, the black and red lines represents median and mean values, respectively, and the ranges of the whiskers denote 1.5× the interquartile range. Over 380 cells were scored across three biological replicates. Statistical significance was determined by uncorrected Dunn's test. **d,e**, Plots (**d**) and representative images (**e**) illustrating the quantification of mitotic error rates in A549 cells after 24 h of aphidicolin treatment (0.2 μM). *FAT1* wild-type or knockout cells were transiently depleted of *YAP1* using RNAi. For the mitotic error analysis, statistical significance was determined by one-way ANOVA with Holm–Šidák multiple correction and the data represent means ± s.d. (biological

repeats: $n = 3$ (*FAT1* WT) and $n = 4$ (*FAT1* knockout)). For the DAPI bridge and Fanconi anaemia complementation group D2 (FANCD2)-flanked DAPI bridge analyses, the red lines represent mean values, the boxes represent interquartile ranges and the ranges of the whiskers denote 1.5× the interquartile range, and statistical significance was determined by Dunn's test with Bonferroni correction. Over 100 mitotic cells were scored across three biological replicates. Scale bars, 5 μM. **f**, Results of live-cell imaging analysis, showing that *FAT1/YAP1* double siRNA knockdown in TERT–RPE-1 cells fully rescued the failed cytokinesis and WGD introduced by *FAT1* knockdown (left) but only partially ameliorated the nuclear shape deformation (right). Statistical significance was determined by one-way ANOVA. At least five biological replicates were quantified per condition. Biological repeats: $n = 4$ (siCTRL), $n = 7$ (siFAT1) and $n = 5$ (siFAT1 + siYAP1). The data represent means ± s.e.m. **g**, Histogram illustrating the WGD populations in *TP53* wild-type versus knockout RPE-1 cells with or without transient mScarlet–YAP[SSA] transfection. The data represent means ± s.e.m. Statistical significance was determined by two-sided Mann–Whitney test ($n = 4$ biological repeats). **h**, Histograms illustrating the total (left) and normalized (right) EdU[+] WGD populations in *FAT1* wild-type versus knockout PC9 cells, with or without transient mScarlet–YAP[SSA] transfection. The data represent means ± s.e.m. ($n = 5$ biological repeats). Statistical significance was determined by repeated measures one-way ANOVA with Benjamini–Hochberg correction (left) or Friedman test (right). KO, knockout.

modulation of nuclear shape, which can affect three-dimensional genome organization[56,64–66]. We confirmed that depletion of *LATS1* or *LATS2* can significantly affect HR efficiency and contribute to CIN[64]. We discovered that FAT1—at least in part through its intracellular C-terminal domain—is involved in the DDR and represses CIN through the modulation of YAP1, whereas its interaction with YAP1 is dependent on the interaction between FAT1 and the E3 ligase MIB2 (ref. 53). Notably, despite being able to reconstitute HR-related functions, overexpression of the HA–FAT1[ICD] construct, devoid of the transmembrane domain, failed to re-repress TEAD transcriptional activity in *FAT1*-knockout cells. In line with previous reports that HRD results in mitotic defects and structural CIN[39,55], *FAT1/YAP1* co-depletion failed to rescue both the HRD-related phenotypes and the formation of chromosomal bridges, suggesting that *FAT1* loss contributes to HRD and structural CIN in a YAP1-independent manner. It has been reported that FAT1 can interact with MST1 and LATS1 (ref. 49), which can influence HR efficiency through their interaction with ATR and RASSF1A to control BRCA2 recruitment at DNA damage sites[64,67].

Conversely, *FAT1/YAP1* co-depletion rescued both cytokinesis failure and WGD. Strikingly, overexpression of constitutively active mScarlet–YAP1[5SA] was sufficient to drive WGD and increased numerical CIN. Taken together, our data reveal that *FAT1* alterations may contribute to HRD, CIN and WGD through two distinct mechanisms (Extended Data Fig. 10h).

Multiple studies have suggested that WGD accelerates cancer evolution and promotes intratumour heterogeneity, which promotes targeted therapy resistance[6,12,46,47]. Here, using PC9 cells that are sensitive to osimertinib as an experimental model, we illustrate that *FAT1* loss, through elevated genomic instability, may provide the evolutionary advantages that fuel targeted therapy resistance. Taken together, genomic observations and experimental results suggest a role for early positive selection of *FAT1* alterations in LUSC tumorigenesis, potentially by driving cancer evolution and WGD through elevated CIN.

Recent studies have also identified that the YAP/TEAD signalling axis promotes therapy resistance[68]. In addition to *FAT1*, three other genes identified through our DDR and CIN screens—namely *RAD21*, *BAP1* and *NCOA6*—have also been demonstrated to modulate the Hippo pathway[69–71]. Although *YAP1* is not amplified in the TRACERx dataset, amplification of the *WWTR1/TAZ* genomic locus (3q25.1) and loss of the *LATS2* genomic locus (13q12) are prevalent in LUSC[3,51]. Consistent with other studies[61,68], our data highlight the importance of *FAT1* and Hippo pathway dysregulation in genome instability, targeted therapy resistance and cancer evolution.

## Online content

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

Wei-Ting Lu [1] ✉, Lykourgos-Panagiotis Zalmas[1], Chris Bailey [1,10], James R. M. Black [1,2,3,10], Carlos Martinez-Ruiz[2,3,10], Oriol Pich [1,10], Francisco Gimeno-Valiente[2,3], Ieva Usaite[1,2,3], Alastair Magness [1], Kerstin Thol[2,3], Thomas A. Webber[1], Ming Jiang [1], Rebecca E. Saunders[1], Yun-Hsin Liu[2,3], Dhruva Biswas[1,2,3], Esther O. Ige [1], Birgit Aerne[1], Eva Grönroos [1], Subramanian Venkatesan [2,3], Georgia Stavrou [2,3], Takahiro Karasaki[1,2,3,4], Maise Al Bakir[1,2,3], Matthew Renshaw [1], Hang Xu [1], Deborah Schneider-Luftman [1], Natasha Sharma[2,3], Laura Tovini [5], TRACERx Consortium*, Mariam Jamal-Hanjani[1,2,3], Sarah E. McClelland [5], Kevin Litchfield[2,3], Nicolai J. Birkbak [6,7], Michael Howell [1], Nicolas Tapon [1], Kasper Fugger [3], Nicholas McGranahan[2,3], Jiri Bartek [8,9] ✉, Nnennaya Kanu [2,3] ✉ & Charles Swanton [1,2,3] ✉

[1]The Francis Crick Institute, London, UK. [2]CRUK Lung Cancer Centre of Excellence, London, UK. [3]University College London Cancer Institute, London, UK. [4]Department of Thoracic Surgery, Respiratory Center, Toranomon Hospital, Tokyo, Japan. [5]Barts Cancer Institute, Queen Mary University of London, London, UK. [6]Department of Molecular Medicine, Aarhus University Hospital, Aarhus, Denmark. [7]Department of Clinical Medicine, Aarhus University, Aarhus, Denmark. [8]Danish Cancer Society Research Centre, Copenhagen, Denmark. [9]Division of Genome Biology, Department of Medical Biochemistry and Biophysics, Science for Laboratory, Karolinska Institutet, Solna, Sweden. [10]These authors contributed equally: Chris Bailey, James R. M. Black, Carlos Martinez-Ruiz, Oriol Pich. *A list of authors and their affiliations appears at the end of the paper. ✉e-mail: wei-ting.lu@crick.ac.uk; jb@cancer.dk; n.kanu@ucl.ac.uk; charles.swanton@crick.ac.uk

**TRACERx Consortium**

Wei-Ting Lu[1], Chris Bailey[1,10], James R. M. Black[1,2,3,10], Carlos Martinez-Ruiz[2,3,10], Oriol Pich[1,10], Francisco Gimeno-Valiente[2,3], Ieva Usaite[1,2,3], Kerstin Thol[2,3], Dhruva Biswas[1,2,3], Eva Grönroos[1], Georgia Stavrou[2,3], Takahiro Karasaki[1,2,3,4], Maise Al Bakir[1,2,3], Mariam Jamal-Hanjani[1,2,3], Nicolai J. Birkbak[6,7], Nicholas McGranahan[2,3], Nnennaya Kanu[2,3] & Charles Swanton[1,2,3]

A full list of members and their affiliations appears in the Supplementary Information.

## Methods

### Ethical approval

The TRACERx study (NCT01888601; Clinicaltrials.gov) is sponsored by University College London (UCL/12/0279) and has been approved by an independent Research Ethics Committee (13/LO/1546). TRACERx is funded by Cancer Research UK (CRUK; C11496/A17786) and coordinated through the CRUK and University College London Cancer Trials Centre.

### Cell line, cell culture, transfection and CRISPR knockout generation

H1944, H1650, H1792, HCC4006, A549 (CCL-185) and U2OS (HTB-96) cells were obtained from Cell Services at the Francis Crick Institute. The FUCCI–H2B-mTurquoise–RPE-1 cells have been described previously[44] and were a kind gift from J. Diffley (Francis Crick Institute). The HA–ER–AsiSI–U2OS cells have been described previously[28] and were a kind gift from G. Legube (Paul Sabatier University). The ER–KRAS–V12–T2P cells (referred to as T2P) have been described previously[38] and were a kind gift from J. Downward (Francis Crick Institute). The TP53 wild-type and TP53-knockout RPE-1 cells have been described previously[74]. The HCT116 iRFP cell lines have been described previously[75] and were a kind gift from K. Vousden (Francis Crick Institute). All of the cell lines were maintained in either McCoy's 5A Medium, Dulbecco Modified Eagle's Medium (DMEM) or RPMI 1640 Medium (all from Thermo Fisher Scientific) fortified with 10% foetal bovine serum in the presence of 1% penicillin–streptomycin (Thermo Fisher Scientific). Cells were grown at 37 °C under 5% $CO_2$. All cell lines used were negative for Mycoplasma contamination and are frequently tested in-house at the Francis Crick Institute. DNA was transfected using GeneJuice (EMD Millipore) according to the manufacturer's instructions.

CRISPR knockout cell lines were generated using a single-cell cloning approach. To minimize the off-target effect, CRISPR plasmids and guides were transiently transfected using GeneJuice (EMD Milipore) and underwent puromycin selection for 5 days before single-cell sorting to generate knockout clones. Single-cell clones were then cultured, validated and frozen. A549 clones were generated with two individual guides (sgFAT1#1 (AAACCCGGGAAGTCGAAGTCCTTGC) and sgFAT1#2 (AAACACGCTGGATGTGTAATGTAAC)), whereas PC9 clones were generated with both guides. The sgNT sequence as follows: GCGAGGTATTCGGCTCCGCG.

### RNAi

A list of the siRNAs used is provided in Supplementary Table 1. Dharmacon ON-TARGETplus siRNA pools (Horizon Discovery) were used for high-content screening, HAC assay and RAD51 foci experiments in H1994 cells (Extended Data Figs. 1 and 2b,c and Supplementary Fig. 4). Ambion Silencer Select siRNA (Thermo Fisher Scientific) was used for all of the other RNAi experiments, except siRNA against MRE11A (L-009271-00-0005; Dharmacon; Fig. 8a). siRNA transfection was carried out using Lullaby reagent (OZ Biosciences) or Lipofectamine RNAiMAX Transfection Reagent (Thermo Fisher Scientific) according to the manufacturers' instructions.

### ssDNA resection and translocation assays

ssDNA resection assays were carried out as described previously[28]. Briefly, DIvA HA–ER–AsiSI–U2OS cells were plated overnight and siRNAs were transfected as described previously. Samples were subjected to a 4 h incubation in 300 nM 4-hydroxytamoxifen. Genomic DNA was extracted using the DNeasy Blood & Tissue kit (69504; Qiagen). For every 500 ng genomic DNA used, five units of RNase H1 (M0297; New England Biolabs) were added at 37 °C for 15 min. In vitro, restriction digestion with the BanI restriction enzyme was performed to assay for the presence of ssDNA around break sites. Then, 200 ng of samples were digested with 16 units of BanI at 37 °C for 12 h (New England Biolabs).

The following primers were used for the assay: 131-nucleotide forward (ACCATGAACGTGTTCCGAAT), 131-nucleotide reverse (GAGCTCCGCAAAGTTTCAA), 851-nucleotide forward (ACAGATCCAGAGCCACGAAA) and 851-nucleotide reverse (CCCACTCTCAGCCTTCTCAG),

The percentage of ssDNA generated by DNA resection was determined by quantitative PCR. ΔCT was defined as the difference in average cycles between a given digested sample and its undigested counterpart. To calculate the percentage of ssDNA, the following equation was used:

$$\text{Percentage of ssDNA} = 1/[2^{(\Delta CT-1)} + 0.5] \times 100.$$

For the semi-quantitative DNA translocation assay[40], 150 ng genomic DNA was used to perform the PCR reactions using the following primers: TRIM37 forward (AATTCGCAAACACCAACCGT), TRIM37 reverse (TCTGAAGTCTGCGCTTTCCA), MIS12 forward (GACTGGCATAAGCGTCTTCG), control chr1_82844750 forward (AGCACATGGGATTTTGCAGG) and control chr1_82844992 reverse (TTCCCTCCTTTGTGTCACCA).

Legitimate and illegitimate re-joining frequencies between MIS12 and TRIM37 (chr17_5390209 and chr17_57184285) were tested by PCR. The uncut site at chromosome 1 was used as a negative control.

### Plasmid construction

A gene fragment containing the C-terminal intracellular domain of FAT1 transcript (NM_005245.4) was ordered from Integrated DNA Technologies. The gene fragment was subsequently subcloned into the pCMV-SPORT6 expression plasmid containing a His-HA tag at the amino (N) terminus of the open reading frame. To generate the mScarlet–HA–FAT1^WT construct, gene fragments containing the mScarlet sequence and FAT1 sequences were ordered and ligated into the HA–FAT1^ICD construct using NEBuilder HiFi DNA Assembly (New England Biolabs). Similarly, deletion of the MIB2 binding domain and generation of the HA–FAT1^WT construct were achieved using NEBuilder HiFi DNA Assembly (New England Biolabs).

The TEAD transcriptional reporter comprises a multimerized TEAD-binding sequence (14xTBS), based on the sequence described by Schlegelmilch et al.[76], upstream of a minimal promoter and a nuclear localization signal in frame with four copies of mNeonGreen followed by a Myc-tag. The sequence components were assembled by GeneArt (Thermo Fisher Scientific) in a pDONOR221 backbone. The final construct was obtained by cloning the insert into pLenti X1 Zeo DEST using Gateway technology (Thermo Fisher Scientific). The YAP^SSA construct was obtained from Addgene (plasmid 27371)[77] and subcloned into a pCMV-SPORT6 expression plasmid containing a His-HA tag at the N terminus of the open reading frame. The His-HA-eIF4A1 plasmid was a kind gift from M. Bushell (CRUK Scotland Institute) and was documented by Meijer et al.[78].

All of the plasmids used in this study were then sequenced using Plasmidsaurus or Full Circle sequencing. The relevant sequences are included in Supplementary Information.

### DNA fibre assay

DNA fibre assays were performed as described in ref. 55. A549 and PC9 cells were incubated with 5-chloro-2′-deoxyuridine for 20 min, challenged with replication stress induced by 2 mM hydroxyurea incubation for 2 h and then labelled with 5-iodo-2′-deoxyuridine for 20 min.

### MCM7 loading assay

PC9 or RPE-1 cells were trypsinized and treated with modified CSK buffer (10 mM HEPES (pH 7.9), 100 mM NaCl, 3 mM $MgCl_2$, 1 mM EGTA, 300 mM sucrose, 1% bovine serum albumin (BSA), 0.2% Triton X-100 and 1 mM dithiothreitol) on ice for 5 min to remove soluble proteins. Cells were washed with 1% BSA in phosphate-buffered saline (PBS) and fixed in 4% paraformaldehyde (Thermo Fisher Scientific) for 10 min.

Cells were then washed, pelleted and permeabilized in 70% ethanol for 15 min. MCM7 staining was performed using a mouse anti-MCM7 antibody (sc-56324; Santa Cruz Biotechnology) followed by an Alexa Fluor 594 goat anti-mouse antibody (A11007; Invitrogen). DNA content was stained using 1 µg ml$^{-1}$ 4′,6-diamidino-2-phenylindole (DAPI) supplemented with 100 µg ml$^{-1}$ RNase A. Cell doublets were excluded from all of the analyses.

## Metaphase spreads

To assess numerical CIN, transfected cells were incubated with KaryoMAX Colcemid at 0.1 µg ml$^{-1}$ for 1 h, collected and incubated for 7 min at 37 °C in pre-warmed hypotonic solution (75 mM KCl). Cells were pelleted and repeatedly resuspended in freshly prepared fixation buffer (3:1 methanol:glacial acetic acid) and spun three times, after which the cells were dropped onto glass slides. For hybridization, the slides were ethanol dehydrated and subsequently incubated overnight at 37 °C in a humidified chamber with All Human centromeric probes (KBI-20000G; Leica) according to the manufacturer's instructions. Metaphase chromosome images were visualized using a 100× oil immersion objective on an Image Solutions DeltaVision microscope.

To assess chromosomal breakage and aberrations, metaphase spreads were prepared as previously described[79]. Briefly, 250,000 wild-type or *FAT1*-knockout cells were seeded in 6 cm dishes for one day before treatment with 0.2 µM aphidicolin for 24 h. Cells were treated with 0.2 µg ml$^{-1}$ colcemid for 1 h at 37 °C before collection by trypsinization and incubation with 5 ml pre-warmed hypotonic swelling buffer (75 mM KCl) in 15 ml tubes at 37 °C for 10 min. 1 ml freshly prepared fixation buffer (3:1 methanol:acetic acid) was added to the 15 ml tube and mixed by inversion. Cells were pelleted by centrifugation at 300*g* for 4 min at room temperature. About 500 µl supernatant was left in the tubes and the cells were slowly resuspended before the addition of 5 ml fixation buffer, mixing and incubation for 10 min at room temperature. The fixation step was repeated by pelleting the cells, resuspension and the addition of fixation buffer, as above. To spread the metaphases, the fixed cells were dropped on glass slides at a 45° angle from a 30 cm height. Slides were left to air dry before staining with Giemsa (7% in 10 mM PIPES (pH 6.8)) for 10 min at room temperature, then mounted with DPX. Metaphase chromosome images were visualized using a 100× oil immersion objective on an upright Zeiss Axio Imager fluorescence microscope equipped with Volocity software.

## ImageStream fluorescence in situ hybridization

Transfected cells were processed for fluorescence in situ hybridization in suspension, as described elsewhere[80]. Briefly, following transfection, cells were harvested and fixed with freshly prepared 3:1 methanol:glacial acetic acid. Cells were subsequently hybridized with chromosome 15 satellite enumeration probe (LPE015G; CytoCell) performed in a thermocycler under the following conditions: 65 °C (2 h pre-annealing), 80 °C (5 min) and 37 °C (16 h) before analysis on an ImageStream X Mark II (Amnis).

## Definition of *FAT1* driver mutation in the TRACERx 421 cohort

*FAT1* driver mutations in the TRACERx 421 cohort were defined as described in ref. [81]. Briefly, *FAT1* non-synonymous variants that were found to be deleterious (either stop–gain or predicted deleterious in two of the three computational approaches applied (that is, Sift[82], Polyphen[83] and MutationTaster[81])) were classified as a driver mutation. Any tumour with any such mutations in *FAT1* was considered to have a *FAT1* loss.

## Determination of WGD events in TRACERx 421

The WGD status for each tumour was estimated in two steps, as described in ref. [81]. Briefly, if the genome-wide copy number of the major allele was ≥2 across at least 50% of the genome, this was assumed to reflect a WGD event[13,84]. A major allele copy number ≥3 across at

least 50% genome was assumed to reflect two WGD events. Then, we leveraged additional information from the estimated copy number of mutations using a novel tool, ParallelGDDetect, available as an R package (https://github.com/amf71/ParallelGDDetect).

## TRACERx 421 cohort

The TRACERx 421 cohort consisted of 233 males and 188 females (421 patients in total), corresponding to a 55:45 male:female ratio. 93% of the cohort was from a White ethnic background and the mean age of the patients was 69 years, ranging between 34 and 92 years. Written informed consent was obtained. None of the patients were compensated for their involvement in the study.

## Mutation clonality in TRACERx 421

Reconstructed phylogenetic trees were used to classify mutations based on their inferred phylogenetic cancer cell fraction (phyloCCF)[81]. We classified as clonal in each tumour region every cluster whose 95% confidence interval of the phyloCCF of its mutations overlapped with the 95% confidence interval of the phyloCCF of the mutations in the mutation cluster within the trunk node of the tree (a minimum threshold of 0.9 was used for the left side of the 95% confidence interval on truncal mutations). Second, we defined as subclonal every mutation cluster in a tumour region whose mean phyloCCF across the corresponding mutations in that region was greater than 0 and not clonal (that is, the mutation cluster did not pass the previous tests). Lastly, any remaining mutation cluster was defined as absent in a tumour region otherwise. Furthermore, clonal mutations were defined as early or late depending on whether they occurred before or after WGD. A clonal mutation copy number consistent with the WGD ploidy was therefore considered early, or late, otherwise.

## dN/dS analyses

The dN/dS point mutation estimate for *FAT1* was calculated using the dndscv and geneci functions in the dNdScv[29] R package. dndscv was run on different subsets of mutations (clonal early versus clonal late) from different subsets of samples (LUAD versus LUSC). The 95% confidence interval of the dN/dS estimate was obtained using the geneci function.

## Enrichment analyses

All enrichment analyses were performed by counting the tumours in each category (for example, LUAD versus LUSC, WGD versus no WGD or early clonal versus late clonal) and performing a chi-squared test.

## SCNA detection and MSAI

The identification of genome-wide allele-specific copy number states for multiregion WES is described in ref. [81]. Briefly, logR data were calculated using VarScan 2, GC corrected using a wave-pattern GC correction method developed by Cheng and colleagues[85] and processed using ASCAT (version 2.3)[86]. Sequenza (version 2.1.2)[87] was utilized to offer supplementary assessments of tumour purity and ploidy for further examination. ASCAT was presented with the automatically selected models for ploidy and purity from either Sequenza or ASCAT, which were manually reviewed to generate SCNA profiles for each tumour area. Samples that had an estimated tumour purity below 10% were excluded. The data on ploidy, purity and copy number segmentation were fed into a multi-sample SCNA estimation approach[51] to create a consistently minimal segmentation and genome-wide evaluation of LOH, as well as loss, gain and amplification copy number states in relation to sample ploidy.

The input SCNA profiles were used to identify allelic imbalance, which was subsequently employed to phase heterozygous single-nucleotide polymorphisms and re-estimate allele-specific copy numbers. Furthermore, MSAI, which occurs when SCNAs disrupt the same genomic region but affect different parental alleles within distinct tumour subclones, was detected as previously described[57].

We identified a subset of these MSAI events as parallel SCNA events that refer to the same class of event (gain/amplification or loss/LOH) in multiple samples from a given tumour but with major alleles from distinct haplotypes in the samples demonstrating the event. We then quantified the proportion of tumours where MSAI events occurred per cytoband, providing an empirical *P* value for cytoband *4q35.2*.

### EdU incorporation assay and cyclin B loading assay

The Click-iT Plus EdU Alexa Fluor 647 Flow Cytometry Assay kit (Thermo Fisher Scientific) was used for EdU incorporation assays. Cells were incubated with 10 μM EdU for 30 min before being fixed and processed per the manufacturer's instructions. Cyclin B1 antibody was obtained from Abcam (ab32053; Abcam). DNA was stained in staining buffer (1 μg ml$^{-1}$ DAPI and 100 μg ml$^{-1}$ RNase A in PBS). Cell doublets were excluded from all of the analyses.

### Microscopy and DNA damage foci studies

**DDR foci and DNA fibre assays.** The preparation of slides for DDR immunofluorescence foci studies was carried out as described in ref. [88]. Briefly, cells were pre-seeded on glass coverslips, subjected to Cs-137 ionizing irradiation and allowed to recover. Cells were then washed once with PBS and pre-extracted with CSK buffer (100 mM NaCl, 300 mM sucrose, 3 mM MgCl$_2$, 10 mM PIPES (pH 6.8), 10 mM β-glycerol phosphate, 50 mM NaF, 1 mM EDTA, 1 mM EGTA, 5 mM Na$_3$VO$_4$ and 0.5% Triton X-100). Cells were then washed once with CSK buffer and fixed with 4% paraformaldehyde on ice for 20 min. Samples were then washed three times with 0.1% Tris-buffered saline (TBS)-Tween, blocked with 10% goat serum for 1 h, washed twice with 0.1% TBS-Tween and incubated with primary antibody overnight at 4 °C (see Supplementary Table 1). The samples were subsequently washed and incubated with secondary antibodies (Alexa Fluor 488/594/647; Invitrogen) with DAPI in 1% goat serum for 1 h at room temperature. Samples were then washed with 0.1% TBS-Tween and mounted with Vectashield (H1200; Vector Laboratories). The samples were then double-blinded for quantification.

Images were acquired using a Zeiss Axio Imager M1 microscope with a 40×/1.3 NA Plan-Neofluar or 63×/1.4 NA Plan-Apochromat objective, equipped with an ORCA-spark CMOS camera (Hamamatsu) and pE-300white LED light source (CoolLED) and controlled by Micro-Manager 2.0 software[89]. Images were processed and counted in an unbiased way using the FindFoci ImageJ plugin[90].

**Segregation errors.** For segregation errors, cells were fixed and permeablized in PTEMF buffer (4% paraformaldehyde, 0.2% Triton X-100, 20 mM PIPES and 2 mM MgCl$_2$) for 15 min at room temperature. The samples were washed three times in PBS, blocked for 1 h with 3% BSA with 0.2% Triton X-100 and incubated overnight with the relevant primary antibodies. The samples were subsequently washed and incubated with secondary antibodies (Alexa Fluor 488/594/647; Invitrogen) with DAPI in 1% goat serum for 1 h at room temperature. Samples were then washed with 0.1% TBS-Tween and mounted with Vectashield (H1200; Vector Laboratories). Images were acquired using a Nikon Ti2 microscope with a 60×/1.45 NA Plan Apo TIRF objective and 1.5× intermediate magnification, equipped with a Prime BSI sCMOS camera (Teledyne Photometrics), motorized XY stage with piezo Z axis (Applied Scientific Instrumentation) and Spectra X LED light engine (Lumencor) and controlled with Micro-Manager 2.0 software[89]. Z stacks were acquired (31 μm × 0.2 μm z steps) and deconvolved using Microvolution software with ten iterations.

**Live-cell widefield microscopy.** Live-cell widefield microscopy was performed using a Nikon Ti2 inverted microscope with a Perfect Focus System using 1.5× intermediate magnification, equipped with a Prime BSI sCMOS camera (Teledyne Photometrics), motorized XY stage with piezo Z axis (ASI) and Spectra X LED light engine (Lumencor) and

controlled with Micro-Manager 2.0 software[89]. Environmental conditions were maintained at 37 °C under 5% CO$_2$ using a chamber and CO$_2$ mixer (Okolab).

For live-cell imaging of the mitotic failure rate, the FUCCI–H2B-mTurquoise–RPE-1 cells were grown and transfected for 72 h on glass-bottomed plates (IBL) in DMEM culture media, as described above. Before imaging, the growth media was replaced with Fluoro-Brite DMEM (Thermo Fisher Scientific) with 10% foetal bovine serum to improve the image quality. To acclimatize to potential temperature fluctuations, the cells were allowed to settle for 1 h before imaging. Phase-contrast and fluorescence images were acquired with 20×/0.75 NA Plan Apo Ph2 objective every 20 min for over 72 h. The images were analysed using Fiji and the cells were tracked manually.

For the mitotic timing experiments, FUCCI–H2B-mTurquoise–RPE-1 cells were set up as above but imaged with a 40×/0.95 NA Plan Apo objective. Locations with high-level mVenus–geminin-expressing G2 cells were identified and recorded. Z stacks (25 μm × 0.5 μm z steps) of H2B-mTurquoise were taken at 5 min intervals over 12 h. Images were deconvolved over ten iterations using Microvolution software.

**Live-cell confocal microscopy.** Live-cell confocal microscopy was performed using either: (1) a Yokogawa CSU-W1 spinning-disk confocal scanhead on a Nikon Ti2 inverted microscope with a 40×/1.15 NA Apo objective and Prime 95B sCMOS camera (Teledyne Photometrics), controlled with NIS-Elements, at 37 °C under 5% CO$_2$ with an environmental chamber and gas mixer (Okolab); or (2) an NL5 slit-scanning confocal unit on a Nikon Eclipse Ti inverted microscope with a 40×/1.25 NA Apo objective and Quest qCMOS camera (Hamamatsu), controlled with Micro-Manager 2.0 software[89] at 37 °C under 5% CO$_2$ using a stage top incubator and gas mixer (Tokai Hit). Locations with high-level mVenus–geminin-expressing G2 cells were first identified and recorded. Z stacks (25 μm × 0.5 μm z steps) of H2B-mTurquoise were taken at 5 min intervals over 12 h.

For Extended Data Fig. 7h,i and Supplementary Videos 4 and 5, a higher resolution was required to delineate whether mitotic error precedes nuclear deformation. A mixture of widefield deconvolution microscopy (n = 28 cells (siCTRL) and n = 27 cells (siFAT1)) and confocal microscopy (n = 50 cells (siCTRL) and n = 27 cells (siFAT1)) techniques were utilized to obtain sufficient images for statistical analysis.

### Statistics and reproducibility

No statistical methods were used to predetermine the sample size. The statistical test types and biological *n* numbers used are detailed in the relevant figure captions. For immunofluorescence data, the numbers of cells sampled per biological repeat are documented accordingly. All of the data points shown are independent samples. For the experiments using statistical tests that assume normal distributions, we assumed that the data distribution was normal, but this was not formally tested.

### Reporting summary

Further information on research design is available in the Nature Portfolio Reporting Summary linked to this article.

## Data availability

The RNA-seq, WES and reduced representation bisulfite sequencing (RRBS) data (in each case from the TRACERx study) used during this study have been deposited in the European Genome-phenome Archive, which is hosted by the European Bioinformatics Institute and Centre for Genomic Regulation, under the accession codes EGAS00001006517 (RNA-seq), EGAS00001006494 (WES) and EGAS00001006523 (RRBS). Access is controlled by the TRACERx data access committee. The Genomics England lung cohort is part of the 100,000 Genomes Project whose data are held in a secure research environment and are only available to registered users. For further information on how to obtain

access, visit https://www.genomicsengland.co.uk/research/academic. Source data are provided with this paper.

## Code availability

The code used to determine genome doubling is available at https://github.com/amf71/ParallelGDDetect. The code used to process data and generate figures is available at https://doi.org/10.5281/zenodo.13884283 (ref. 91).

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

## Acknowledgements

C.M.-R. is supported by the Rosetrees Trust (M630) and Wellcome Trust. K.L. is funded by the UK Medical Research Council (MR/V033077/1), Rosetrees Trust and Cotswold Trust (A2437), Royal Marsden Cancer Charity (thanks to the Ross Russell Family and Macfarlanes donations), Melanoma Research Alliance and CRUK (C69256/A30194). M.J.-H. received the CRUK Career Establishment Award, as well as funding from CRUK, the International Association for the Study of Lung Cancer International Lung Cancer Foundation, the Lung Cancer Research Foundation, the Rosetrees Trust, the UK and Ireland Neuroendocrine Tumour Society, the National Institute for Health Research (NIHR) and the NIHR University College London Hospitals Biomedical Research Centre. N.J.B. is supported by the Lundbeck Foundation (R272-2017-4040), Aarhus University Research Foundation (AUFF-E-2018-7-14) and Novo Nordisk Foundation (NNF21OC0071483 and NNF23OC0085954). L.-P.Z., T. A.W. and E.G. received support from the ERC Consolidator Grant THESEUS (grant agreement no. 617844). E.G. was further funded by the ERC Advanced Grant PROTEUS (grant agreement no. 835297). N.M. is funded by the Wellcome Trust and Royal Society (grant number 211179/Z/18/Z), and also receives funding from CRUK (DRCPFA-Nov23/100003), the Rosetrees Trust, the NIHR Biomedical Research Centre at University College London Hospitals and the CRUK-funded University College London Experimental Cancer Medicine Centre. W.-T.L. is grateful for support from the Kuok Foundation. N.K. is supported by the Breast Cancer Research Foundation (BCRF 23-157), Rosetrees Trust and CRUK. J.B. is funded by the Danish Cancer Society (R322-A17482) and Novo Nordisk Foundation (NNF20OC0060590) and is grateful for the support. C.S. is a Royal Society Napier Research Professor (RSRP\R\210001). His work is supported by the Francis Crick Institute, which receives core funding from CRUK (CC2041), the UK Medical Research Council (CC2041) and the Wellcome Trust (CC2041). For the purpose of open access, C.S. has applied a CC BY public copyright licence to any author accepted manuscript version arising from this submission. C.S. is funded by CRUK (TRACERx (C11496/A17786), PEACE (C416/A21999) and the CRUK Cancer Immunotherapy Catalyst Network), the CRUK Lung Cancer Centre of Excellence (C11496/A30025), the Rosetrees Trust, the Butterfield and Stoneygate Trusts, the Novo Nordisk Foundation (ID16584), the Royal Society Professorship Enhancement Award (RP/EA/180007 and RF\ERE\231118), the NIHR University College London Hospitals Biomedical Research Centre, the CRUK–University College London Centre, the Experimental Cancer Medicine Centre, the US Breast Cancer Research Foundation (BCRF-22-157), the CRUK Early Detection and Diagnosis Primer Award (grant EDDPMA-Nov21/100034), the Mark Foundation for Cancer Research ASPIRE Award (grant 21-029-ASP) and the ASPIRE Phase II award (grant 23-034-ASP). C.S. is in receipt of an ERC Advanced Grant (PROTEUS) from the European Research Council under the European Union's Horizon 2020 research and innovation programme (grant agreement number 835297). The TRACERx study (NCT01888601; Clinicaltrials.gov) is sponsored by University College London (UCL/12/0279) and has been approved by an independent Research Ethics Committee (13/LO/1546). TRACERx is funded by CRUK (C11496/A17786) and coordinated through the CRUK and University College London Cancer Trials Centre, which has a core grant from CRUK (C444/A15953). We gratefully acknowledge the patients and relatives who participated in the TRACERx study. This research was made possible through access to data in the National Genomic Research Library, which is managed by Genomics England (a wholly owned company of the Department of Health and Social Care). The National Genomic Research Library holds data provided by patients and collected by the National Health Service (NHS) as part of their care, as well as data collected as part of their participation in research. The National Genomic Research Library is funded by the National Institute for Health Research and NHS England. The Wellcome Trust, CRUK and the Medical Research Council are involved in the research infrastructure of Genomic England. We thank G. Legube (Paul Sabatier University), J. Diffley (Francis Crick Institute), J. Downward (Francis Crick Institute) and K. Vousden (Francis Crick Institute) for providing materials. We thank J. Zeng, A. Bertolin, J. Diffley, T. Klockner, U. Baruchel (Francis Crick Institute), S.F. Bakhoum (Memorial Sloan Kettering Cancer Center)

and the members of the C.S. laboratory for discussion and insights. We thank the scientific platforms in the Francis Crick Institute—particularly Advance Light Microscopy, Cell Services, Flow Cytometry and High Throughput Screening—for support. We are grateful for assistance from the Flow Cytometry unit of the CRUK City of London Centre at the Barts Cancer Insitute and University College London Cancer Institute. This work was supported by the Francis Crick Institute, which receives core funding from CRUK (CC2041), the UK Medical Research Council (CC2041) and the Wellcome Trust (CC2041). This work was also supported by the CRUK Lung Cancer Centre of Excellence and the CRUK City of London Centre Award (C7893/A26233).

## Author contributions

W.-T.L., N.K., L.-P.Z. and C.S. conceived of the project concept. L.-P.Z. and M.H. designed the high-throughput screening experiments. L.-P.Z., M.J. and M.H. executed the high-throughput screening experiments. L.-P.Z., H.X., R.E.S. and M.H. analysed the high-throughput screening experiments. C.B., J.R.M.B., C.M.-R., O.P., F.G.-V., I.U., K.T., Y.-H.L., D.B., T.K., M.A.B., D.S.-L., N.S., K.L., N.J.B. and N.M. performed the genomics and statistical analyses. A.M. and M.R. contributed to the imaging analysis and imaging method development. L.T. and S.E.M. developed the ImageStream fluorescence in situ hybridization methods. W.-T.L., L.-P.Z., T.A.W., E.I., B.A., S.V., G.S., K.F. and N.K. carried out the experiments and developed the methods. L.-P.Z., E.G., S.E.M., K.L., N.J.B., N.T., K.F. and N.M. provided feedback on the experimental design and data analyses. M.J.-H. designed the PEACE and TRACERx study protocols. L.-P.Z., E.G., M.J.-H., S.E.M., N.T., K.F. and N.M. provided feedback on the manuscript. W.-T.L. and N.K. jointly coordinated and designed the experiments, performed the experiment analyses and wrote the manuscript. J.B. and C.S. provided strategic oversight and helped to write the manuscript. L.-P.Z., T. A.W. and E.G. received support from the ERC Consolidator Grant THESEUS (grant agreement no. 617844). E.G. was further funded by the ERC Advanced Grant PROTEUS, (grant agreement no. 835297).

## Funding

## Competing interests

D.B. reports personal fees from NanoString and AstraZeneca and has a patent (PCT/GB2020/050221) issued on methods for cancer prognostication. M.A.B. has consulted for Achilles Therapeutics. M.J.-H. has received funding from CRUK, the National Institutes of Health National Cancer Institute, the International Association for the Study of Lung Cancer International Lung Cancer Foundation, the Lung Cancer Research Foundation, the Rosetrees Trust, the UK and Ireland Neuroendocrine Tumour Society and the NIHR. M.J.-H. has consulted for, and is a member of, the Achilles Therapeutics Scientific Advisory Board (SAB) and Steering Committee, has received speaker honoraria from Pfizer, Astex Pharmaceuticals, Oslo Cancer Cluster, Bristol Myers Squibb and Genentech. M.J.-H. is listed as a co-inventor on a European patent application relating to methods for the detection of lung cancer (PCT/US2017/028013). This patent has been licensed to commercial entities and, under the terms of their employment, M.J.-H. is due a share of any revenue generated from such license(s). M.J.-H. is also listed as a co-inventor on a GB priority patent application (GB2400424.4) with title 'Treatment and prevention of lung cancer'. K.L. has a patent pending on indel burden and checkpoint inhibitor response, has received speaker fees from Roche Tissue Diagnostics and research funding from the CRUK Therapeutic Discovery Laboratories–Ono Pharmaceutical–LifeArc alliance and Genesis Therapeutics and has held consulting roles with Ellipses Pharma,

Monopteros and Kynos Therapeutics. N.J.B. is listed as a co-inventor on a patent related to the identification of responders to cancer treatment (PCT/GB2018/051912), has submitted a patent application (PCT/GB2020/050221) on methods for cancer prognostication and has a patent on methods for predicting anti-cancer response (US14/466,208). N.M. has stock options in, and has consulted for, Achilles Therapeutics and holds a European patent in determining human leukocyte antigen (HLA) LOH (PCT/GB2018/052004), has a patent pending on determining HLA disruption (PCT/EP2023/059039) and is a co-inventor on a patent on the identification of responders to cancer treatment (PCT/GB2018/051912). N.K. acknowledges grants from AstraZeneca. C.S. acknowledges grants from AstraZeneca, Boehringer Ingelheim, Bristol Myers Squibb, Pfizer, Roche VENTANA, Invitae (previously ArcherDX; a collaboration on minimal residual disease sequencing technologies), Ono Pharmaceutical and Personalis. He is Chief Investigator for the AZ MeRmaiD-1 and -2 clinical trials and is the Steering Committee Chair. He is also Co-Chief Investigator of the NHS Galleri Trial funded by GRAIL and a paid member of GRAIL's SAB. He receives consultant fees from Achilles Therapeutics (and is also a SAB member), Bicycle Therapeutics (and is also a SAB member), Genentech, Medicxi, the China Innovation Centre of Roche (formerly the Roche Innovation Centre – Shanghai); Metabomed, until July 2022, Relay Therapeutics (and is also a SAB member), SAGA Diagnostics (and is also a SAB member) and the Sarah Cannon Research Institute. C.S. has received honoraria from Amgen, AstraZeneca, Boehringer Ingelheim, Bristol Myers Squibb, GlaxoSmithKline, Illumina, MSD, Novartis, Pfizer and Roche VENTANA. C.S. has previously held stock options in ApoGen Biotechnologies and GRAIL and currently has stock options in EPIC Bioscience, Bicycle Therapeutics, Relay Therapeutics and Achilles Therapeutics. C.S. is also a co-founder of Achilles Therapeutics. C.S. declares patents and patent applications relating to methods for the detection of lung cancer (PCT/US2017/028013), neoantigen targeting (PCT/EP2016/059401), the identification of patent response to immune checkpoint blockade (PCT/EP2016/071471), the identification of patients who respond to cancer treatment (PCT/GB2018/051912), the determination of HLA LOH (PCT/GB2018/052004), the prediction of survival rates of patients with cancer (PCT/GB2020/050221) and methods and systems for tumour monitoring (PCT/EP2022/077987). C.S. is an inventor on a European patent application (PCT/GB2017/053289) relating to assay technology to detect tumour recurrence. This patent has been licensed to a commercial entity and, under the terms of their employment, C.S. is due a share of any revenue generated from such license(s). The remaining authors declare no competing interests.

## Additional information

**Extended data** is available for this paper at https://doi.org/10.1038/s41556-024-01558-w.

**Correspondence and requests for materials** should be addressed to Wei-Ting Lu, Jiri Bartek, Nnennaya Kanu or Charles Swanton.

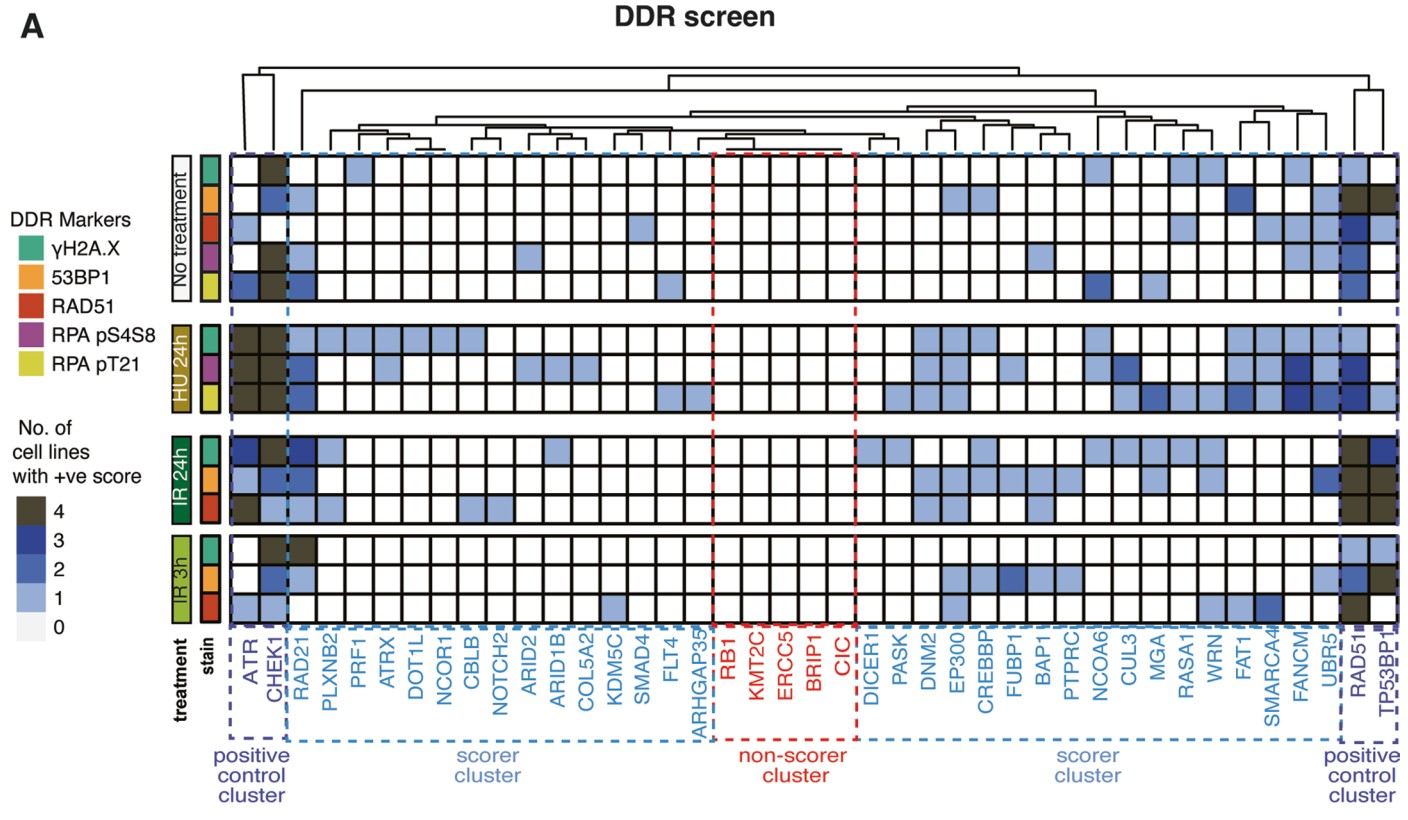

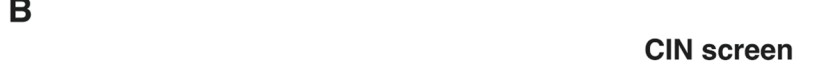

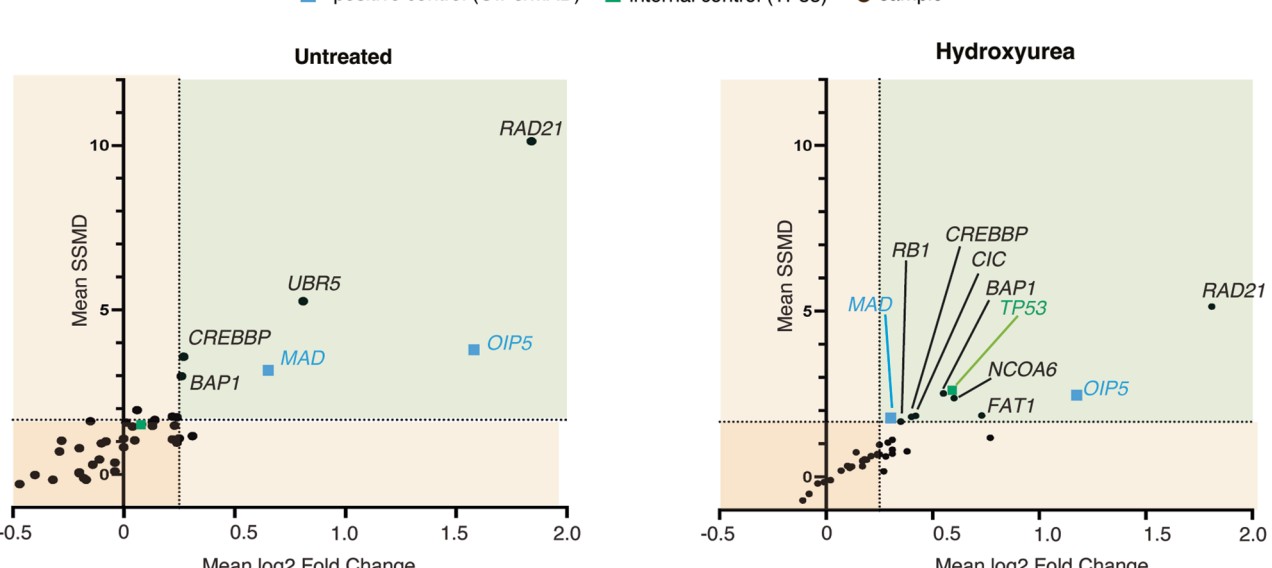

**Extended Data Fig. 1 | Summary of the DDR and CIN loss-of-function screen of genome-doubling-associated drivers from the TRACERx 100 cohort.** **a**, Unsupervised clustering of the results from the DDR screen of 37 TRACERx drivers in 4 LUAD cell lines. Each row represents a DDR foci readout under a given genotoxic stress. Random forest classifiers were used to identify genes that had feature scores similar to positive controls; known DDR genes such as *ATR, CHEK1, RAD51* and *TPS3BP1* are highlighted. Hierarchical clustering was used to calculate

the Euclidean distance between TRACERx drivers and the known DDR positive control genes. **b**, HAC assay was used to assess the impact of gene depletion and mild replication stress on CIN, through modelling chromosome loss using an HT-1080 reporter system. Cells were processed and data was analysed as described in Methods. Data are displayed as scatter plots of log fold changes (LFCs) and Strictly Standardized Mean Difference (SSMDs), highlighting negative and positive control genes, and genes that passed the hit threshold (green area).

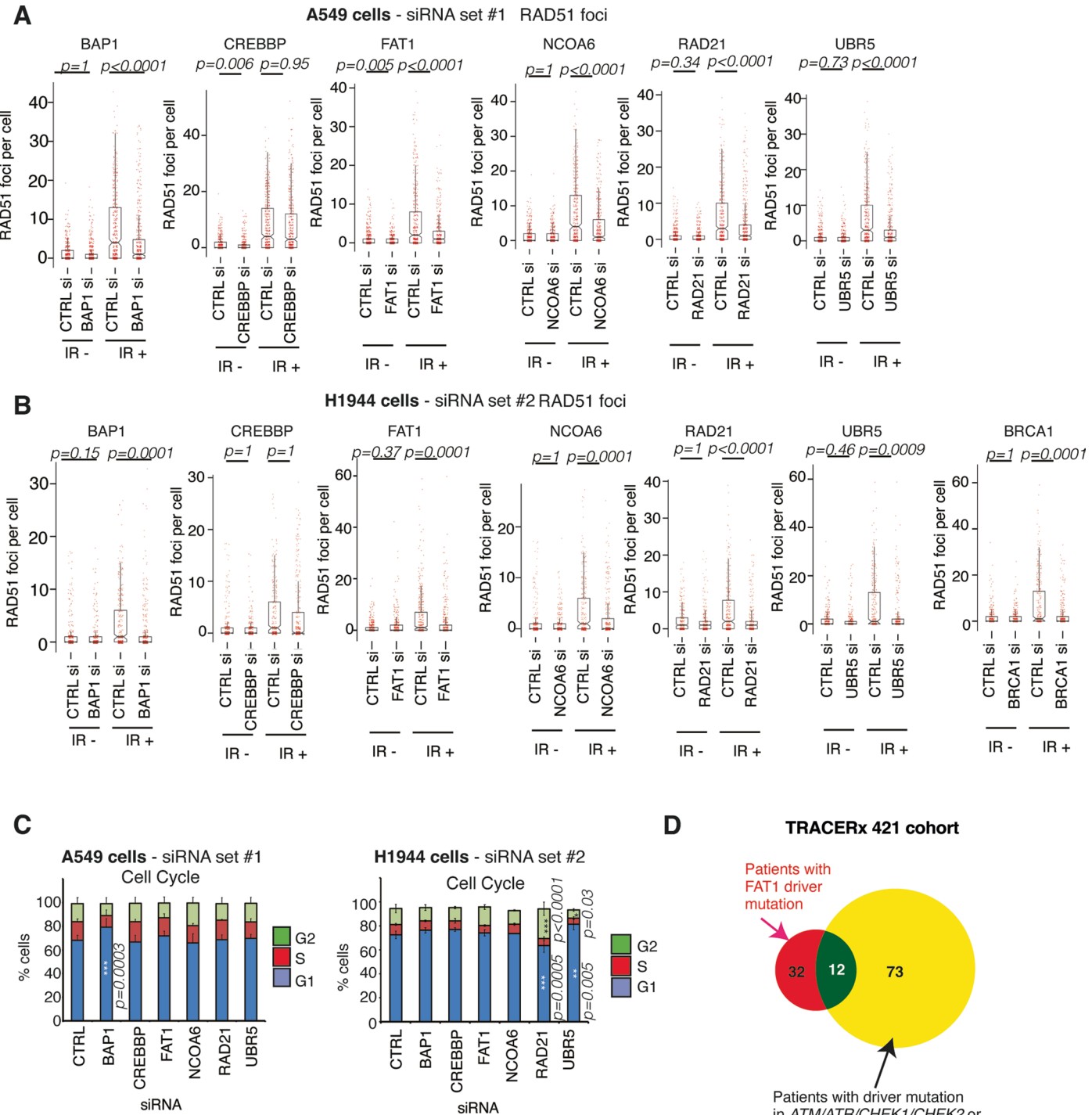

**Extended Data Fig. 2 | Depletion of several TRACERx driver genes reduces RAD51 foci formation post-ionizing radiation. a-b**, Box plots quantifying RAD51 foci in the presence and absence of 6Gy IR in A549 (A) and H1944 (B) cells. Cells were irradiated and allowed 1h recovery. The boxes represent the interquartile range, horizontal lines represent the median and the whiskers range denotes 1.5 x interquartile ranges, n=3 biological replicates, >150 cells quantified per biological replicate, Kruskal-Wallis test with Dunn's multiple comparison tests. **c**, Histograms representing cell cycle profiles following siRNA knockdown of 6 candidate DDR+CIN genes in A549 (left) and H1994 (right) cells, using 2 sets of different siRNAs, n=3 biological repeats, 2-way ANOVA with Sidak corrections. **d**, Driver mutation distribution of *FAT1*, and HR-related genes (*ATM-CHEK2*, *ATR-CHEK1*, and members of *FA/BRCA* pathway) in the TRACERx 421 cohort.

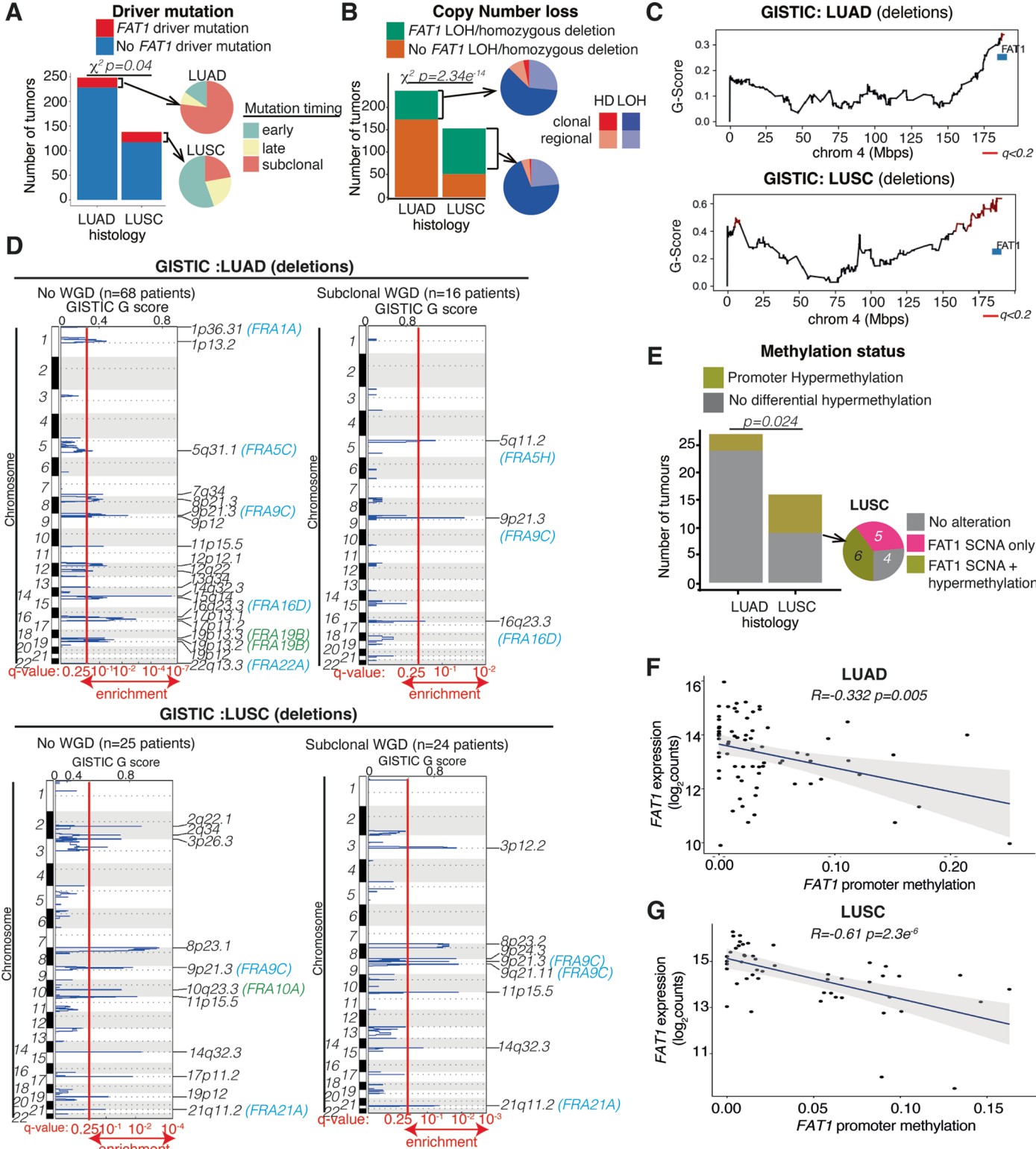

**Extended Data Fig. 3 | FAT1 mutation, copy number loss and hypermethylation are enriched in the TRACERx421 cohort. a,** FAT1 driver mutations are enriched in LUSC tumours in the TRACERx421 LUSC cohort, and mutations are timed early in tumor evolution. **b,** Graph showing the frequency of FAT1 SCNA loss in the TRACERx 421 cohort. **c,** GISTIC analysis demonstrating FAT1 genomic loci (4q35.2) SCNA loss is positively selected in both the LUAD and LUSC TRACERx cohort. **d,** Summary of GISTIC score analysis of LUAD and LUSC

TRACERx tumors from Fig. 2b segregated by WGD status. Patients with no WGD and subclonal WGD are included. SCNA loci overlapping with common or rare chromosome fragile sites[90,91] were annotated (common fragile sites=blue, rare fragile sites=green). **e,** FAT1 promoter hypermethylation is enriched in TRACERx LUSC tumors. Notably, FAT1 promoter hypermethylation also co-occurs with FAT1 SCNA loss. **f-g,** Correlation between FAT1 promoter hypermethylation and expression levels in LUAD (**f**) and LUSC tumors (**g**).

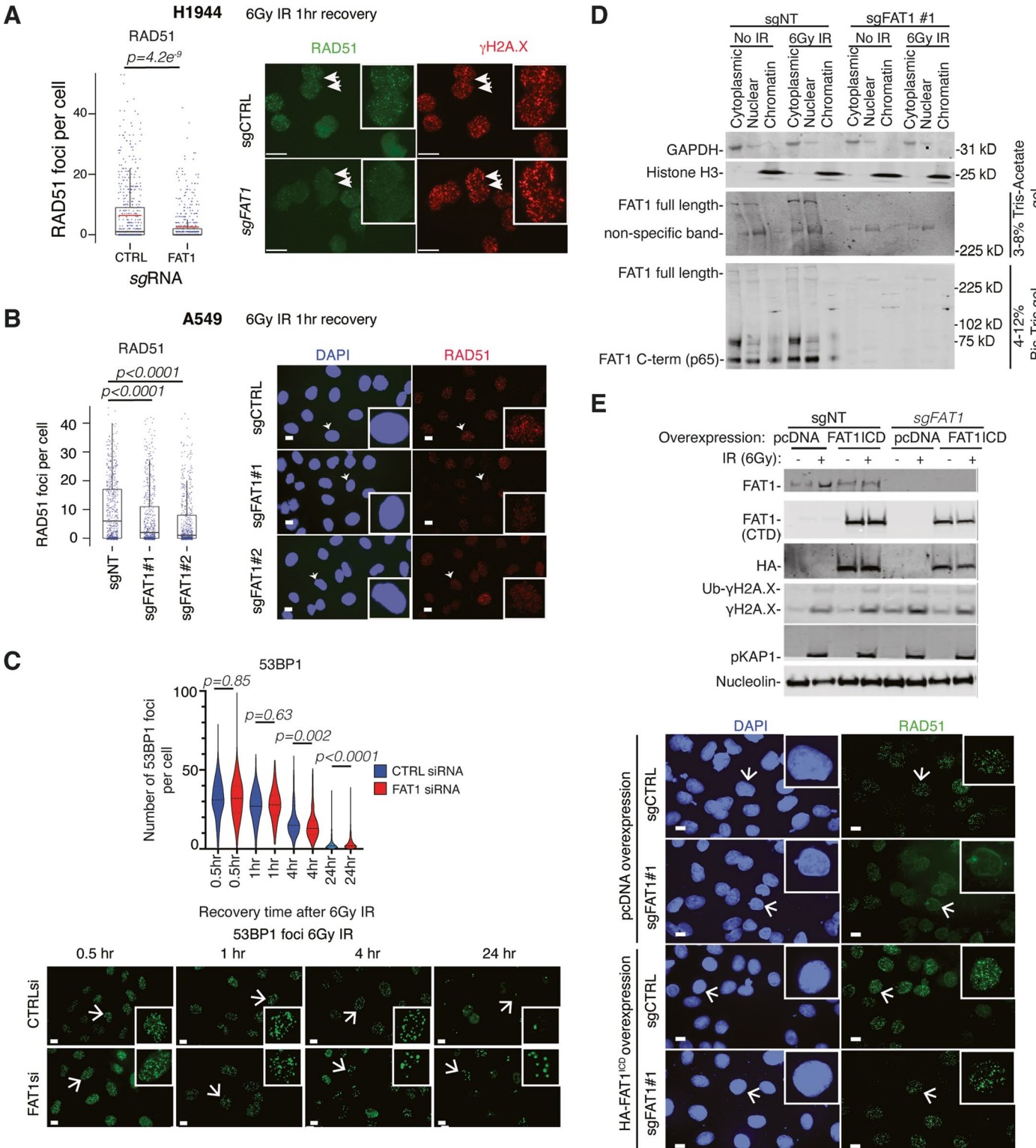

**Extended Data Fig. 4 | See next page for caption.**

**Extended Data Fig. 4 | *FAT1* CRISPR KO cells show reduced RAD51 foci formation post-ionizing. a**, Impact of *FAT1* on RAD51/γH2A.X foci formation in polyclonal *FAT1* wild-type or CRISPR knockout H1944 cells following 6Gy IR. Cells were irradiated and allowed 1 h to recover. Left, quantification showing RAD51 foci in the presence of 6Gy IR. Red-line = mean. The boxes represent the interquartile range, the lines represent the median and the whiskers range denote 1.5 x interquartile ranges, 2-sided Mann-Whitney test, n=3 biological replicates. Right, representative figures illustrating RAD51/γH2A.X foci following 6Gy IR. Bar = 10 μM. Polyclonal *FAT1* CRISPR knockout cells were irradiated and allowed 1h recovery. **b**, Left, representative figures illustrating RAD51 foci in the presence of 6Gy IR. Scale bar =10 μm. Right: box plots showing *FAT1* CRISPR knockout A549 cells displayed a reduction in RAD51 foci formation. The boxes represent the interquartile range, the lines represent the median and the whiskers range denotes 1.5 x interquartile ranges, Dunn's test, n=3 biological replicates. **c**, Quantification of 53BP1 foci at various time points post 6h IR irradiation

in A549 cells. n=2 biological repeats, over 300 cells were quantified in each condition per time point. Solid line = median, dotted lines = quartile ranges, 2-sided Mann-Whitney tests were carried out between two conditions within the same time point. **d**, Representative western blot showing cellular fractionation of control and *FAT1* CRISPR KO A549 cells. Localization of full-length FAT1 protein and C-terminal isoforms were visualized using a FAT1 antibody targeting the C-terminal of the FAT1 protein. Note the loss of FAT1 full-length and C-terminal (p65) signal in *FAT1* KO cells. **e**, A549 *FAT1* KO cells were transfected with the HA-FAT1$^{ICD}$ construct used in Fig. 3b,c. Top, representative western blot showing the level of endogenous FAT1 and HA-FAT1$^{ICD}$ construct in A549 cells following 6Gy IR. Levels of KAP1/TRIM28 phosphorylation, γH2A.X, and ubiquitination of γH2A.X are not affected. Bottom, representative figures illustrating RAD51 foci formation in the presence of 6Gy IR, with or without overexpression of HA-FAT1$^{ICD}$. Scale bar =10 μm. HD, homozygous deletions; LOH, loss of heterozygosity.

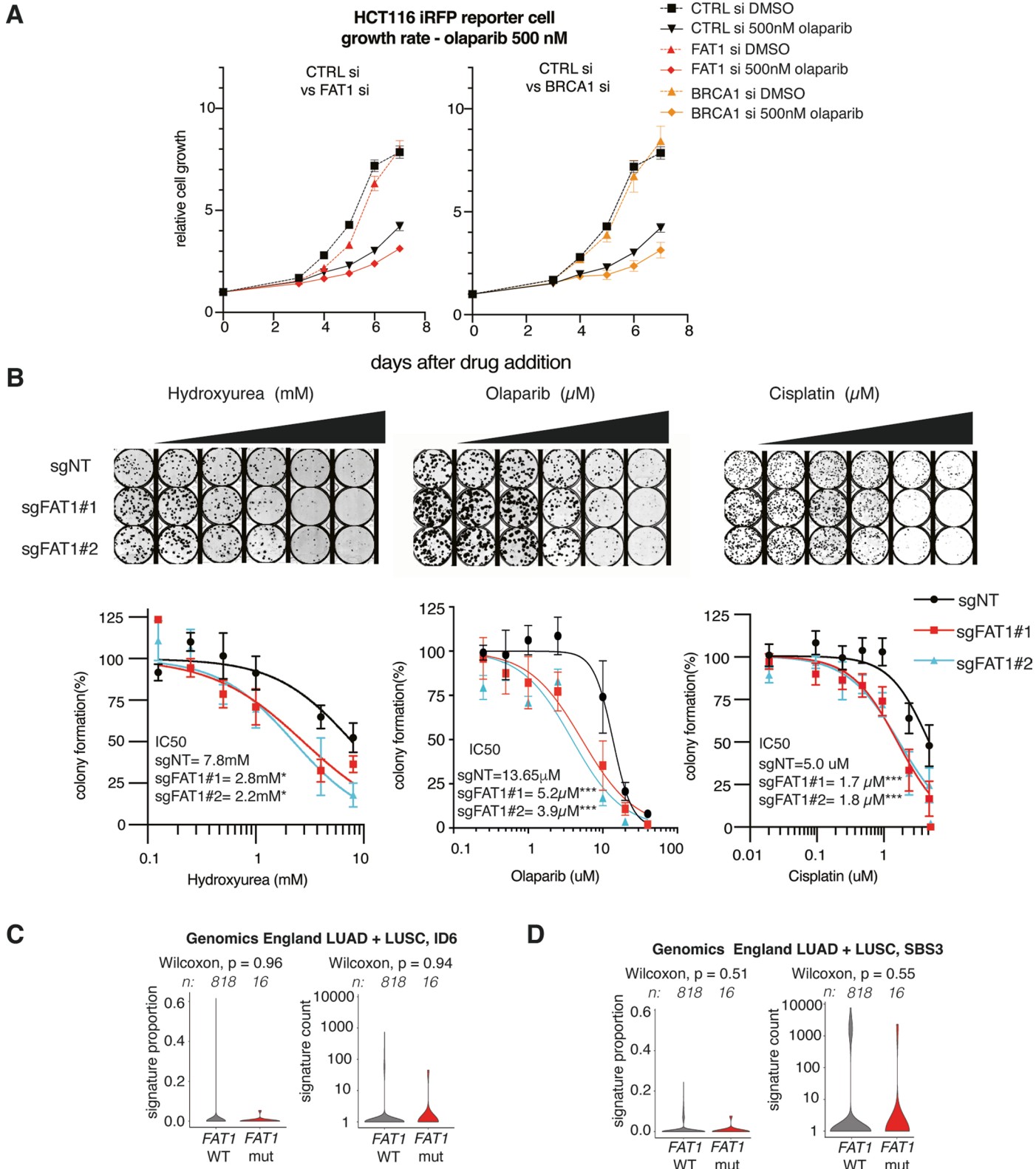

**Extended Data Fig. 5 | *FAT1* KO A549 cells display mild sensitivity towards hydroxyurea, cisplatin and Olaparib. a**, Comparison of growth rate trajectories of CTRL, *FAT1* and *BRCA1* knockdown in HCT116-iRFP cells[75] in the presence and absence of Olaparib. **b**, Representative images and quantification of clonogenic assays (left: hydroxyurea, replication stress inducer; middle: Olaparib, PARP inhibitor; right: cisplatin, DNA crosslinker). **c-d**, Genomics England mutational signature analysis showing the signature proportion (left) and signature count (right) of the mutational profile of COSMIC ID6 (**c**) and SBS3 (**d**) mutational signatures, both dependent on indels as a result of end-joining activity.

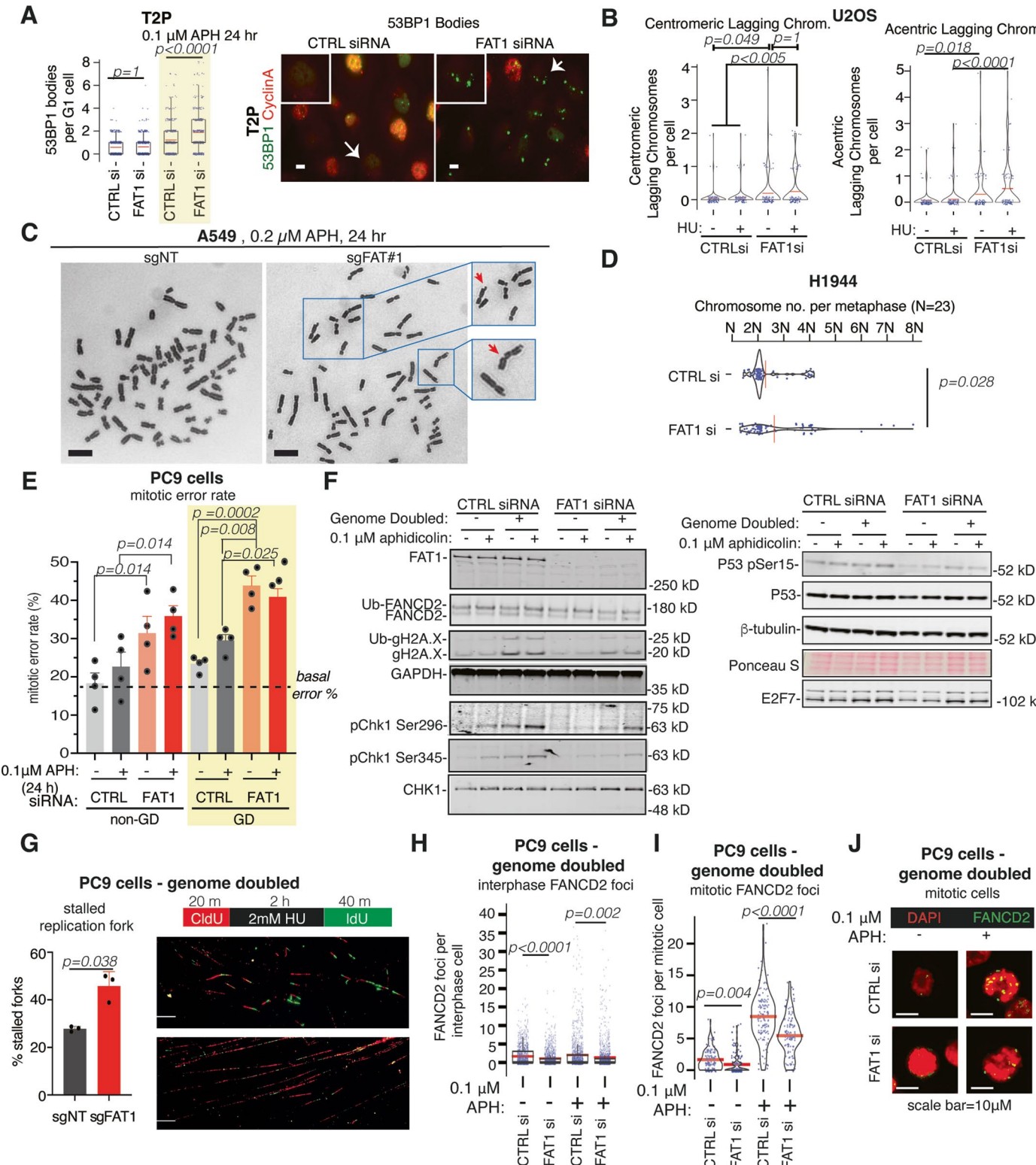

**Extended Data Fig. 6 | See next page for caption.**

**Extended Data Fig. 6 | *FAT1* loss results in chromosomal instability in lung cancer cells. a**, Transient *FAT1* siRNA knockdown induces the formation of 53BP1 nuclear bodies in T2P cells. The boxes represent the interquartile range, the lines represent the median and the whiskers range denotes 1.5 x interquartile ranges, Dunn's test, red bar = mean, n=3 biological replicates. Scale bar =10 μm. **b**, Transient *FAT1* siRNA significantly increases the number of lagging chromosomes per cell after replication stress (left, centromeric; right, acentric), Dunn's test. Over 100 mitotic cells were scored across 3 biological replicates. Red bar = mean. **c**, Representative metaphase spreads of *FAT1* WT and KO A549 cells after 24 hours of aphidicolin treatment. Scale bar = 5μM **d**, Metaphase chromosome number following transient *FAT1* knockdown in H1944 cells. A significant increase in metaphase chromosome number is observed in H1944 cells following *FAT1* knockdown. N=3 biological repeats, red bar = mean.

2-sided Welch's T-test. **e**, Mitotic error rate in PC9 cells following transient *FAT1* knockdown in PC9 cells. one-way ANOVA, N=4 biological repeats, mean±SEM. **f**, Western blot showing FAT1 knockdown efficiency in PC9 cells. FANCD2 monoubiquitylation and γH2A.X, or expression level of E2F7 are not significantly affected by *FAT1* ablation. **g**, *FAT1* knockdown leads to more replication fork stalling in genome-doubled PC9 cells. >600 forks were counted across 3 biological repeats, 2-sided Paired T-test, scale bar =20 μM. **h–j**, *FAT1* knockdown significantly reduces interphase FANCD2 foci (**h**, histogram) and mitotic FANCD2 foci (**i** and **j**, histogram and representative image, respectively) despite the increased rate of fork collapse. The boxes represent the interquartile range, black horizontal bar represent the median and the whiskers range denotes 1.5 x interquartile ranges, red line= mean, N=3 biological replicates, >1200 interphase and >120 mitotic cells scored per condition, Dunn's test.

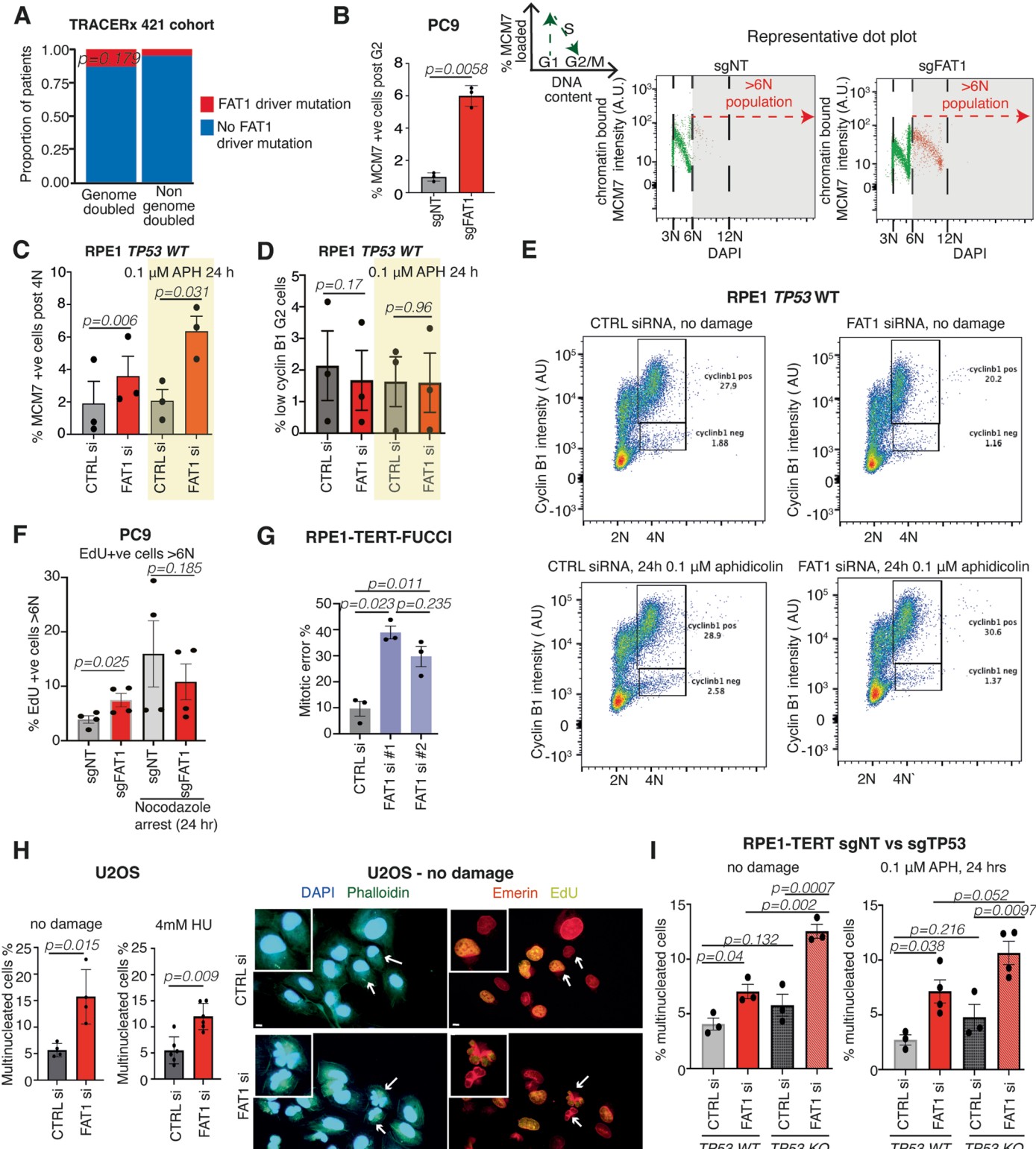

**Extended Data Fig. 7 | See next page for caption.**

**Extended Data Fig. 7 | *FAT1* loss promotes WGD and multinucleation.**
**a**, Proportion of *FAT1* driver mutations in genome-doubled TRACERx421 tumors. Fisher's Exact Test, p=0.179. **b**, *FAT1* CRISPR KO leads to elevated MCM7 reloading rate >6N in *TP53* mutant PC9 cells, 2-sided Paired T-test, mean±SD, n=3 biological replicates. **c**, *FAT1* ablation leads to elevated MCM7 reloading rate >4N in *TP53* WT RPE1 cells with transient *FAT1* siRNA transfection, mean±SEM, 2-sided paired T-test, n=3 biological repeat. **d,e**, Cyclin B1 levels in the G2/M population following CTRL or *FAT1* depletion in *TP53* WT RPE1 cells. The histogram in **d** shows no difference in G2/M cyclin B1 levels, mean±SEM, 2-sided paired T-test, n=3 biological repeats. The representative dot plots in **e** show cyclin B1 levels of the G2 population. **f**, Nocodazole treatment arrests PC9 cells at mitosis and causes endoreplication (EdU positive cells >6N). *FAT1* knockdown does not significantly further increase EdU-positive cells >6N when treated with nocodazole, mean±SEM, 2-sided paired T-Test, n=4 biological replicates.
**g**, Live cell tracking data showing mitotic error rate of RPE1-TERT-FUCCI cells. 2 different *FAT1* siRNAs produced a significant increase in mitotic error rate.

One-way ANOVA with Sidak correction, mean±SD, n=3 biological repeats.
**h**, Quantification of fixed microscopy images showing that *FAT1* siRNA knockdown increases the rate of multinucleated U2OS cells, with or without replication stress induced with hydroxyurea. Left, multinucleation was quantified using phalloidin as a marker for cell boundaries and nuclear envelope stain emerin was used to mark the number of nuclei per cell. Mean±SD, 2-sided paired T-test, Biological repeats: n=4 without damage, n=6 with HU. Right, representative image of cells, scale bar = 10 μM, white arrows = multinucleated cells. **i**, Quantification of fixed microscopy images showing *FAT1* knockdown elevates EdU incorporation rate in multinucleated *TP53* KO RPE1-TERT cells, but not in the *TP53* WT counterpart. Following aphidicolin-induced replication stress, *FAT1* depletion increases EdU incorporation rate in both *TP53* WT and *TP53* KO cells, mean±SD, one-way ANOVA with Holm-Sidak correction. Biological repeats, no damage n=3; Ctrl siRNA with aphidicolin n=3, *FAT1* siRNA with aphidicolin n=4.

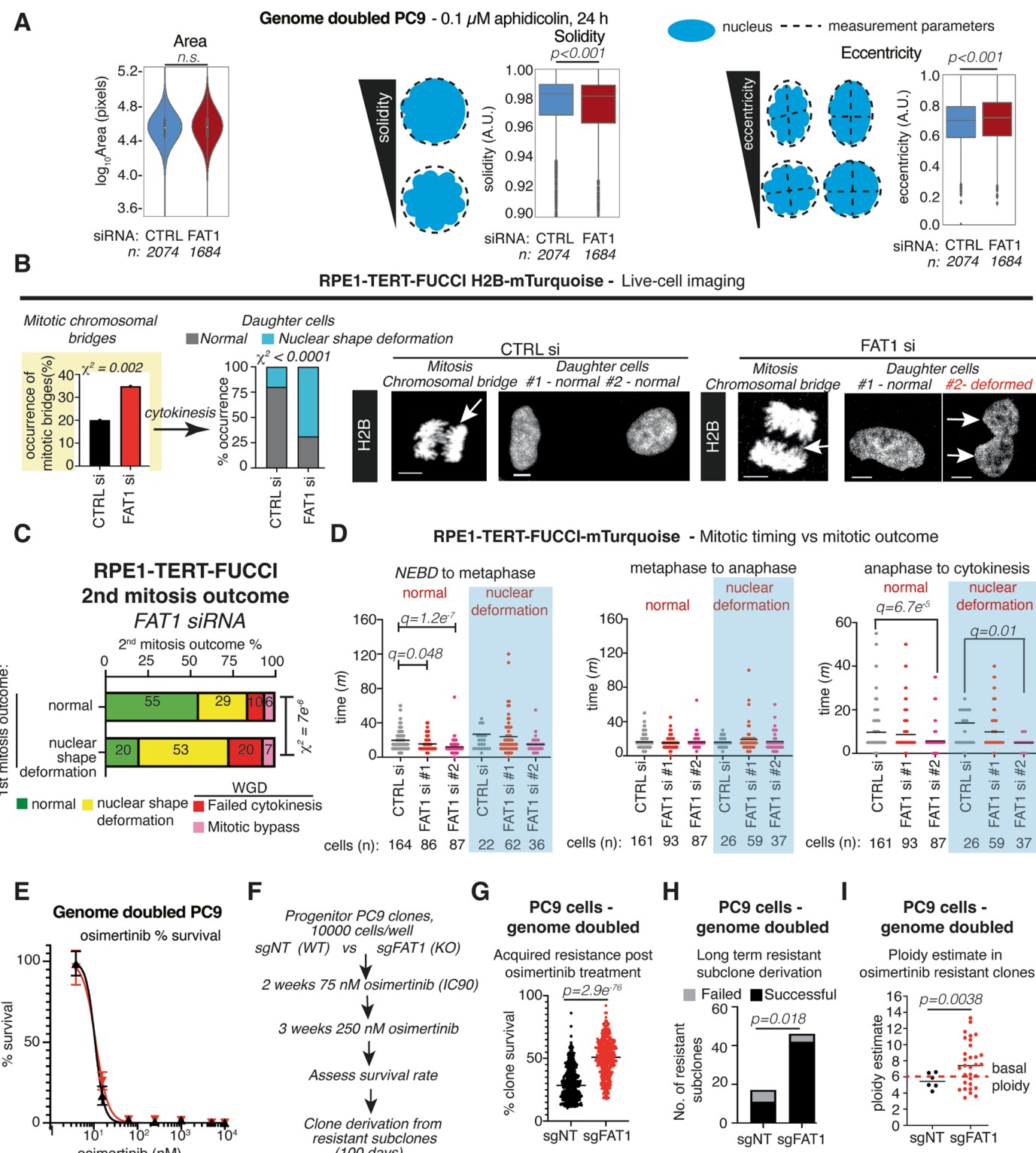

**Extended Data Fig. 8 | See next page for caption.**

**Extended Data Fig. 8 | *FAT1* loss results in nuclear shape deformation and exacerbates targeted therapy resistance. a**, Nuclear morphometric measurements of PC9 cells following CTRL and *FAT1* siRNA knockdown. Genome-doubled PC9 cells were treated for 24h with replication stress inducer aphidicolin. Z-project images were segmented and analysed for the difference in nuclear morphology. More than 1680 cells were analysed per condition over 3 biological repeats. 2-sided Mann-Whitney test, p<0.001. **b**, Live-cell imaging demonstrated higher chromatin bridge formation rate following transient *FAT1* siRNA knockdown in RPE1-TERT cells. In addition, the daughter cells are more likely to display nuclear shape deformation morphologies following chromatin bridge formation when *FAT1* is depleted. **c**, Mitotic outcomes of RPE1-TERT cells monitored after initial nuclear shape deformation. Among *FAT1* depleted cells, those cells with nuclear shape deformation were less likely to undergo normal mitotic segregation in the following mitosis. 60x magnification live cell microscopy performed using 5 minute intervals, 25 z-steps. scale bar = 5 µM, Chi-squared test. At least 50 mitotic events were tracked per condition over 8 biological repeats. **d**, Estimation of the effect of *FAT1* depletion on mitotic timing in RPE1-TERT-FUCCI cells. At least 100 cells were scored per siRNA knockdown across 3 biological repeats. Cells were imaged at 5-minute intervals over 12 h. Black bar = mean, Kruskal–Wallis test with False Discovery Rate Correction. **e**, IC90 determination of osimertinib treatment in PC9 cells. **f**, Experimental flowchart showing the generation of osimertinib-resistant PC9 subclones. **g,h**, Graphs demonstrating the impact of *FAT1* CRISPR KO on the acquisition of osimertinib resistance, as indicated by clone survival (**g**; 2-sided Mann-Whitney Test) and the number of long-term derivation of resistant subclones (**h**; Fisher's exact test). **i**, Among Osimertinib-resistant subclones, *FAT1* CRISPR KO cells exhibited significantly variable DNA ploidy compared with non-targeting controls. 2-sided Welch's T-test, non-targeting = 6 clones, *FAT1* KO = 31 clones.

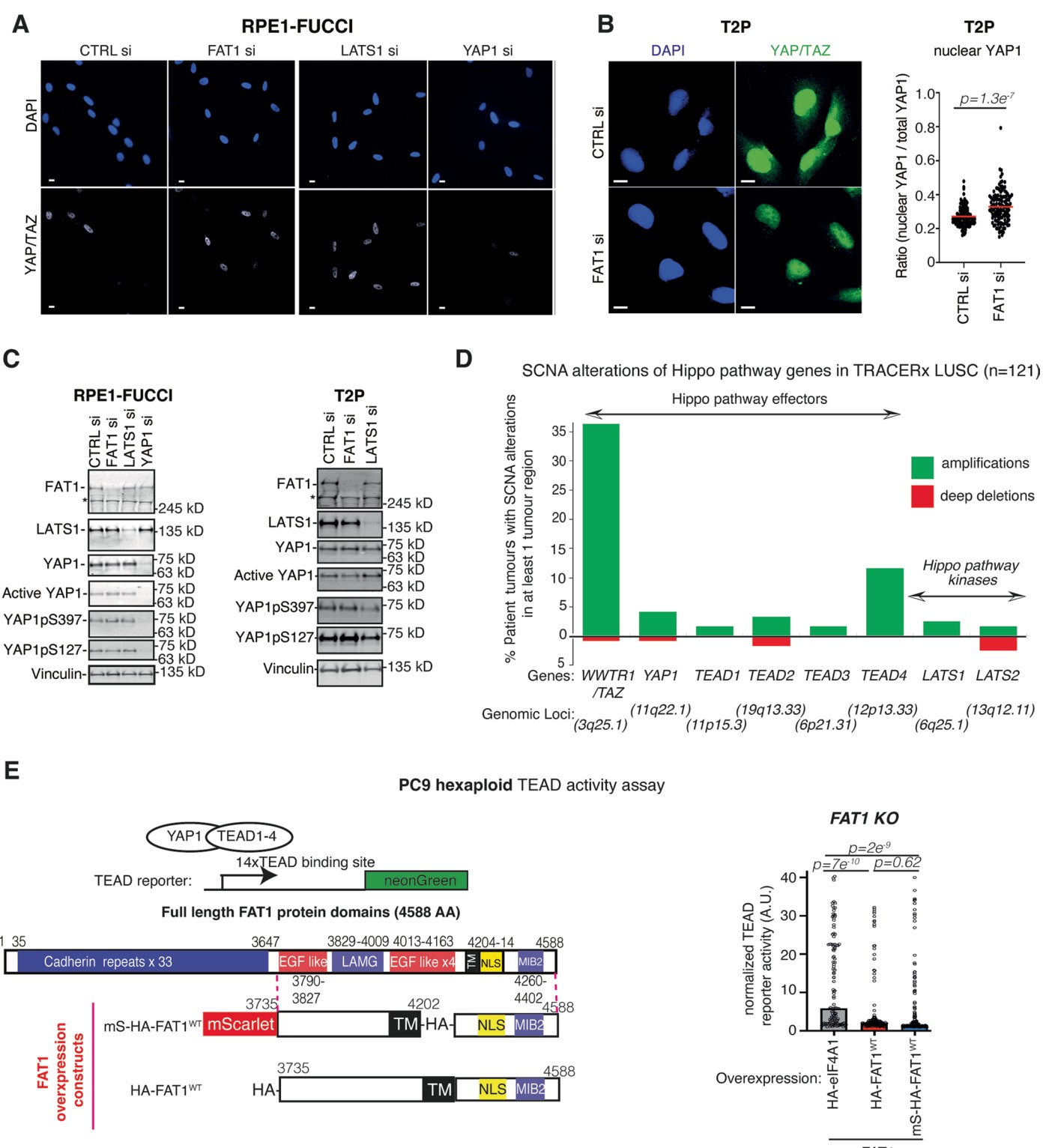

**Extended Data Fig. 9 | FAT1 loss results in dysregulation of YAP1 nuclear localization. a,b** Transient siRNA depletion of *FAT1* increases the relative YAP1 nuclear localization ratio in RPE1-FUCCI cells (**a**; PTEMF fixation) and T2P cells (**b** (left panel); PFA fixation). Scale bars =10 µm. Representative images are shown. The right panel in **b** shows quantification of the results in T2P cells. Red bar =mean, 2-sided Mann-Whitney test. **c**, Transient siRNA depletion of *FAT1* and *LATS1* does not consistently alter the phosphorylation level of YAP1 in RPE1 and T2P cells. **d**, Plot showing the incidences of amplification and deep deletions of Hippo pathway members in the TRACERx 421 LUSC cohort. The effectors

*TEAD4* and *WWTR1/TAZ* are amplified in >10% and >35% of cases, respectively. **e**, The localization of the HA-epitope tag does not affect the ability to repress TEAD reporter transcriptional activity. Left, Scheme showing the location of the HA-epitope tag in different FAT1 expression constructs. The HA-epitope tag was located either at the N terminus or internally within the FAT1 construct. Right, Histogram showing normalized TEAD transcriptional activity upon overexpression of different FAT1 construct. The edge of histograms denotes the mean values, One-way ANOVA with Holm-Sidak multiple comparisons test, for each condition >95 cells were scored across 3 biological repeats.

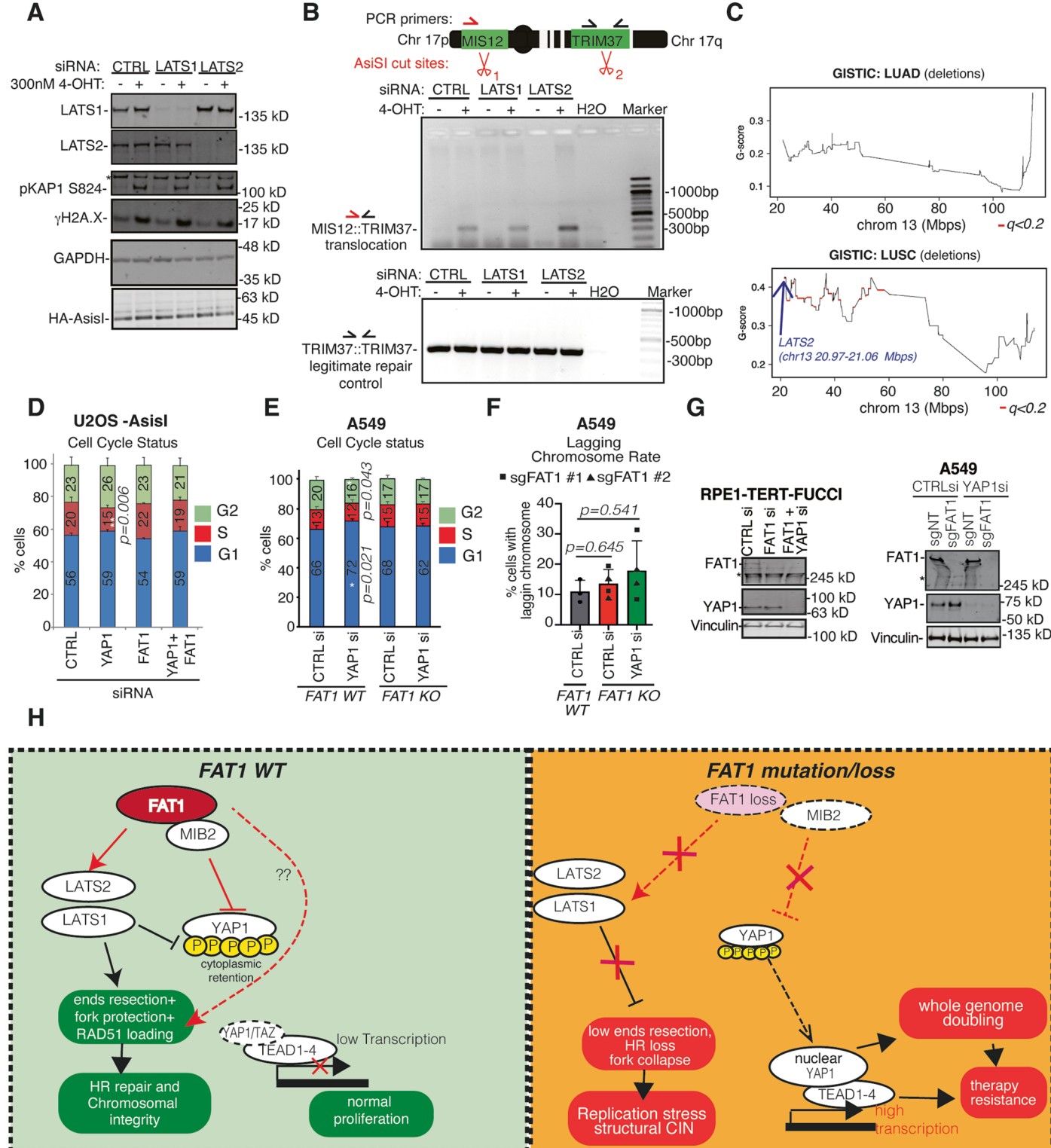

**Extended Data Fig. 10 | Loss of _LATS2_ is enriched in the TRACERx LUSC cohort. a**, Representative western blot following _LATS1/2_ knockdown in the U2OS-AsiSI DIvA system reveals that DNA damage markers such as γH2A.X and phospho-KAP1 are activated proficiently upon damage. **b**, Middle, representative gel showing that transient knockdown of _LATS2_, but not _LATS1_, induces an illegitimate repair product in U2OS-AsiSI DIvA cells. PCR products generated from the legitimate repair product were used as the loading control (bottom). n=3 biological repeats. **c**, Plots of GISTIC scores illustrating that the SCNA loss of the _LATS2_ genomic loci (_13q12.11_) is positively selected in the LUSC but not the LUAD TRACERx421 cohort. Red lines denote q value <0.2. **d,e**, Histograms

representing cell cycle profiles following loss of _FAT1_, _YAP1_ or combined loss of both _FAT1_ and _YAP1_ by siRNA in U2OS-AsiSI cells (**d**) and A549 _FAT1_ WT/KO cells (**e**), n=3 biological repeats, one-way ANOVA with Sidak corrections. **f**, Histogram quantifying the percentage of _FAT1_ WT or KO A549 cells with lagging chromosomes after aphidicolin treatment following transient depletion of _YAP1_ using RNAi, mean±SD, one-way ANOVA with Holm-Sidak multiple correction, biological repeats: _FAT1_ WT =3, _FAT1_ KO =4. **g**, Representative western blot showing _YAP1_ and _FAT1_ knockdown efficiency in the RPE1-FUCCI and A549 cells. **h**, Proposed model showing potential interaction between FAT1 and the Hippo pathway.

# Reporting Summary

## Statistics

For all statistical analyses, confirm that the following items are present in the figure legend, table legend, main text, or Methods section.

| n/a | Confirmed | |
|---|---|---|
| ☐ | ☒ | The exact sample size (*n*) for each experimental group/condition, given as a discrete number and unit of measurement |
| ☐ | ☒ | A statement on whether measurements were taken from distinct samples or whether the same sample was measured repeatedly |
| ☐ | ☒ | The statistical test(s) used AND whether they are one- or two-sided<br>*Only common tests should be described solely by name; describe more complex techniques in the Methods section.* |
| ☐ | ☒ | A description of all covariates tested |
| ☐ | ☒ | A description of any assumptions or corrections, such as tests of normality and adjustment for multiple comparisons |
| ☐ | ☒ | A full description of the statistical parameters including central tendency (e.g. means) or other basic estimates (e.g. regression coefficient) AND variation (e.g. standard deviation) or associated estimates of uncertainty (e.g. confidence intervals) |
| ☐ | ☒ | For null hypothesis testing, the test statistic (e.g. *F*, *t*, *r*) with confidence intervals, effect sizes, degrees of freedom and *P* value noted<br>*Give P values as exact values whenever suitable.* |
| ☒ | ☐ | For Bayesian analysis, information on the choice of priors and Markov chain Monte Carlo settings |
| ☒ | ☐ | For hierarchical and complex designs, identification of the appropriate level for tests and full reporting of outcomes |
| ☐ | ☒ | Estimates of effect sizes (e.g. Cohen's *d*, Pearson's *r*), indicating how they were calculated |

*Our web collection on statistics for biologists contains articles on many of the points above.*

## Software and code

Policy information about availability of computer code

| Data collection | LiCOR image studio v5.0<br>Oxford Optronix GelCount<br>MicroManager 2.0<br>Microvolution plugin for FIJI<br>FIJI ImageJ v 1.54J<br>BD FACSDiva software<br>ABI QuantStudio softwareNikon NIS elements<br>The code used to determine genome doubling is available at https://github.com/amf71/ParallelGDDetect. |
|---|---|
| Data analysis | R (version 3.6.2)<br>Alignment and QC:<br>FastQC (version 0.11.8)<br>FastQ Screen (version 0.13.0)<br>bwa-mem (version 0.7.17)<br>Sambamba (version 0.7.0)<br>Picard Tools (version 2.21.9)<br>GATK (version 3.8.1)<br>Somalier (version 0.2.7)<br>Samtools (version 1.9)<br>Conpair (version 0.2)<br>Variant Calling: |

SAMtools (version 1.10)
VarScan2 (version 2.4.4)
MuTect (version 1.1.7)
bam-readcount (version 0.7.4)
Annovar (version: Revision 529)
R packages used in version 3.6.3:
fst (version 0.9.4)
tidyverse (version 1.3.0)
survival (version 3.2.13)
ggplot2 (version 3.3.2)
dplyr (version 1.0.2)
tidyr (version 1.1.2)
gridExtra (version 2.3)
cowplot (version 1.1.0)
survminer (version 0.4.9)
ggpubr (version 0.4.0)
reshape2 (version 1.4.4)
tibble (version 3.0.4)
gtable (version 0.3.0)
RColorBrewer (version 1.1-2)
plyr (version 1.8.6)
ggrepel (version 0.8.2)
GenomicRanges (version 1.38.0)
rlist (version 0.4.6.2)
tidytext (version 0.2.3)
stringr (version 1.4.0)
data.table (version 1.13.2)
DiagrammR (version 1.0.1)
magrittr (version 2.0.1)
BSgenome.Hsapiens.UCSC.hg19 (version 1.4.0)
deconstructSigs (version 1.8.0)
Li-COR Image studio Lite (5.0)
Fiji (2.0.0)
FindFoci Plug-In (Herbert et al Plos ONE 2014)
FloJo (10.8.1)
Prism (9.4.1)

For manuscripts utilizing custom algorithms or software that are central to the research but not yet described in published literature, software must be made available to editors and reviewers. We strongly encourage code deposition in a community repository (e.g. GitHub). See the Nature Portfolio guidelines for submitting code & software for further information.

# Data

Policy information about availability of data

All manuscripts must include a data availability statement. This statement should provide the following information, where applicable:

- Accession codes, unique identifiers, or web links for publicly available datasets
- A description of any restrictions on data availability
- For clinical datasets or third party data, please ensure that the statement adheres to our policy

The RNA-seq, whole-exome sequencing and RRBS data (in each case from the TRACERx study) used during this study have been deposited at the European Genome–phenome Archive, which is hosted by the European Bioinformatics Institute and the Centre for Genomic Regulation, under the accession codes EGAS00001006517 (RNA-seq), EGAS00001006494 (WES) and EGAS00001006523 (RRBS). Access is controlled by the TRACERx data access committee. The Genomics England lung cohort is part of the 100,000 Genomes Project whose data are held in a secure research environment and are only available to registered users, for further information on how to obtain access visit https://www.genomicsengland.co.uk/research/academic.

# Research involving human participants, their data, or biological material

Policy information about studies with human participants or human data. See also policy information about sex, gender (identity/presentation), and sexual orientation and race, ethnicity and racism.

| Reporting on sex and gender | The TRACERx 421 cohort consisted of 233 males and 188 females (421 patients total), corresponding to a 55:45 M:F ratio. 93% of the cohort was from a white ethnic background and the mean age of the patients was 69, ranging between 34 and 92. Written informed consent was obtained. None of the patients were compensated for their involvement in the study. See Frankell et al Nature 2023 for cohort detail. |
|---|---|
| Reporting on race, ethnicity, or other socially relevant groupings | The ethnicity breakdown for the TRACERx 421 cohort is the following:<br>White- British 371 (88%)<br>White- Irish 17 (4%)<br>White - European 13 (3%)<br>White -Other 3 (1%)<br>Mixed 4 (1%)<br>Black 1 (<1%)<br>Caribbean 4 (1%) |

Indian 3 (1%)
Middle Eastern 4 (1%)
South American 1 (<1%)

| Population characteristics | Please note that the study started recruiting patients in 2016, when TNM version 7 was standard of care. The up-to-date inclusion/exclusion criteria now utilizes TNM version 8.

TRACERx inclusion and exclusion criteria

Inclusion Criteria:
_Written Informed consent
_Patients ≥18 years of age, with early stage I-IIIB disease (according to TNM 8th edition) who are eligible for primary surgery.
_Histopathologically confirmed NSCLC, or a strong suspicion of cancer on lung imaging necessitating surgery (e.g. diagnosis determined from frozen section in theatre)
_Primary surgery in keeping with NICE guidelines planned
_Agreement to be followed up at a TRACERx site
_Performance status 0 or 1
_Minimum tumor diameter at least 15mm to allow for sampling of at least two tumour regions (if 15mm, a high likelihood of nodal involvement on pre-operative imaging required to meet eligibility according to stage, i.e. T1N1-3)

Exclusion Criteria:
_Any other* malignancy diagnosed or relapsed at any time, which is currently being treated (including by hormonal therapy).
_Any other* current malignancy or malignancy diagnosed or relapsed within the past 3 years**.
*Exceptions are: non-melanomatous skin cancer, stage 0 melanoma in situ, and in situ cervical cancer
**An exception will be made for malignancies diagnosed or relapsed more than 2, but less than 3, years ago only if a pre-operative biopsy of the lung lesion has confirmed a diagnosis of NSCLC.
_Psychological condition that would preclude informed consent
_Treatment with neo-adjuvant therapy for current lung malignancy deemed necessary
_Post-surgery stage IV
_Known Human Immunodeficiency Virus (HIV), Hepatitis B Virus (HBV), Hepatitis C Virus (HCV) or syphilis infection.
_Sufficient tissue, i.e. a minimum of two tumor regions, is unlikely to be obtained for the study based on pre-operative imaging

Patient ineligibility following registration
_There is insufficient tissue
_The patient is unable to comply with protocol requirements
_There is a change in histology from NSCLC following surgery, or NSCLC is not confirmed during or after surgery.
_Change in staging to IIIC or IV following surgery
_The operative criteria are not met (e.g. incomplete resection with macroscopic residual tumors (R2)). Patients with microscopic residual tumors (R1) are eligible and should remain in the study
_Adjuvant therapy other than platinum-based chemotherapy and/or radiotherapy is administered. |
| Recruitment | TRACERx: Patients seen with a new diagnosis of lung cancer in lung cancer units across the United Kingdom, according to the eligibility criteria above, were recruited. No selection bias has been identified to date. |
| Ethics oversight | The TRACERx study was approved by the NRES Committee London with the following details:
Study title: TRAcking non small cell lung Cancer Evolution through therapy (Rx)
REC reference: 13/LO/1546
Protocol number: UCL/12/0279
IRAS project ID: 138871 |

Note that full information on the approval of the study protocol must also be provided in the manuscript.

# Field-specific reporting

Please select the one below that is the best fit for your research. If you are not sure, read the appropriate sections before making your selection.

☒ Life sciences ☐ Behavioural & social sciences ☐ Ecological, evolutionary & environmental sciences

For a reference copy of the document with all sections, see nature.com/documents/nr-reporting-summary-flat.pdf

# Life sciences study design

All studies must disclose on these points even when the disclosure is negative.

| Sample size | For the bioinformatics studies, the sample size of 421 patients represents the half-way point of the TRACERx longitudinal study. In total 432 tumours (1553 tumour regions) of the 421 patients were analysed in this study. TRACERx is a programme of work of multiple projects built around a single observational cohort study. It is not possible to perform a sample size calculation for each project, especially post hoc. Please see Frankell et al, Nature 2023 for detailed explanation. |

For the investigations involving cell lines, a minimum of three independent experiments were performed unless otherwise stated. Measurements were taken from distinct samples except from live cell imaging experiments when the same cell was recorded over a period of time. No statistical methods were used to predetermine sample size. The statistical test types and biological n number used in each experiment are detailed in the relevant figure legends.

Data exclusions | Please see study inclusion/exclusion criteria below. Additionally, samples which fail quality control metrics including low tumor purity (<10%) were also excluded from analysis.

Please note that the study started recruiting patients in 2016, when TNM version 7 was standard of care. The up-to-date inclusion/exclusion criteria now utilizes TNM version 8.

TRACERx inclusion and exclusion criteria

Inclusion Criteria:
_Written Informed consent
_Patients ≥18 years of age, with early stage I-IIIB disease (according to TNM 8th edition) who are eligible for primary surgery.
_Histopathologically confirmed NSCLC, or a strong suspicion of cancer on lung imaging necessitating surgery (e.g. diagnosis determined from frozen section in theatre)
_Primary surgery in keeping with NICE guidelines planned
_Agreement to be followed up at a TRACERx site
_Performance status 0 or 1
_Minimum tumor diameter at least 15mm to allow for sampling of at least two tumour regions (if 15mm, a high likelihood of nodal involvement on pre-operative imaging required to meet eligibility according to stage, i.e. T1N1-3)

Exclusion Criteria:
_Any other* malignancy diagnosed or relapsed at any time, which is currently being treated (including by hormonal therapy).
_Any other* current malignancy or malignancy diagnosed or relapsed within the past 3 years**.
*Exceptions are: non-melanomatous skin cancer, stage 0 melanoma in situ, and in situ cervical cancer
**An exception will be made for malignancies diagnosed or relapsed more than 2, but less than 3, years ago only if a pre-operative biopsy of the lung lesion has confirmed a diagnosis of NSCLC.
_Psychological condition that would preclude informed consent
_Treatment with neo-adjuvant therapy for current lung malignancy deemed necessary
_Post-surgery stage IV
_Known Human Immunodeficiency Virus (HIV), Hepatitis B Virus (HBV), Hepatitis C Virus (HCV) or syphilis infection.
_Sufficient tissue, i.e. a minimum of two tumor regions, is unlikely to be obtained for the study based on pre-operative imaging

Patient ineligibility following registration
_There is insufficient tissue
_The patient is unable to comply with protocol requirements
_There is a change in histology from NSCLC following surgery, or NSCLC is not confirmed during or after surgery.
_Change in staging to IIIC or IV following surgery
_The operative criteria are not met (e.g. incomplete resection with macroscopic residual tumors (R2)). Patients with microscopic residual tumors (R1) are eligible and should remain in the study
_Adjuvant therapy other than platinum-based chemotherapy and/or radiotherapy is administered.

Replication | TRACERx is a prospective longitudinal study. As such, the results shown are not the result of an experimental set up. This is the half-way point of the TRACERx study and reflects hypothesis-generating analysis.

For the investigations involving cell lines, a minimum of three independent experiments were performed unless otherwise stated. N number indicating biological replicates are available in the manuscripts. Measurements were taken from distinct samples except from live cell imaging experiments when the same cell was recorded over a period of time.

Randomization | No randomization was conducted. Randomization is not applicable for TRACERx data because this is an observational study. Randomization is not applicable for experiments involving cell lines.

Blinding | Not applicable for this study, except for fixed cell imaging studies. For experiments comprising imaging analysis, laser intensity and channel intensity were standardized using the control cells as a reference. Initially, microscope slides were renamed and blinded before microscopy imaging and analysis so that the sample identity is not known at the point of imaging. However, since FAT1 loss induces cell morphology change this is not strictly necessary as the identity become rather obvious during imaging. For genomics data analysis (TRACERx , TCGA or Genomics England) blinding is not applicable. For flow cytometry and WBs the experiments are not blinded.

# Reporting for specific materials, systems and methods

We require information from authors about some types of materials, experimental systems and methods used in many studies. Here, indicate whether each material, system or method listed is relevant to your study. If you are not sure if a list item applies to your research, read the appropriate section before selecting a response.

## Materials & experimental systems

| n/a | Involved in the study |
|---|---|
| ☐ | ☒ Antibodies |
| ☐ | ☒ Eukaryotic cell lines |
| ☒ | ☐ Palaeontology and archaeology |
| ☒ | ☐ Animals and other organisms |
| ☒ | ☐ Clinical data |
| ☒ | ☐ Dual use research of concern |
| ☒ | ☐ Plants |

## Methods

| n/a | Involved in the study |
|---|---|
| ☒ | ☐ ChIP-seq |
| ☐ | ☒ Flow cytometry |
| ☒ | ☐ MRI-based neuroimaging |

## Antibodies

| Antibodies used | The specificity of all the antibodies were validated by the supplier, with the relevant data page listed below.<br><br>Santa Cruz SC-56324 mouse anti mcm7 used in facs<br>https://www.scbt.com/p/mcm7-antibody-47dc141-human<br>abcam ab32053 rabbit anti cyclin b1 used in facs<br>https://www.abcam.com/en-gb/products/primary-antibodies/cyclin-b1-antibody-y106-ab32053<br>Thermo Fisher (clickIT EdU kit) c10634 anti edu used in facs and IF(against nucleotides, not protein)<br>https://www.thermofisher.com/order/catalog/product/C10634?SID=srch-hj-c10634<br>Abcam 6326 rat anti Anti-BrdU for CIdU used in fibre assay (against nucleotides, not protein, widely used in multiple DDR papers)<br>https://www.abcam.com/en-gb/products/primary-antibodies/brdu-antibody-bu1-75-icr1-proliferation-marker-ab6326<br>BD (clone B44) 347580 mouse anti Anti-BrdU for IDU used in fibre assay( against nucleotides, not protein, widely used in multiple DDR papers)<br>https://www.sigmaaldrich.com/GB/en/product/sigma/b2531?<br>utm_source=google&utm_medium=cpc&utm_campaign=21473730186&utm_content=165772576758&gclid=CjwKCAjwoJa2BhBPEi<br>wA0l0ImFXQUeI5sudLYDRU70jmLQAgn_INvq-U6YK6kxL5jDO2_5G-d8rIoBoCE90QAvD_BwE<br>MerckMillipore 05-636 mouse anti γH2A.X used in high content screening and IF<br>https://www.merckmillipore.com/GB/en/product/Anti-phospho-Histone-H2A.X-Ser139-Antibody-clone-JBW301,MM_NF-05-636?<br>ReferrerURL=https%3A%2F%2Fwww.google.com%2F<br>Santa Cruz sc-22760 rabbit anti 53bp1  used in high content screening – discontinued due to Santa Cruz losing animal license.<br>https://www.scbt.com/p/53bp1-antibody-h-300<br>Santa Cruz sc-8349 rabbit anti rad51 used in high content screening– discontinued due to Santa Cruz losing animal license.<br>https://www.scbt.com/p/rad51-antibody-h-92<br>Abcam ab87277 rabbit anti rpaps4s8 used in high content screening<br>https://www.abcam.com/en-gb/products/primary-antibodies/rpa32-rpa2-phospho-s4-s8-antibody-ab87277<br>Abcam ab109394 rabbit anti rpa pT21 used in high content screening<br>https://www.abcam.com/en-gb/products/primary-antibodies/rpa32-rpa2-phospho-t21-antibody-epr28462-ab109394<br>Abcam ab63801 rabbit anti rad51 used in IF<br>https://www.abcam.com/en-gb/products/primary-antibodies/rad51-antibody-ab63801<br>Cell Signaling 4526 mouse anti patmS1981 used in IF<br>https://www.cellsignal.com/products/primary-antibodies/phospho-atm-ser1981-10h11-e12-mouse-mab/4526<br>Novus Biologicals nb100-305 rabbit anti 53BP1 used in IF<br>https://www.bio-techne.com/p/antibodies/53bp1-antibody_nb100-305<br>Abcam ab16780 mouse anti BRCA1 used in IF<br>https://www.abcam.com/en-gb/products/primary-antibodies/brca1-antibody-ms110-ab16780<br>Francis Crick institute (in house monoclonal) E43.2 mouse anti cyclin A used in IF<br>NB: This is the same clone sold by Santa Cruz Sc-53229, and was used in multiple papers.<br>https://www.scbt.com/p/cyclin-a-antibody-e43-2?srsltid=AfmBOoqon6i6jwR5N5L9n1EpI04yAxIx6OM_Udav79LNDDhAwsEU56Te<br>Cell Signaling 9201 rabbit anti CtIP used in IF<br>https://www.cellsignal.com/products/primary-antibodies/ctip-d76f7-rabbit-mab/9201<br>ImmunoVision HCT-0100 human anti CREST used in IF<br>https://immunovision.com/index.php/autoimmune-polyclonal-antibodies/autoimmune-positive-controls/<br>Cell Signaling 58982 rabbit anti CENPF used in IF<br>https://www.cellsignal.com/products/primary-antibodies/cenp-f-d6x4l-rabbit-mab/58982<br>Cell Signaling 8878  anti Alexa Fluor 488 phalloidin used in if<br>https://www.cellsignal.com/products/buffers-dyes/alexa-fluor-488-phalloidin/8878<br>Cell Signaling 8940  anti Alexa Fluor 647 phalloidin used in IF<br>https://www.cellsignal.com/products/buffers-dyes/alexa-fluor-647-phalloidin/8940<br>Atlas antibodies AMAb90562 mouse anti emerin used in if<br>https://www.atlasantibodies.com/products/primary-antibodies/precisa-monoclonals/anti-emd-antibody-amab90562-100ul/?<br>language=en<br>Santa Cruz SC-101199 mouse anti Yap1 used in if and wb<br>https://www.scbt.com/p/yap-antibody-63-7<br>Novus Biological  NB100-182 rabbit anti FancD2 used in if and wb<br>https://www.bio-techne.com/p/antibodies/fancd2-antibody_nb100-182<br>Cell Signaling 65344 rabbit anti Ubr5 used in wb<br>https://www.cellsignal.com/products/primary-antibodies/ubr5-d6o8z-rabbit-mab/65344<br>Novus Biologicals NBP2-32275 rabbit anti Fat1 used in wb<br>https://www.bio-techne.com/p/antibodies/fat1-antibody_nbp2-32275#reviews |
|---|---|

Proteintech 27071-1-AP rabbit anti Rad21 used in wb
https://www.ptglab.com/products/RAD21-Antibody-27071-1-AP.htm
Cell Signaling 9718 rabbit anti γH2A.X used in wb
https://www.cellsignal.com/products/primary-antibodies/phospho-histone-h2a-x-ser139-20e3-rabbit-mab/9718
Proteintech 10398-1-AP rabbit anti Bap1 used in wb
https://www.ptglab.com/products/BAP1-Antibody-10398-1-AP.htm
Bethyl A300-363A-T rabbit anti Crebbp used in wb
NB: Bethyl Laboratories was rebranded as Fortis Life Science
https://www.fortislife.com/products/primary-antibodies/rabbit-anti-cbp-antibody/BETHYL-A300-363
Francis Crick insitute (in house monoclonal) clone 12AC5 mouse anti ha used in wb
*NB: In house monoclonal antibody. It is however the same clone as
https://www.sigmaaldrich.com/GB/en/product/roche/roaha?srsltid=AfmBOor8dii5mITqC6Z-
h7XNiOfy_COduOp1bRgZnUHPEz8eKL2Dbgcg
Proteintech 25241-1-AP rabbit anti ncoa6 used in wb
https://www.ptglab.com/products/NCOA6-Antibody-25241-1-AP.htm
Abcam ab1791 rabbit anti Histone H3 used in wb
https://www.abcam.com/en-gb/products/primary-antibodies/histone-h3-antibody-nuclear-marker-and-chip-grade-ab1791
Abcam ab9485 rabbit anti gapdh used in wb
https://www.abcam.com/en-gb/products/primary-antibodies/gapdh-antibody-loading-control-ab9485
Bethyl A300-767A rabbit anti pkap1 used in wb
NB: Bethyl Laboratories was rebranded as Fortis Life Science
https://www.fortislife.com/products/primary-antibodies/rabbit-anti-phospho-kap-1-s824-antibody/BETHYL-A300-767
Abcam ab22758 rabbit anti nucleolin used in wb
https://www.abcam.com/en-gb/products/primary-antibodies/nucleolin-antibody-ab22758
Cell Signaling 2661 rabbit anti pChk2 Thr68 used in wb
https://www.cellsignal.com/products/primary-antibodies/phospho-chk2-thr68-antibody/2661
Cell Signaling 3477 rabbit anti Lats1 used in wb
https://www.cellsignal.com/products/primary-antibodies/lats1-c66b5-rabbit-mab/3477
Cell Signaling 5888 rabbit anti Lats2 used in wb
https://www.cellsignal.com/products/primary-antibodies/lats2-d83d6-rabbit-mab/5888
Cell Signaling 2349 rabbit anti pchk1 s296 used in wb
https://www.cellsignal.com/products/primary-antibodies/phospho-chk1-ser296-antibody/2349
Cell Signaling 2348 rabbit anti pchk1 s345 used in wb
https://www.cellsignal.com/products/primary-antibodies/phospho-chk1-ser345-133d3-rabbit-mab/2348
Cell Signaling 2360 mouse anti chk1 used in wb
https://www.cellsignal.com/products/primary-antibodies/chk1-2g1d5-mouse-mab/2360
Cell Signaling 9286 mouse anti p53s15 used in wb
https://www.cellsignal.com/products/primary-antibodies/phospho-p53-ser15-16g8-mouse-mab/9286
Cell Signaling 2527 rabbit anti p53 used in wb
https://www.cellsignal.com/products/primary-antibodies/p53-7f5-rabbit-mab/2527
Abcam ab6046 rabbit anti beta tubulin used in wb
https://www.abcam.com/en-gb/products/primary-antibodies/beta-tubulin-antibody-loading-control-ab6046
Affinity DF2444 rabbit anti e2f7 used in wb
https://www.affbiotech.com/pdf?id=6525
Cell Signaling 29495 rabbit anti active yap1 used in wb
https://www.cellsignal.com/products/primary-antibodies/non-phospho-active-yap-ser127-e6u8z-rabbit-mab/29495
Cell Signaling 13619 rabbit anti yap1 ps397 used in wb
https://www.cellsignal.com/products/primary-antibodies/phospho-yap-ser397-d1e7y-rabbit-mab/13619
Cell Signaling 4911 rabbit anti yap1 ps127 used in wb
https://www.cellsignal.com/products/primary-antibodies/phospho-yap-ser127-antibody/4911
Santa Cruz sc-73614 mouse anti vinculin used in wb
https://www.scbt.com/p/vinculin-antibody-7f9
Cell Signaling 6943 rabbit anti Src pTyr416 used in wb
https://www.cellsignal.com/products/primary-antibodies/phospho-src-family-tyr416-d49g4-rabbit-mab/6943
Cell Signaling 65890 rabbit anti yes used in wb
https://www.cellsignal.com/products/primary-antibodies/yes-d9p3e-rabbit-mab/65890
Cell Signaling 9154 rabbit anti phospho mek1/2 Ser217/221 used in wb
https://www.cellsignal.com/products/primary-antibodies/phospho-mek1-2-ser217-221-41g9-rabbit-mab/9154
Cell Signaling 4376 rabbit anti pErk1/2 T202/Y204 used in wb
https://www.cellsignal.com/products/primary-antibodies/phospho-p44-42-mapk-erk1-2-thr202-tyr204-20g11-rabbit-mab/4376
Cell Signaling 4695 rabbit anti total erk1/2 used in wb
https://www.cellsignal.com/products/primary-antibodies/p44-42-mapk-erk1-2-137f5-rabbit-mab/4695

| Validation | The antibodies used have been validated accordingly to manufacturer's instructions. Molecular size markers have been included on each of the western blots, and the molecular weights for each of the antibodies were validated as per manufacturer's datasheets detailed above. |
| --- | --- |

# Eukaryotic cell lines

Policy information about cell lines and Sex and Gender in Research

| Cell line source(s) | H1944 (lung, female), H1650 (lung, male), H1792 (lung, male), HCC4006(lung, male), A549 (CCL-185, lung male) and U2OS (HTB-96, osteosarcomam female) cells were obtained from Cell Services at the Francis Crick Institute, UK. |
| --- | --- |

RPE1-FUCCI-H2B-Turquoise cells have been described previously and were a kind gift from John Diffley (The Francis Crick Institute, UK). U2OS-HA-ER-AsiSI cells have been described previously 18 and were a kind gift from Gaelle Legube (Univ. Paul Sabatier, Toulouse, France). T2P-ER-KRAS-V12 cells (referred to as T2P) have been described previously and were a kind gift from Julian Downward (The Francis Crick Institute, UK). RPE1 TP53WT and RPE1 TP53 KO cells have been described previously. HCT116 (male, colon) iRFP cell lines have been described previously and were a kind gift from Karen Vousden (The Francis Crick Institute, UK).

**Authentication**

For Cell Authentication we use STR (Short Tandem Repeat) Profiling for all our Human cell lines using the Promega PowerPlex16HS system. This profile is compared back to any available on commercial cell banks (such as ATCC). We confirm the species is correct using a primer system based on the Cytochrome C Oxidase Subunit 1 gene from mitochondria – we call this test Species ID. Authentication is carried out in house within the Francis Crick Institute.

**Mycoplasma contamination**

For Mycoplasma screening we primarily use two different tests – Agar Culture (which involves culturing any mycoplasma that may be present in the cell culture on specialised agar) and Fluorescent staining using the Hoescht Stain. A third detection method, the PCR mycoplasma test (ATCC), is used on occasion when a rapid result is required.Mycoplasma test is carried out in house routinely within the Francis Crick Institute. No  mycoplasma contamination was detected.

**Commonly misidentified lines**
(See ICLAC register)

No Commonly misidentified lines were used in this study.

# Flow Cytometry

## Plots

Confirm that:

☒ The axis labels state the marker and fluorochrome used (e.g. CD4-FITC).

☒ The axis scales are clearly visible. Include numbers along axes only for bottom left plot of group (a 'group' is an analysis of identical markers).

☒ All plots are contour plots with outliers or pseudocolor plots.

☒ A numerical value for number of cells or percentage (with statistics) is provided.

## Methodology

**Sample preparation**

For HAC assay, HT1080 cells containing EGFP-HAC were maintained in 6 µg/ml blasticidin S selection media. For siRNA treatment, 100000  cells/well were seeded in 6 well plates before the day of the experiment. Cells were transfected with each siRNAs using the Lulluby reagent. Cells were grown without blasticidin S selection for 14 days. Silencing efficiency was monitored by Western blot analysis. On Day 14, cells were collected and analyzed by flow cytometry to determine the proportion of cells that gained or lost EGFP fluorescence. All experiments were carried out in triplicate.

For ImageStream FISH experiments, transfected cells were processed for FISH in suspension as described in Worrall et al, Cell Reports 2018. Briefly, following transfection, cells were harvested and fixed with freshly prepared 3:1 methanol-glacial acetic acid. Cells were subsequently hybridized with chromosome 15 satellite enumeration probe (LPE015G, Cytocell) performed in a thermocycler under the following conditions: 65°C (2 hr pre annealing), 80°C (5 min), 37°C (16 hr), prior to analysis on an ImageStream X Mk II (Amnis).

For the DR-GFP assay, samples were prepared as detailed in Seluanov et al JoVE 2010. For ploidy determination experiments, cells were trypsinized, washed in PBS and then fixed in 70% ethanol. Cells were then washed again, and treated with staining buffer containing propidium iodide and 100 µg/ml RNAse A.

For MCM7 loading assay, PC9 cells or RPE1 cells were trypsinized and treated with CSK buffer (10 mM HEPES pH7.9, 100 mM NaCl, 3 mM MgCl2, 1 mM EGTA, 300 mM sucrose, 1%BSA, 0.2% Triton X-100, 1 mM DTT) on ice for 5 minutes to remove soluble proteins. Cells were washed with 1%BSA in PBS and fixed in 4% PFA (Thermo Fisher Scientific for 10 minutes). Cells were then washed, pelleted and permeabilised in 70% EtOH for 15 minutes. MCM7 staining was performed using a mouse anti-MCM7 antibody (Santa Cruz, sc-56324) followed by an Alexa 594 goat-anti-mouse antibody (A11007, Invitrogen). DNA content was stained using 1 ug/ml DAPI supplemented with 100 ug/ml RNAse A.  For the EdU incorporation assay and cyclin B loading assays, the Click-iT Plus EdU Alexa Fluor 647 Flow Cytometry Assay kit (Thermo Fisher) was used for EdU incorporation assays.  Cells were incubated with 10 uM EdU for 30 minutes before being fixed and processed as per the manufacturer's instructions. Cyclin B1 antibody was obtained from Abcam (ab32053, Abcam). DNA content was stained in staining buffer (1 µg/ml DAPI, 100 µg/ml RNAse A in PBS).

Cell doublets were excluded from all analyses.

**Instrument**

Flow cytometry analyses were carried out on a BD Fortessa. All cell-sorting experiments were carried out on a BD  Aria Fusion or Aria III. Image Stream MkII flow cytometer (Merck) is used for flow-FISH experiments.

**Software**

FlowJo10.4.2 was used for flow analysis

**Cell population abundance**

Not applicable. Cell lines were used for flow analysis. Gating strategy described as below.

Gating strategy

All samples were first gated to exclude cellular debris using SSC-A/FSC-A and a diagonal gate using FSC-H/FSC-A was then used to exclude doublets. Whenever applicable, a diagonal gate using nucleic acid stains (PE-H/PE-A) was further used to refine doublet gating. Fluorescence minus one controls were used to set up EdU incorporation experiments, DNA ploidy analysis, MCM7 loading assays and cyclin B loading experiments. For DR-GFP HR reporter assays, No-ISceI control and mCherry co-transfection control were used to refine the gating strategy.
Gating strategy examples will be included in the Supp. information in the revised manuscript.

☒ Tick this box to confirm that a figure exemplifying the gating strategy is provided in the Supplementary Information.

