## [Peer Review File · Nature Cell Biology]

TRACERx analysis identifies a role for FAT1 in regulating chromosomal instability and whole-genome doubling via Hippo signaling

Corresponding Author: Professor Charles Swanton

Version 0:

Decision Letter:

*Please delete the link to your author homepage if you wish to forward this email to co-authors.

Dear Professor Swanton,

Please first accept our apology for the delay getting back to you with a decision due to difficulties in retrieving reviewers' comments.

Your manuscript, "Functional characterisation of driver events from TRACERx reveals mechanisms of NSCLC evolution", has now been seen by 2 referees, who are experts in CIN and cancer (referee 1 and 3). As you will see from their comments (attached below) they find this work of potential interest, but have raised substantial concerns, which in our view would need to be addressed with considerable revisions before we can consider publication in Nature Cell Biology.

We would like to clarify that although we have engaged a third referee (Reviewer 2) with expertise on lung cancer on the referee panel, the substantial concerns raised by the other two referees were sufficient for us to form a decision in the absence of this expert's feedback, and we felt a further delay would be counterproductive for the authors. We will send you the third report if/when we receive it.

Nature Cell Biology editors discuss the referee reports in detail within the editorial team, including the chief editor, to identify key referee points that should be addressed with priority, and requests that are overruled as being beyond the scope of the current study. To guide the scope of the revisions, I have listed these points below. We are committed to providing a fair and constructive peer-review process, so please feel free to contact me if you would like to discuss any of the referee comments further.

In particular, it would be essential to:

A) Strengthen the proposed mechanism as suggested by Reviewer 1;

B) Clarify the significance of the study given the weak phenotype as questioned by Reviewer 3:

"However, most described phenotypes are rather weak, which is not a critique per se, but this might make the results less suitable for the broad readership NCB..."

C) Provide the rationale with respect of choice and usage of cell lines and FAT1 perturbation as questioned by Reviewer 3:

"An important point of critique that I do need to make is that the authors use various cell lines throughout the paper for different types of experiments. In some cases, this is well-justified, in other cases less so. To ensure physiological relevance of all findings, there should be overlap between the cell lines being used for the different assays and per assay type 2-3 cell lines should be tested at least. Also, the choice of 'general' cell lines should be well-justified with 2 cell lines at least being used for all experiments."

"Also, different methods to alleviate FAT1 are being used, sometimes siRNA, sometimes CRISPR KO. It is not completely clear why this is, and while it is not necessary to redo all experiments for both siRNA and CRISPR knockout, it would strengthen the data if key experiments of the paper are done with both siRNA and CRISPR KO approaches including all necessary replicates (multiple knockout lines per gene, i.e independently engineered KO cell lines, ideally from different gRNAs and/or siRNAs)."

D) All other referee concerns pertaining to strengthening existing data, providing controls, methodological details, clarifications and textual changes as applicable should also be addressed.

E) Finally please pay close attention to our guidelines on statistical and methodological reporting (listed below) as failure to do so may delay the reconsideration of the revised manuscript. In particular please provide:

We would be happy to consider a revised manuscript that would satisfactorily address these points, unless a similar paper is published elsewhere, or is accepted for publication in Nature Cell Biology in the meantime.

- ensure that it conforms to our format instructions and publication policies (see below and <https://www.nature.com/nature/for-authors>).

- provide a point-by-point rebuttal to the full referee reports verbatim, as provided at the end of this letter.

- provide the completed Reporting Summary (found here <https://www.nature.com/documents/nr-reporting-summary.pdf>). This is essential for reconsideration of the manuscript will be available to editors and referees in the event of peer review. For more information see <http://www.nature.com/authors/policies/availability.html> or contact me.

Nature Cell Biology is committed to improving transparency in authorship. As part of our efforts in this direction, we are now requesting that all authors identified as 'corresponding author' on published papers create and link their Open Researcher and Contributor Identifier (ORCID) with their account on the Manuscript Tracking System (MTS), prior to acceptance. ORCID helps the scientific community achieve unambiguous attribution of all scholarly contributions. You can create and link your ORCID from the home page of the MTS by clicking on 'Modify my Springer Nature account'. For more information please visit www.springernature.com/orcid.

This journal strongly supports public availability of data. Please place the data used in your paper into a public data repository, or alternatively, present the data as Supplementary Information. If data can only be shared on request, please explain why in your Data Availability Statement, and also in the correspondence with your editor. Please note that for some data types, deposition in a public repository is mandatory - more information on our data deposition policies and available repositories appears below.

Link Redacted

We would like to receive a revised submission within six months.

We hope that you will find our referees' comments, and editorial guidance helpful. Please do not hesitate to contact me if there is anything you would like to discuss.

Best wishes,

Zhe Wang

Zhe Wang, PhD
Senior Editor
Nature Cell Biology

Tel: +44 (0) 207 843 4924
email: zhe.wang@nature.com

Reviewers' Comments:

Reviewer #1:

Remarks to the Author:

In this manuscript, Lu et al. build on their observation in human lung cancer samples to identify and validate a novel role for FAT1 as a driver of genomic instability. They first make the observation that FAT1 loss induces homologous recombination deficiency (HRD) then show that it also coincides with increased levels of chromosomal instability (CIN) manifested by elevated rates of chromosome missegregation and micronuclei formation. They go on to show that FAT1 depleted cells undergo increased rates of whole genome doubling (WGD) derived from cytokinesis failure and nuclear deformity. As a result of increased CIN and WGD they demonstrate that FAT1-depleted cells are more likely to exhibit treatment resistance. Finally, the authors show that FAT1 loss alters the HIPPO pathway in cancer cells and demonstrate some interesting rescue experiments of some of the phenotypes by co-depletion of FAT1 and YAP1 a key member of this pathway.

Overall, this is an impressive body of work that identifies a poorly characterized molecular driver of lung cancer evolution and an important mechanism of CIN in human cancer. The authors have covered significant ground in phenomenologically characterizing the cellular consequences of FAT1 loss while conducting rigorous experimental validation with appropriate controls. Overall this is a very nice study that cell biologist and cancer biologists should find interesting as such we enthusiastically support publication with the following areas of constructive feedback:

- 1) The authors identify genomic footprints of HRD in human lung cancer in tumors where FAT1 is lost including LST, TAI, HRD etc. The authors should look to see if evidence for micro homology which arises from mmEJ (an alternative pathways to HR) is also seen in these tumors. These regions are often seen in tumors that have lost HR.
- 2) The authors should quantify separately chromatin bridges and lagging chromosomes (in addition to lumping them together as mitotic errors). Similarly it would be nice to see side by side centric vs. acentric micronuclei upon FAT1 loss. The authors also argue that the presence of both centric and acentric micronuclei suggests that FAT1 promotes CIN through both HRD and mitotic errors. However, it's been shown that DNA damage alone can lead to mitotic errors so the centric micronuclei and lagging chromosomes could simply be the results of HRD and not independent result of FAT1 loss. The authors should qualify this in the text.
- 3) Related to the point above, it would be helpful to observe micronuclei formation (centric and acentric) as well as missegregation defects upon depletion of FAT1 as well as another bona fide HRD gene such BRCA or other listed in Fig. 1E. This could help shed light as to whether the majority of the defects from FAT1 loss are solely due to HRD.
- 4) It would be useful to include a photo of a chromosome spread upon FAT1 depletion showing chromosomal breaks as well as numerical abnormalities to support Figure 2L
- 5) The authors do a great job dissecting mitotic bypass vs. cytokinesis failure using live cell imaging. Prior work has demonstrated that cytokinesis failure can result from chromatin bridges at the mid zone (as opposed to yet another independent pathway activated by FAT1 loss). Thus it is tempting to ask whether this is a direct result from HRD. The authors should attempt to interpret their existing data to dissect these possibilities and discuss them in the text.
- 6) Does YAP1/FAT1 co-depletion rescue the HRD phenotype? This could be an important separation of function experiment if the results are negative or rather supporting the centrality of HRD in FAT1-loss-related phenotypes

Reviewer #2:

None

Reviewer #3:

Remarks to the Author:

In this study, the authors functionally test the effect of a number of candidate lung cancer driver genes identified in the TRACERx study. They identify 37 tumor suppressor genes associated with WGD and show for some of them that mutation of these genes leads to HR defects. They then more deeply follow up on one of these genes, FAT1, as this gene is commonly mutated in lung cancer. They find that loss of FAT1 leads to HR defects, increased replication stress, increased aneuploidy, increased numbers of micronuclei and an increased frequency of abnormal cytokinesis. As FAT1 is known to play a role in HIPPO signaling, they inactivate components in the HIPPO pathway and find that they can reproduce the FAT1 KO phenotype by alleviating key components of the HIPPO pathway (LATS1/LATS2) and conversely that they can rescue the FAT1 phenotype by concomitantly inactivating YAP1, a downstream transcription factor in the HIPPO signaling cue. They conclude that FAT1 mutations facilitate WGD events and thus cancer cell evolution in lung cancer.

Overall, the paper is well-written with minor suggestions for improvement (also see some structural suggestions below). The experimental design is ok overall: most conclusions are backed up through various approaches that largely yield data that agree.

General points of critique:

However, most described phenotypes are rather weak, which is not a critique per se, but this might make the results less suitable for the broad readership NCB. However, this is the editor's call, not mine. An important point of critique that I do need to make is that the authors use various cell lines throughout the paper for different types of experiments. In some cases, this is well-justified, in other cases less so. To ensure physiological relevance of all findings, there should be overlap between the cell lines being used for the different assays and per assay type 2-3 cell lines should be tested at least. Also, the choice of 'general' cell lines should be well-justified with 2 cell lines at least being used for all experiments.

Also, different methods to alleviate FAT1 are being used, sometimes siRNA, sometimes CRISPR KO. It is not completely clear why this is, and while it is not necessary to redo all experiments for both siRNA and CRISPR knockout, it would strengthen the data if key experiments of the paper are done with both siRNA and CRISPR KO approaches including all necessary replicates (multiple knockout lines per gene, i.e. independently engineered KO cell lines, ideally from different gRNAs and/or siRNAs).

Regarding the quality of the imaging data (also see some specific points below), particularly the IF images, better quality images need to be provided for most panels. This can be done by providing zoomed in (higher res) images to simplify interpretation such that the images really support the subsequent quantification.

Finally, while the authors suggest a broader mechanism, the exact mechanism of the phenotype remains unresolved: the authors suggest that loss of FAT1 via loss of YAP1 nuclear localization leads to replication stress (and an HR defect), that leads to increased numbers of lagging chromosomes, leading to more micronuclei and increased cytokinesis defects. However, how exactly YAP1 mislocalization leads to replication stress is not clear, nor whether FAT1 also influences other pathways that might also lead to replication stress and downstream genomic instability.

More specifically, a number of smaller and larger issues (presented in order of the manuscript) that the authors might want to address to improve the manuscript is copied below.

1. Page 2 line 7: "associated with, the irreversible accumulation". Comma should be removed: "associated with the irreversible accumulation".
2. Page 3, Fig S2: It is unclear what was analysed here and what the figure represents. The legend is very brief. This also makes it difficult to interpret the data towards the conclusion in line 7 that these genes are required to maintain chromosomal stability.
3. Related to this point: it is not fully clear whether HU is required to see loss of the HAC when the target genes are knocked out. This is important as a required predisposition (like HU treatment) to observe a CIN phenotype could also imply that the targeted gene helps cells tolerate CIN (like loss of p53 does), but does not drive CIN per se. This should be explained better especially as the authors continue to show only for FAT1 that loss leads to increased CIN rates.
4. In general, the authors use the term CIN quite generically. While this might be a semantic issue, CIN is most commonly used to describe an increased rate of mitotic abnormalities. Furthermore, HR defects can lead to structural CIN while WGD is a form of numerical CIN. To increase readability, it would be best if authors more explicitly separate these types of CIN and also separate the upstream replication stress and HR defects from the resulting CIN as the latter occurs in mitosis, while the former takes place during S-phase. This is also relevant for the conclusion, as in my view loss of FAT1 leads to replication/HGR problems that then promote CIN as a secondary effect (also see last points below).
5. Page 3, line 6 & Fig 1C: the effects of FAT1 on DNA damage foci loss are most obvious following HU treatment and less so following irradiation. Why is this? Maybe because loss of FAT1 leads to replication stress and HU synergizes with this? Also, this seems to be different for most of the other candidate genes. Maybe these play a role in the DNA damage response? The authors might want to comment on this in the text/discussion.

6. Page 3, line 10: "The loss of HR proficiency": while the authors explain their focus on DDR and CIN well, the focus on HR as one type of repair comes somewhat unexpected. Why focus on HR and e.g. not NHEJ or replication in general? The choice to focus on HR should be better introduced.
7. Figure S3: the western blots, especially the one for FAT1, are not very convincing. Ideally they should be replaced.
8. All figures showing IF images of various types of DNA foci (for instance Figs S4A, S5A, but many others): from the images it is not clear what is quantified: total signal or foci? I assume foci? Related to this, more zoomed images need to be shown to better appreciate the effects on the number of foci.
9. Page 3, line 26: "mutations which impair HR efficiency" should be "mutations that impair HR efficiency"
10. Sup Fig6C: the CtIP signalling is barely visible from the IF images. Please provide better (and zoomed) images so that the foci can be seen, also in a printed version.
11. The authors conclude from Sup Fig 7C that Rad51 foci are reduced in FAT1 siRNA cells, but this seems to be only the case for the 4 h time point; for the other time points the number of foci seems to be up (significant for 24 hours, and a trend for 30 mins and 1 h). So I am not sure how strong this conclusion is.
12. Page 4 – section "FAT1 ablation leads to CIN". This section should be merged with the section "FAT1 ablation leads to mitotic errors" as both sections discuss the same type of findings, only using different approaches. For instance, reading this section, I missed time lapse imaging approaches, which in fact were done and support the phenotype, but which are only discussed later in the paper. This is, in my view, confusing.
13. Page 4, line 19: the effect on number of micronuclei is quite weak. The authors should include a positive control for the induction of micronuclei, for instance by inhibiting Wee1 or Mps1 to compare their phenotype to.
14. Page 4, line 40, ratio dN/dS: were these non-synonymous mutations single amino acid substitutions or frameshifts, i.e. loss of function or only partial loss of function or even gain of function mutations? Also, why does the ratio drop in WGD tumours? Do progressed tumours select against mutation of FAT1?
15. Related to the above point: it would be good to also include absolute numbers of FAT1 mutations for both tumour types before and after WGD to put the ratios in perspective.
16. Page 4, line 43: "a significant enrichment of copy number deletion events was identified": is this in tumour(s) (sections) with or without WGD, i.e. before or after WGD? How does this compare to FAT1 mutations? I.e. is the wildtype copy lost? The decreased dN/dS ratio suggests that the wildtype copies are retained and even selected for after WGD.
17. Page 5, line 12: "mutations are selected prior to WGD in LUSC tumours": while this could well be true, it would be fair to add: "but selected against after WGD".
18. Page 5, line 15: "likely confounded by the contribution of other HR-related gene alterations": this might be the case, alternatively the effect is not significant because the mutation ratio (and rate(?)) drops after WGD. If this is an alternative explanation, it should be included in the text/discussion.
19. Fig 3E,F but also in general: when using CRISPR, showing data for 2 sgRNAs is recommended. From Fig 3E it seems the data is generated from 1 sgRNA with 4 biological replicates: are these separate KO cell lines or 4 biological replicates from 1 cell line? If the latter, authors need to repeat this experiment for 2 more separate KO cell lines (polyclonal or single clones), ideally including a second sgRNA.
20. Page 5, line 34: "at least in the initial cell cycle": it is not completely clear to me where this conclusion comes from. Either remove or explain better.
21. Fig 3F: why are there up to 30% of cells in a tetraploid S-phase in nocodazole arrested cells: how long have the cells been treated with EdU? Did cells start leaking out of mitosis? It might be worthwhile to optimise the experimental conditions (e.g. 16h of noco arrest), which might lower the error bars for the noco-treated conditions.
22. Fig 3G: the statistical test between HU treated control vs Fat1 KD cells is missing. It looks like that there are fewer cells bypassing mitosis in AMP-treated cells that are devoid of FAT1. Is this not the opposite of what you would expect? Please add the test and comment on this in the results.
23. Fig 13A: also for these type of images, zoomed in versions are required. How is multinucleation quantified?
24. Page 6, line 14: "blebbing": I thought that 'blebbing' referred to abnormal dynamic behaviour of a cell during mitosis. I would call this 'nuclear shape abnormalities' or something along those lines. Looking at Sup Fig 14B, the blebbing looks like partial binucleation to me, i.e. failed/incomplete cytokinesis. This fits well with an increased rate of WGD due to tetraploidization in cases where cytokinesis fails altogether.

25. Related to the above and the conclusion on page 6, lines 29-31: the conclusions suggest that FAT1 is involved in two separate processes related to replication/HR and chromosome segregation: Is this phenotype not simply a consequence of the HR-defect-induced replication stress that leads to lagging chromosomes and ultimately when not resolved by decatenation into cytokinesis failure? Also see related points below.

26. The section on heterogeneity in the next section (page 6 lines 37-page 7 line 5) shows similar data as presented in the previous section and is a better fit there. This will simplify comparing the effects between cell lines at once.

27. The section on evolution (page 7, lines 7-14) is quite shallow. To strengthen the point that FAT1 depletion drives cancer cell evolution, more experiments are required, possibly using other drugs, particularly drugs that benefit from copy number changes. Alternatively the authors could leave this out or deemphasize these experiments and include them into the CIN section as well.

28. Related to the above point: Can the authors rule out that Osimertinib resistance arose from EGFR gene mutations?

29. Sup Fig 17E: From the figure it appears that the basal ploidy of PC9 cells is 6, but the text says these cells are triploid. Which of the two is correct?

30. Page 7, line 26: "elevated nuclear localization": where does the YAP come from: from the images in Fig 4B there does not appear to be a cytoplasmic pool. In Sup Fig 18A there seems to be some cytoplasmic YAP, but these images look quite overexposed when looking at the nuclear pool. Are there additional ways to show translocation, e.g. by isolating cytoplasmic and nuclear fractions?

31. Fig 4C-E; Sup Fig 18C: To see if signalling indeed goes through FAT1 (alone), a DKO approach should be used: i.e. by combining FAT1KO with LATS1 or LATS2 KO to determine whether the effect is indeed epistatic. The final experiment with YAP1/FAT1 DKO partially addresses the epistatic relation of FAT1 in the HIPPO pathway, but not completely.

32. Figure 4F only shows data for LATS1. The authors should also include relevant data for LATS2.

33. Page 9, lines 5-6: "that were associated with chromosomal instability (CIN)": related to several points raised before: from the data in this paper, it seems that the CIN resulting from FAT1 ablation is secondary to the replication/HR defects and in my view it would be better to present that data as such. Related to this it would help to add a summary figure of the proposed mechanism towards the end of the results or early in the discussion on page 9.

Methods should be written concisely, but should contain all elements necessary to allow interpretation and replication of the results. As a guideline, Methods sections typically do not exceed 3,000 words. The Methods should be divided into subsections listing reagents and techniques. When citing previous methods, accurate references should be provided and any alterations should be noted. Information must be provided about: antibody dilutions, company names, catalogue numbers and clone numbers for monoclonal antibodies; sequences of RNAi and cDNA probes/primers or company names and catalogue numbers if reagents are commercial; cell line names, sources and information on cell line identity and authentication. Animal studies and experiments involving human subjects must be reported in detail, identifying the committees approving the protocols. For studies involving human subjects/samples, a statement must be included confirming that informed consent was obtained. Statistical analyses and information on the reproducibility of experimental results should be provided in a section titled "Statistics and Reproducibility".

All Nature Cell Biology manuscripts submitted on or after March 21 2016 must include a Data availability statement as a separate section after Methods but before references, under the heading "Data Availability". For Springer Nature policies on data availability see <http://www.nature.com/authors/policies/availability.html>; for more information on this particular policy see <http://www.nature.com/authors/policies/data/data-availability-statements-data-citations.pdf>. The Data availability statement should include:

- Accession codes for primary datasets (generated during the study under consideration and designated as "primary accessions") and secondary datasets (published datasets reanalysed during the study under consideration, designated as "referenced accessions"). For primary accessions data should be made public to coincide with publication of the manuscript. A list of data types for which submission to community-endorsed public repositories is mandated (including sequence, structure, microarray, deep sequencing data) can be found here <http://www.nature.com/authors/policies/availability.html#data>.
- Unique identifiers (accession codes, DOIs or other unique persistent identifier) and hyperlinks for datasets deposited in an approved repository, but for which data deposition is not mandated (see here for details <http://www.nature.com/sdata/data-policies/repositories>).
- At a minimum, please include a statement confirming that all relevant data are available from the authors, and/or are included with the manuscript (e.g. as source data or supplementary information), listing which data are included (e.g. by figure panels and data types) and mentioning any restrictions on availability.
- If a dataset has a Digital Object Identifier (DOI) as its unique identifier, we strongly encourage including this in the Reference list and citing the dataset in the Methods.

We recommend that you upload the step-by-step protocols used in this manuscript to the Protocol Exchange. More details can be found at www.nature.com/protocolexchange/about.

FIGURES – Colour figure publication costs \$600 for the first, and \$300 for each subsequent colour figure. All panels of a

multi-panel figure must be logically connected and arranged as they would appear in the final version. Unnecessary figures and figure panels should be avoided (e.g. data presented in small tables could be stated briefly in the text instead).

All imaging data should be accompanied by scale bars, which should be defined in the legend.

Cropped images of gels/blots are acceptable, but need to be accompanied by size markers, and to retain visible background signal within the linear range (i.e. should not be saturated). The boundaries of panels with low background have to be demarked with black lines. Splicing of panels should only be considered if unavoidable, and must be clearly marked on the figure, and noted in the legend with a statement on whether the samples were obtained and processed simultaneously. Quantitative comparisons between samples on different gels/blots are discouraged; if this is unavoidable, it should only be performed for samples derived from the same experiment with gels/blots were processed in parallel, which needs to be stated in the legend.

The total number of Supplementary Figures (not including the "unprocessed scans" Supplementary Figure) should not exceed the number of main display items (figures and/or tables (see our Guide to Authors and March 2012 editorial <http://www.nature.com/ncb/authors/submit/index.html#supinfo>; <http://www.nature.com/ncb/journal/v14/n3/index.html#ed>). No restrictions apply to Supplementary Tables or Videos, but we advise authors to be selective in including supplemental data.

GUIDELINES FOR EXPERIMENTAL AND STATISTICAL REPORTING

REPORTING REQUIREMENTS – We are trying to improve the quality of methods and statistics reporting in our papers. To that end, we are now asking authors to complete a reporting summary that collects information on experimental design and reagents. The Reporting Summary can be found here <https://www.nature.com/documents/nr-reporting-summary.pdf>. If you would like to reference the guidance text as you complete the template, please access these flattened versions at <http://www.nature.com/authors/policies/availability.html>.

Version 1:

Decision Letter:

Our ref: NCB-LE51663A

26th June 2024

Dear Dr. Swanton,

Thank you for submitting your revised manuscript "The role of FAT1 and the Hippo pathway in chromosomal instability and whole-genome doubling" (NCB-LE51663A). It has now been seen by the original referees and their comments are below. The reviewers find that the paper has improved in revision, and therefore we'll be happy in principle to publish it in Nature Cell Biology, pending minor revisions to satisfy the referees' final requests and to comply with our editorial and formatting guidelines.

Thank you again for your interest in Nature Cell Biology Please do not hesitate to contact me if you have any questions.

Sincerely,

Zhe Wang, PhD
Senior Editor
Nature Cell Biology

Tel: +44 (0) 207 843 4924
email: zhe.wang@nature.com

Reviewer #1 (Remarks to the Author):

I would like to congratulate the authors on a very extensive and thorough revision of this work. This is a very interesting study that will be of interest to the cell and cancer biology communities. I wholeheartedly support publication of this work and have no further comments.

Reviewer #3 (Remarks to the Author):

The authors have added a significant body of extra data to their study. Importantly, they have significantly strengthened the data on the mechanisms that underlies the genomic instability driven by FAT1 loss. They manage to separate the HR/DNA damage effects from CIN effects and also show that the aberrant activity of YAP1 (via LATS1) drives the WGD phenotype, but not the HR defects. While this does not completely explain how FAT1 loss drives the various genomic instability phenotypes, this additional work is a substantial step forward and, combined with all the additional controls for me now makes the revised study suitable for the broad readership of NCB.

Version 2:

Decision Letter:

Dear Dr Swanton,

I am pleased to inform you that your manuscript, "TRACERx analysis identifies a role for FAT1 in regulating chromosomal instability and whole-genome doubling via Hippo signaling", has now been accepted for publication in Nature Cell Biology.

Once your paper has been scheduled for online publication, the Nature press office will be in touch to confirm the details. An online order form for reprints of your paper is available at <https://www.nature.com/reprints/author->

reprints.html"><https://www.nature.com/reprints/author-reprints.html>. All co-authors, authors' institutions and authors' funding agencies can order reprints using the form appropriate to their geographical region.

Please note that *Nature Cell Biology* is a Transformative Journal (TJ). Authors may publish their research with us through the traditional subscription access route or make their paper immediately open access through payment of an article-processing charge (APC). Authors will not be required to make a final decision about access to their article until it has been accepted. Find out more about Transformative Journals

Authors may need to take specific actions to achieve compliance with funder and institutional open access mandates. If your research is supported by a funder that requires immediate open access (e.g. according to Plan S principles) then you should select the gold OA route, and we will direct you to the compliant route where possible. For authors selecting the subscription publication route, the journal's standard licensing terms will need to be accepted, including self-archiving policies. Those licensing terms will supersede any other terms that the author or any third party may assert apply to any version of the manuscript.

If you have not already done so, we strongly recommend that you upload the step-by-step protocols used in this manuscript to protocols.io (<https://protocols.io>), an open online resource that allows researchers to share their detailed experimental know-how. All uploaded protocols are made freely available and are assigned DOIs for ease of citation. Protocols and Nature Portfolio journal papers in which they are used can be linked to one another, and this link is clearly and prominently visible in the online versions of both. Authors who performed the specific experiments can act as primary authors for the Protocol as they will be best placed to share the methodology details, but the Corresponding Author of the present research paper should be included as one of the authors. By uploading your Protocols onto protocols.io, you are enabling researchers to more readily reproduce or adapt the methodology you use, as well as increasing the visibility of your protocols and papers. You can also establish a dedicated workspace to collect your lab Protocols. Further information can be found at <https://www.protocols.io/help/publish-articles>.

With kind regards,

Zhe Wang, PhD
Senior Editor
Nature Cell Biology

Tel: +44 (0) 207 843 4924
email: zhe.wang@nature.com

** Visit the Springer Nature Editorial and Publishing website at http://editorial-jobs.springernature.com?utm_source=ejp_NCB_email&utm_medium=ejp_NCB_email&utm_campaign=ejp_NCB for more information about our career opportunities. If you have any questions please click [here](mailto:editorial.publishing.jobs@springernature.com). **

Point-by point responses to the comments of the reviewers:

Reviewer #1:

Remarks to the Author:

In this manuscript, Lu et al. build on their observation in human lung cancer samples to identify and validate a novel role for FAT1 as a driver of genomic instability. They first make the observation that FAT1 loss induces homologous recombination deficiency (HRD) then show that it also coincides with increased levels of chromosomal instability (CIN) manifested by elevated rates of chromosome missegregation and micronuclei formation. They go on to show that FAT1 depleted cells undergo increased rates of whole genome doubling (WGD) derived from cytokinesis failure and nuclear deformity. As a result of increased CIN and WGD they demonstrate that FAT1-depleted cells are more likely to exhibit treatment resistance. Finally, the authors show that FAT1 loss alters the HIPPO pathway in cancer cells and demonstrate some interesting rescue experiments of some of the phenotypes by co-depletion of FAT1 and YAP1 a key member of this pathway.

Overall, this is an impressive body of work that identifies a poorly characterized molecular driver of lung cancer evolution and an important mechanism of CIN in human cancer. The authors have covered significant ground in phenomenologically characterizing the cellular consequences of FAT1 loss while conducting rigorous experimental validation with appropriate controls. Overall this is a very nice study that cell biologist and cancer biologists should find interesting as such we enthusiastically support publication with the following areas of constructive feedback:

1) The authors identify genomic footprints of HRD in human lung cancer in tumors where FAT1 is lost including LST, TAI, HRD etc. The authors should look to see if evidence for micro homology which arises from mmEJ (an alternative pathway to HR) is also seen in these tumors. These regions are often seen in tumors that have lost HR.

We thank the reviewer for the kind words above and their excellent suggestion.

To substantiate our observation that the loss of *FAT1* significantly downregulates HR repair, we obtained EJ-5 and EJ-2 reporters from Jeremy Stark's lab to assess canonical end-joining and alternative end-joining efficiencies respectively. As shown below in Fig R1C1A, EJ-5 distal end-joining reporter; R1C1B, EJ-2 alternative end-joining reporter (integrated into the revised figure S12AB in the manuscript), FAT1 knockdown significantly reduces the efficiency of alternative end-joining but does not significantly affect distal end-joining efficiency. It should also be noted that, unlike the loss of key DDR genes *ATM* and *NBS1* (Bennardo *et al*, 2008; Bhargava *et al*, 2017), FAT1 knockdown does not result in an upregulation of the distal end-joining. It is known that alternative end-joining requires some end-resection activity at the DNA breaks (Bennardo *et al*, 2008), which is in line with our previous observation that FAT1 knockdown impairs DNA end-resection.

To assess the impact of *FAT1* loss on microhomology arising from MMEJ in the genome, we utilized Genomics England WGS sequencing data (since the majority of TRACERx data is WES). Despite observing a significant increase in indel counts in *FAT1* mutated samples, we did not observe an enrichment of COSMIC mutational profiles ID6 and SBS3 (Fig R1C1C-E below), both associated with indels as a result of end-joining (Alexandrov *et al*, 2020).

In addition, we further strengthened our HR mutational signature data with a new analysis of TRACERx sequencing data. In line with the previous TCGA results, we observed a significant increase in HR-deficient related measurements LST and TAI rates in *FAT1* mutated tumour samples (see Fig R1C1F-H below). In conclusion, utilizing bioinformatics and experimental data from TCGA, TRACERx and Genomics England, we propose that *FAT1* loss significantly affects HR repair efficiency. We appreciate the reviewer's suggestion which encouraged us to perform this more comprehensive analysis.

Fig R1C1. **A:** Left, cartoon showing the design of the end-joining reporter integrated in U2OS cells. Right, 53BP1 siRNA knockdown but not FAT1 knockdown affects the distal end-joining rate. **B:** Left, cartoon showing the

design of the alternative end-joining reporter integrated in U2OS cells. Right, FAT1 siRNA knockdown significantly reduces alternative end-joining efficiency. **C:** Box plot comparing the number of indels in FAT1 WT vs mutated tumors in the Genomics England LUAD and LUSC cohort, Wilcoxon Rank-Sum test. **D-E:** Mutational profile cartoon (Left) and Genomic England mutational signature analysis (right) showing the mutational profile of COSMIC ID6 (C) and SBS3 (D) mutational signatures, both dependent on indels as a result of end-joining activity. **F-H:** Right, Cartoon depicting examples of HR-deficient related measurements such as LST (F), TAI (G) and HRD-LOH (H); middle and right, TCGA LUAD data correlating FAT1 copy number loss (middle), significance was determined using permutation test; TRACERx LUAD data correlating FAT1 driver mutations (right) with LST(F), TAI(G) and HRD-LOH(H), significance was determined using a mixed-effects linear model with purity as a fixed covariate and tumour as a random variable.

We have now incorporated the new data as Fig3 D-F, Fig 3I and FigS12.

We added the following sentences in the manuscript to reflect this line of investigation.

Sentences starting on page4 line 34 “Consistently, by analysing both the TRACERx and TCGA LUAD datasets, we observed a correlation between FAT1 loss and genomic signatures associated with HR deficiency (HRD)³¹⁻³⁴, including elevated telomeric allelic imbalance (TAI), large-scale transitions (LST) and loss of heterozygosity (LOH) (Fig3D-F). Using established reporters for total and alternate end-joining activity³⁵, we confirmed that transient FAT1 siRNA depletion reduced alternative end-joining efficiency but did not significantly reduce distal end-joining (FigS12AB). Indeed, utilizing WGS data from Genomics England, no significant difference was observed in ID6 and SBS3 mutational profiles, which are both associated with indels due to NHEJ activity³⁶.”

Sentences starting on page 5 line 2 “Indeed, our analysis of the TCGA and Genomics England dataset revealed that FAT1 loss correlated with an increased Weighted Genome Instability Index (wGII) and total indel count respectively, indicating an elevation of structural and numerical CIN (Fig3HI).”

References:

Alexandrov LB, Kim J, Haradhvala NJ, Huang MN, Tian Ng AW, Wu Y, Boot A, Covington KR, Gordenin DA, Bergstrom EN et al (2020) The repertoire of mutational signatures in human cancer. Nature 578: 94-101

Bennardo N, Cheng A, Huang N, Stark JM (2008) Alternative-NHEJ is a mechanistically distinct pathway of mammalian chromosome break repair. PLoS Genet 4: e1000110

Bhargava R, Carson CR, Lee G, Stark JM (2017) Contribution of canonical nonhomologous end joining to chromosomal rearrangements is enhanced by ATM kinase deficiency. Proc Natl Acad Sci U S A 114: 728-733

2) The authors should quantify separately chromatin bridges and lagging chromosomes (in addition to lumping them together as mitotic errors).

We apologise for this omission. We included the data on chromatin bridges (revised Fig4C), but we omitted the lagging chromosome figures due to space constraints. In the revised figures, the chromatin bridges and lagging centromeric and acentric chromosome formation rates are quantified and included in revised Fig4C and revised FigS13DE. For ease of the reviewing process, we have consolidated these figures below as FigR1C2 (A, total error; B, chromatin bridges; C, acentric lagging chromosomes; and D centromeric lagging chromosomes). In line with other data presented in the manuscript, FAT1 knockdown significantly increases the of chromatin bridges, and acentric and centromeric lagging chromosomes, particularly when U2OS cells are challenged with the replication stress inducer hydroxyurea.

U2OS

5 hr 4mM HU 24 hr recovery

FigR1C2, A,B; Transient FAT1 knockdown significantly increases mitotic error rate (lagging chromosomes + DAPI bridges; A) and the occurrence of DAPI bridges (B) in U2OS cells. C and D, Transient FAT1 siRNA in U2OS cells significantly increases the number of lagging chromosomes per cell after replication stress (C, centromeric; D, acentric). *** $p < 0.001$, ** $p < 0.01$, * $p < 0.05$, Dunn's test. Over 100 mitotic cells were scored across 3 biological replicates.

For live-cell imaging movies performed using spinning disk microscopy, mitotic errors mainly consist of chromosomal bridges, possibly because they are relatively easier to spot during live-cell microscopy. Also, it should be noted that RPE1-FUCCI cells were used for the live cell microscopy. Unlike U2OS, RPE1 are untransformed cells, and the loss of *FAT1* can spontaneously elevate chromatin bridge formation (revised FigS16C and S18B).

We thank the reviewer for their suggestion and we have now added *new FigS13DE to support Fig4C. Chromatin bridges, lagging centromeric chromosomes and lagging acentric chromosome rates are quantified in Fig4C, and Fig S13DE respectively.*

Similarly, it would be nice to see side by side centric vs. acentric micronuclei upon *FAT1* loss.

Again, we apologise for not highlighting our figures clearly in the text, side-by-side comparisons between centric and acentric micronuclei were previously included in revised figure 4B (also shown below). We observed that *FAT1* loss significantly increased the rate of both centric and acentric micronuclei formation in U2OS cells.

Manuscript Fig4B, Transient FAT1 siRNA knockdown induces the formation of both acentric and centromeric micronuclei. Scale bar=5 μ M. * $p < 0.05$, ** $p < 0.01$, *** $p < 0.001$; One-way ANOVA with Bonferroni correction, $n = 4(B)$

The authors also argue that the presence of both centric and acentric micronuclei suggests that FAT1 promotes CIN through both HRD and mitotic errors. However, it's been shown that DNA damage alone can lead to mitotic errors so the centric micronuclei and lagging chromosomes could simply be the results of HRD and not independent result of FAT1 loss. The authors should qualify this in the text.

We thank the reviewer for their suggestion. Indeed, HRD can by itself lead to elevated mitotic errors and centric micronuclei (Chan *et al*, 2018; Daniels *et al*, 2004; Di Bona & Bakhoun, 2024). These effects may arise due to elevated HRD caused by *FAT1* loss. We have altered our discussion to show appreciation for this possibility.

Text starting on page 10 line 19 in the discussion section, now reads: "*We demonstrate that FAT1 depletion attenuates HR and causes unresolved replication stress, both of which likely contribute to mitotic errors and micronuclei formation*"⁴⁰

References:

Chan YW, Fugger K, West SC (2018) Unresolved recombination intermediates lead to ultra-fine anaphase bridges, chromosome breaks and aberrations. *Nat Cell Biol* 20: 92-103

Daniels MJ, Wang Y, Lee M, Venkitaraman AR (2004) Abnormal cytokinesis in cells deficient in the breast cancer susceptibility protein BRCA2. *Science* 306: 876-879

Di Bona M, Bakhoun SF (2024) Micronuclei and Cancer. *Cancer Discov*: OF1-OF13

3) Related to the point above, it would be helpful to observe micronuclei formation (centric and acentric) as well as missegregation defects upon depletion of FAT1 as well as another bona fide HRD gene such BRCA or other listed in Fig. 1E. This could help shed light as to whether the majority of the defects from FAT1 loss are solely due to HRD.

We thank the reviewer for this excellent suggestion. Indeed, comparing the rate of micronuclei formation between FAT1 loss versus the loss or inhibition of key players in HR will provide more information regarding the mechanisms of action.

Coincidentally, in a recent publication, the Lee Zou lab used the same U2OS system as our study to investigate the outcome of inhibiting key upstream HR mediator ATR in micronuclei generation (Joo *et al*, Mol Cell 2023). The result is highly comparable to our findings (see below, left: fig1A of Joo *et al* vs right: revised fig 4A in this manuscript).

REDACTED

Joo and co-workers used 2 different inhibitors of ATR, VE821 and AZ20 demonstrating that pharmacological inhibition of ATR increases micronuclei formation rate in U2OS cells by 1.5 fold; which is highly comparable to our observations with FAT1 depletion. Furthermore, upon exposure to replication stress (aphidicolin in the case of Joo et al, Hydroxyurea in our case) a 2-fold increase in micronuclei formation rate can be seen, again highly comparable to our observations.

We wholeheartedly agree with the reviewer that it is crucial to delineate whether the majority of the defects from *FAT1* loss are solely due to HRD. To this end, we have now performed further experiments to investigate this. These analyses are detailed below in our response to Reviewer 1 comment 6.

References:

Joo YK, Black EM, Trier I, Haakma W, Zou L, Kabeche L (2023) ATR promotes clearance of damaged DNA and damaged cells by rupturing micronuclei. *Mol Cell* 83: 3642-3658 e3644

4) It would be useful to include a photo of a chromosome spread upon FAT1 depletion showing chromosomal breaks as well as numerical abnormalities to support Figure 2L

We thank the reviewer for this suggestion (*NB*, Fig2L is Fig4H in the revised manuscript).

We have now repeated the metaphase spreads experiments and analyses with a microscope with higher resolution, as shown below (Fig R1C4). In line with our previous observations, we have observed an increase in mitotic chromosomal aberrations when *FAT1* KO A549 cells were challenged with aphidicolin-induced replication stress.

The new results are incorporated in the manuscript as revised Fig4E, and FigS13F

Fig R1C4, A, Histogram (left) showing cells with FAT1 loss exhibit a significantly increased number of chromosomal aberrations upon challenge with replication stress induced by aphidicolin treatment. $**p < 0.01$, $***p < 0.001$, one-way ANOVA with Holm-Sidak corrections, 60 metaphases were scored across 3 biological replicates per condition. **B**, representative image of radial chromosomes and chromosome gaps observed in FAT1 KO A549 cells. **C**, further representative images comparing metaphase spreads of *FAT1* WT vs *FAT1* KO cells.

5) The authors do a great job dissecting mitotic bypass vs. cytokinesis failure using live cell imaging. Prior work has demonstrated that cytokinesis failure can result from chromatin bridges at the mid zone (as opposed to yet another independent pathway activated by FAT1 loss). Thus it is tempting to ask whether this is a direct result from HRD. The authors should attempt to interpret their existing data to dissect these possibilities and discuss them in the text.

We thank the reviewer for their kind words and their important points concerning cytokinesis failure. Our revised data now highlights that deregulated YAP1 is both required for and sufficient to cause the observed whole-genome-duplication (please see the detailed explanation in the response to reviewer 1 comment 6, also figure 7 and the associated text starting on page 8 line 37 of the revised manuscript). We propose that sole dysregulation of YAP1 triggers the WGD, and the depletion of YAP1 in *FAT1* KO cells does not impact the HRD phenotypes tested. Overall, these lines of evidence support the new concept that we propose: of two complementary phenotypes downstream of *FAT1* loss, as depicted in our new Model (FigS26, also response to Reviewer 3 Comment 33)

Indeed, it is possible that HRD/midbody defects might contribute to the observed cytokinetic failure, however our data points to two complementary sets of phenotypes, where HRD and the ensuing replication stress and structural CIN are independent of YAP1, while the robust cytokinesis failure phenotype/ numerical CIN reflects the impact of deregulated YAP1. Please see also our response to comment 6 for detailed results. While our results cannot formally exclude any contribution of HRD to

the observed cytokinesis failure in FAT1 deficient cells, we have no direct evidence to support that such an HRD-dependent route directly qualifies cytokinesis failure. In fact, it is known that the abscission checkpoint pathway mediated by Aurora B exists to suppress whole genome doubling in the presence of chromatin bridges (Steigemann *et al*, 2009).

To reflect these considerations, we have altered the sentence starting on page 10 line 19 in the discussion section, which now reads: *“We demonstrate that FAT1 depletion attenuates HR and causes unresolved replication stress, both of which are likely contributors to mitotic errors and micronuclei formation³⁰”*

We also added the following sentence in the discussion (page 11, starting on line 16) to highlight that YAP1 dysregulation is responsible for the elevated cytokinesis failure rate observed in our experiments. Accordingly, the sentences starting on page 11, starting on line 16 now reads: *“Conversely, FAT1/YAP1 co-depletion rescued cytokinesis failure and WGD. Strikingly, the overexpression of the constitutively active mScarlet-YAP15SA protein is sufficient to drive WGD and contribute to numerical CIN. Taken together, our data reveal that FAT1 alterations may contribute to HRD, CIN and WGD through two distinct mechanisms (see FigS26)”*

References:

Steigemann P, Wurzenberger C, Schmitz MH, Held M, Guizetti J, Maar S, Gerlich DW (2009) Aurora B-mediated abscission checkpoint protects against tetraploidization. *Cell* 136: 473-484

6) Does YAP1/FAT1 co-depletion rescue the HRD phenotype? This could be an important separation of function experiment if the results are negative or rather supporting the centrality of HRD in FAT1-loss-related phenotypes.

We thank the reviewer for this suggestion. The issue regarding separation-of-function was raised multiple times by both reviewers and we agree that the experiment suggested above is essential to address this point. Due to the importance of this matter, we performed an in-depth analysis to carefully dissect the relative contribution of FAT1 and YAP1 in response to HR activation, mitotic errors, and WGD (see Fig R1C6, Page 16 of the rebuttal).

To this end, we first utilised the well-known HR reporter assay (Seluanov *et al*, 2010) in the DNA damage field, which we previously used in Fig1E in our manuscript. To our surprise, double siRNA knockdown of FAT1 and YAP1 did not rescue HR efficiency (Fig R1C6, panel A). To further confirm that FAT1/YAP1 co-depletion did not rescue the HRD phenotype, we also performed the DSB resection assay using the well-characterised U2OS-AsiSI cells (Iacovoni *et al*, 2010). Similar to the HR reporter assay, we observed that co-depletion of both FAT1 and YAP1 did not reconstitute the ssDNA resection rate that is required for efficient HR (Fig R1C6, panel B). It should be noted that the depletion of YAP1 by itself reduces HR efficiency, possibly due to a reduction of the S-phase cell population (Fig R1C6, panel C, left). However, co-depletion of FAT1 and YAP1 reverted the cell cycle distribution to wild-type levels, despite failing to rescue the HR deficiency associated with FAT1 knockdown (Fig R1C6, panel ABC). To further confirm our results in a lung cancer cell line, we used RNAi to knock down YAP1 in FAT1 WT/KO A549 cell lines. Consistent with our previous observation in U2OS cells, YAP1 knockdown altered cell cycle progression in FAT1 WT, but not FAT1 KO A549 cells (Fig R1C6, panel C right). In line with our other experiments, in A549 cells co-depletion of FAT1 and YAP1 did not rescue the RAD51 foci formation rate that is required for efficient HR repair in response to ionizing radiation (Fig R1C6, panel D). Taken together, these data demonstrate that FAT1/YAP1 co-depletion fails to rescue the HRD phenotype associated with FAT1 loss, at least in response to DNA double-strand breaks.

Next, as mentioned by both reviewers, unresolved recombination intermediates can form midbody chromosomal bridges due to HRD (Chan *et al.*, 2018; Daniels *et al.*, 2004). Thus, we next assessed

whether YAP1/FAT1 co-depletion might also rescue the formation of chromosomal bridges. In line with our previous results related to HR deficiency (Fig R1C6, panel ABD), YAP1/FAT1 co-depletion failed to rescue the elevated mitotic error rate and chromosomal bridges observed in *FAT1* KO A549 cells (FigR1C6, panel E). This observation is in line with published literature reporting that unresolved HRD intermediates induce chromosomal bridge formation (Chan *et al.*, 2018; Daniels *et al.*, 2004).

Taken together, one can hypothesize that FAT1 potentially possesses dual roles. One outcome of the depletion of FAT1 is HRD and leads to unresolved recombination intermediates which leads to nuclear deformation (visible in movie 4; also FigR1C6, panel F, right) – these events appear to be YAP1 independent. In contrast, the elevated rate of cytokinesis failure and subsequent WGD appear to be dependent on YAP1 (FigR1C6, panel F, left).

To further test the above hypothesis, we repeated the EdU incorporation assay (see revised Fig5AB) to visualize the impact of constitutively active YAP1 protein on WGD. Accordingly, we modified the YAP1^{5SA} construct (Zhao *et al.*, 2007) with an N-terminal mScarlet tag, and transiently overexpressed the constitutively active YAP1^{5SA} protein in *FAT1* WT/KO PC9 cells. Using flow cytometry, we separated mScarlet high and negative populations and assessed WGD by monitoring EdU incorporation beyond the G2 cell cycle phase. It should be noted that the hyperactivated YAP1^{5SA} construct is a very potent activator of TEAD transcriptional activity, capable of activating TEAD transcription beyond the level of FAT1 loss (which we will dissect further in our response to Reviewer 3 – pages 24-26 of this document). Indeed, upon YAP1^{5SA} overexpression, we observed a significant and consistent increase in the WGD population in both *TP53* WT/KO RPE1 cells, and *FAT1* WT/KO PC9 cells (FigR1C6, panel GH), suggesting that constitutively active YAP1 is sufficient to promote WGD. This observation confirmed our hypothesis above where that *FAT1* loss introduces two distinct phenotypes: HRD and mitotic defects, which are YAP1 independent; and the induction of WGD, which is dependent on YAP1 dysregulation.

Fig R1C6, A, DR-GFP Homologous Recombination reporter assay to assess the impact of FAT1/YAP1 siRNA co-depletion; MRE11 siRNA serves as a positive control. HR efficiencies are normalized to control samples. one-way ANOVA with Sidak correction, $***p < 0.001$. Error bars=SD, $n \geq 3$ biological replicates. **B**, ssDNA resection rate following FAT1/YAP1 siRNA co-depletion using the DivA U2OS-AsiSI site-directed resection assay. Paired T-test, $***p < 0.001$. Error bars =SD, $n=3$ biological replicates. **C**, Histograms representing cell cycle profiles following the loss of FAT1, YAP1, or combined loss of both FAT1 and YAP1 genes in U2OS-AsiSI cells (left) and A549 FAT1 WT/KO cells (right), $N=3$, one-way ANOVA with Sidak corrections. $*p < 0.05$, $**p < 0.01$. **E**, Box plots showing RAD51 foci formation following the loss of FAT1, or combined loss of both FAT1 and YAP1 genes after 5Gy IR irradiation, >380 cells were quantified over 3 biological replicates, $*p < 0.05$, $***p < 0.001$, Dunn's test. **D**, Plots showing the quantification of mitotic error rates in A549 cells after 24 hours of aphidicolin treatment.

FAT1 WT or KO cells were transiently depleted with YAP1 RNAi. Mitotic error: * $p < 0.05$, ** $p < 0.01$, one-way ANOVA with Holm-Sidak multiple correction, error bars=SD. DAPI bridges and FANCD2 flanked DAPI bridges: * $p < 0.05$, ** $p < 0.01$, *** $p < 0.001$, Dunn's test. $N > 100$ cells over 3 biological repeats. **F**, FAT1/YAP1 double siRNA knockdown fully rescues the failed cytokinesis introduced by FAT1 knockdown (left) but only partially ameliorates the nuclear envelope deformation (right). *** $p < 0.001$, * $p < 0.05$, one-way ANOVA, at least 5 biological replicates were quantified per condition. **G**, Histograms illustrating the total (left) WGD populations in TP53 WT/KO RPE1 cells, with or without transient mScarlet-YAP^{55A} transfection. * $p < 0.05$, ** $p < 0.01$, one-way ANOVA. **H**, Histograms illustrating the total (left) and normalized (right) WGD populations in FAT1 WT/KO PC9 cells, with or without transient mScarlet-YAP^{55A} transfection. Left, * $p < 0.05$, ** $p < 0.01$, one-way ANOVA; right, * $p < 0.05$, Friedman Test.

Again, we thank both reviewers for their astute suggestion. We feel the manuscript benefited significantly from this line of investigation. The new results are incorporated in the revised manuscript as revised Fig 7 and Fig S25.

We have also highlighted this finding in a revised section, starting on page 8 line 37:

“Co-depletion of FAT1 and YAP1 reverses WGD but not HR deficiency

Given the involvement of FAT1 in the Hippo signalling pathway (Fig6BC), we proceeded to systematically delineate whether both WGD, numerical and structural CIN associated with FAT1 loss can be reversed by co-depleting YAP1. We first investigated whether the HR deficiency might be reversed by FAT1/YAP1 co-depletion. Surprisingly, despite FAT1/YAP1 co-depletion reversing the cell-cycle arrest associated with YAP1 single knockdown (FigS24AB), the co-depletion of FAT1/YAP1 did not rescue HR activation defects after DSB formation (Fig7A-C). This result was observed using orthogonal methods including the DR-GFP reporter assay, ssDNA formation after endonuclease-induced DSBs, and RAD51 foci formation after IR (Fig7A-C). Next, since unresolved recombination intermediates due to HR repair deficiency can cause mitotic errors^{57, 58}, we next quantified the rate of mitotic errors. YAP1/FAT1 co-depletion failed to rescue the elevated mitotic error and chromosomal bridges observed in FAT1 KO A549 cells (Fig7DE, FigS24C).

We next investigated whether YAP1 co-depletion might rescue the CIN phenotypes associated with FAT1 loss. Live-cell imaging experiments demonstrated that co-depletion of FAT1 and YAP1 reversed the cytokinesis failure phenotypes (Fig7F left, FigS24D). However, the nuclear envelope deformation observed following FAT1 depletion was only partially rescued (Fig7F right).

Based on these observations, conceivably FAT1 possesses a dual role. One outcome of FAT1 loss is HRD and leads to unresolved recombination intermediates, replication stress, structural CIN and nuclear deformation (Fig3,4,7A-D) – these events appear to be YAP1 independent. In contrast, the elevated rate of cytokinesis failure and subsequent WGD appear to be dependent on YAP1 activity (Fig7F). To test this hypothesis, we performed an EdU incorporation assay (Fig5AB) to visualize the impact of constitutively active mScarlet-YAP155A overexpression on WGD. Consistent with a previous report⁵, mScarlet-YAP1^{55A} overexpression promoted WGD in RPE1 cells (Fig7G). This was observed independent of TP53 status. Concurrently, we observed an emerging WGD population in FAT1 WT and FAT1 KO PC9 cells (Fig7H). Taken together, these results suggest that constitutively active YAP1 is sufficient to promote WGD. We postulate that FAT1 loss might drive genome instability through two different routes, one through WGD, dependent on YAP1, and the second through HR deficiency, driving structural CIN, in a TEAD/YAP1-independent manner.”

References:

Chan YW, Fugger K, West SC (2018) Unresolved recombination intermediates lead to ultra-fine anaphase bridges, chromosome breaks and aberrations. *Nat Cell Biol* 20: 92-103

Daniels MJ, Wang Y, Lee M, Venkitaraman AR (2004) Abnormal cytokinesis in cells deficient in the breast cancer susceptibility protein BRCA2. *Science* 306: 876-879

Iacovoni JS, Caron P, Lassadi I, Nicolas E, Massip L, Trouche D, Legube G (2010) High-resolution profiling of gammaH2AX around DNA double strand breaks in the mammalian genome. *EMBO J* 29: 1446-1457

Seluanov A, Mao Z, Gorbunova V (2010) Analysis of DNA double-strand break (DSB) repair in mammalian cells. *J Vis Exp*

Zhao B, Wei X, Li W, Udan RS, Yang Q, Kim J, Xie J, Ikenoue T, Yu J, Li L et al (2007) Inactivation of YAP oncoprotein by the Hippo pathway is involved in cell contact inhibition and tissue growth control. *Genes Dev* 21: 2747-2761

Reviewer #2:

None

Reviewer #3:

Remarks to the Author:

In this study, the authors functionally test the effect of a number of candidate lung cancer driver genes identified in the TRACERx study. They identify 37 tumor suppressor genes associated with WGD and show for some of them that mutation of these genes leads to HR defects. They then more deeply follow up on one of these genes, FAT1, as this gene is commonly mutated in lung cancer. They find that loss of FAT1 leads to HR defects, increased replication stress, increased aneuploidy, increased numbers of micronuclei and an increased frequency of abnormal cytokinesis. As FAT1 is known to play a role in HIPPO signalling, they inactivate components in the HIPPO pathway and find that they can reproduce the FAT1 KO phenotype by alleviating key components of the HIPPO pathway (LATS1/LATS2) and conversely that they can rescue the FAT1 phenotype by concomitantly inactivating YAP1, a downstream transcription factor in the HIPPO signaling cue. They conclude that FAT1 mutations facilitate WGD events and thus cancer cell evolution in lung cancer.

Overall, the paper is well-written with minor suggestions for improvement (also see some structural suggestions below). The experimental design is ok overall: most conclusions are backed up through various approaches that largely yield data that agree.

General points of critique:

However, most described phenotypes are rather weak, which is not a critique per se, but this might make the results less suitable for the broad readership of NCB. However, this is the editor's call, not mine.

We agree with the reviewer that several of the phenotypes indeed appear to be moderate in comparison to "core" genes responsible for genome integrity, such as *BRCA1/2*, *ATM* and *ATR*.

However, we anticipate that except for hereditary breast cancer (*BRCA1/2* driver mutations), most CIN phenotypes for early cancer drivers are likely to be subtle; as we and others have shown that excessive CIN is deleterious to cancer cell survival (Bakhoum & Cantley, 2018; Jamal-Hanjani et al., 2017). This is the trade-off for accelerated cancer evolution. It is also noteworthy that *FAT1* driver mutations have been identified in normal somatic skin tissues, making it potentially one of the earliest events in carcinogenesis (Martincorena et al., 2015). Whilst the sensitivity of *FAT1* KO cells to specific genotoxic drugs might be relatively weak, its biological importance should not be ignored. Thus, we argue that it is important for the community to understand how *FAT1* loss can potentially sculpt cancer evolution through elevated CIN.

Furthermore, we would like to highlight some key phenotypes related to *FAT1* loss we have identified during the revision process:

1. *FAT1* loss resulted in a similar rate of elevated micronuclei formation compared to pharmacological inhibition of the key DDR protein ATR (**Please see response to reviewer 1 comment 3**).
2. *FAT1* loss leads to the formation of radial chromosomal structures, which are typically observed when HR genes are lost. (**Please see response to reviewer 1 comment 4**)
3. *FAT1* loss reduces HR efficiency and alternative end-joining, but not classical end-joining. Furthermore, *FAT1* mutation does not elevate end-joining related mutagenesis signatures, ID6 and SBS3 (**see response to reviewer 1 comment 1**).
4. We have identified two separate functions of *FAT1*; The co-depletion of *FAT1* and *YAP1* rescues failed cytokinesis/WGD, but does not rescue HRD deficiency and the subsequent mitotic defects (**please see response to reviewer 1 comment 6 and reviewer 3 comment 25**).
5. Hyperactivation of *YAP1* is sufficient to cause WGD (**please see response to reviewer 1 comment 6**).

Lastly, it is worth highlighting that *FAT1* is lost or mutated in approximately 10% of the lung TRACERx cancer cohort (n=421 patients), in approximately 35% of Fanconi Anaemia individuals who developed squamous cell carcinoma (n=55) (fig1C, Webster et al, Science 2022); and approximately 10% of TCGA HPV negative Head and neck squamous cell carcinomas (n=521) (fig1H, this manuscript; fig1C, Webster et al Science 2022). Furthermore, our recent paper showed that lung cancer patients with *FAT1* mutation are predisposed to worse treatment outcomes (Biswas et al., Nature Cancer, under revision). Despite its association with carcinogenesis and patient outcomes, the molecular function of *FAT1* remains elusive. We believe our manuscript will be of interest to the broad readership of Nature Cell Biology as it offers insight into the functions of *FAT1* and the impact of its malfunction in lung cancer. Given the high frequency of defects in the *FAT1*-Hippo pathway in a range of other malignancies, our present study is likely to have broader implications for the advancement of research in the fields of cell and cancer biology.

In terms of the impact on cancer evolution, we wish to highlight the significance of *FAT1* loss as a two-step driver hit with a greater potential to fuel tumorigenesis by concomitantly causing:

- i) replication stress / chromosomal instability/structural CIN; and
- ii) whole-genome-duplication/numerical CIN.

These are two key events that would otherwise occur by two distinct driver events, often separated by periods of time. Overall, the loss of *FAT1* may therefore accelerate tumor evolution more efficiently compared to some other cancer evolutionary paths, due to the deregulation of the downstream steps along the Hippo pathway with their complementary functional tumour-fuelling consequences.

References:

Bakhoun, S. F., and L. C. Cantley. 2018. 'The Multifaceted Role of Chromosomal Instability in Cancer and Its Microenvironment', *Cell*, 174: 1347-60.

Jamal-Hanjani M, Wilson GA, McGranahan N, Birkbak NJ, Watkins TBK, Veeriah S, Shafi S, Johnson DH, Mitter R, Rosenthal R et al (2017) Tracking the Evolution of Non-Small-Cell Lung Cancer. *N Engl J Med* 376: 2109-2121

Martincorena I, Roshan A, Gerstung M, Ellis P, Van Loo P, McLaren S, Wedge DC, Fullam A, Alexandrov LB, Tubio JM et al (2015) Tumor evolution. High burden and pervasive positive selection of somatic mutations in normal human skin. *Science* 348: 880-886

Webster ALH, Sanders MA, Patel K, Dietrich R, Noonan RJ, Lach FP, White RR, Goldfarb A, Hadi K, Edwards MM et al (2022) Genomic signature of Fanconi anaemia DNA repair pathway deficiency in cancer. *Nature* 612: 495-502

An important point of critique that I do need to make is that the authors use various cell lines throughout the paper for different types of experiments. In some cases, this is well-justified, in other cases less so. To ensure physiological relevance of all findings, there should be overlap between the cell lines being used for the different assays and per assay type 2-3 cell lines should be tested at least. Also, the choice of ‘general’ cell lines should be well-justified with 2 cell lines at least being used for all experiments.

Whenever possible, we attempted to utilise lung cancer cell lines because *FAT1* is a key driver in lung cancer. We apologise if the narration in our manuscript appears to give the impression that we were switching between multiple cell lines— in fact, we strived to achieve quite the opposite. We have produced a table highlighting the multiple cell lines and orthogonal methods used to validate each of the take-home messages of our manuscript (**please see pages 22-23 of the rebuttal**).

The U2OS model was used in our study since it is the most widely used cancer cell line in the DNA damage response (DDR)/genome integrity field, therefore data obtained with this model allow the broadest possible comparisons with the state-of-the-art, highlighting the new advances provided by our present results. Furthermore, the U2OS cells have been modified by many laboratories as a vehicle for diverse reporters and molecular assays, some of which we also employed here.

Many landmark papers employed the U2OS cells to pioneer the existence and role of replication stress in cancer development such as (Bartkova *et al.*, 2005; Bartkova *et al.*, 2006; Burrell *et al.*, 2013). This includes the demonstration of micronuclei formation in response to replication stress (Joo *et al.*, 2023; Lukas *et al.*, 2011; Spies *et al.*, 2019). For a similar reason, we used T2P, RPE1 cells and the FUCCI derivative of RPE1 in this manuscript since they are important tools used in the mitosis field due to their non-transformed nature (Bakhoun *et al.*, 2014; Venkatesan *et al.*, 2021; Zeng *et al.*, 2023). The RPE1-FUCCI-H2B Turquoise line was necessary to visualize cell division in real-time. It should be noted that the use of reporter cell lines such as RPE-FUCCI and U2OS-DNA repair reporter cell lines are essential tools in cell cycle and genome instability fields respectively. In both lines of investigations above we attempted to further validate our results with lung cancer cell lines.

PC9 cell line was used because it provides yet another lung cancer cell line as orthogonal validation. The use of PC9 also enables us to follow up on recent investigations proposing that elevated chromosomal instability can accelerate cancer evolution and enable tumor cells to pursue distinct evolutionary routes to become resistant to targeted therapy (Caswell *et al.*, 2024; Dharanipragada *et al.*, 2023; Lukow *et al.*, 2021; Pfeifer *et al.*, 2024; Sebastijan Hobor, 2024, In press. DOI:10.1038/s41467-024-47606-9). PC9 harbours an EGFR^{T790M}-activating mutation, therefore it is sensitive to EGFR inhibitors such as Osimertinib or Erlotinib, thus enabling us to study targeted therapy resistance.

We believe the inclusion of these widely used cell line models allows documentation of the broader applicability of the mechanisms we report, and it will likely inspire many other researchers to follow up on our study, well beyond the scope of lung cancer.

References:

Bakhoun SF, Kabeche L, Murnane JP, Zaki BI, Compton DA (2014) DNA-damage response during mitosis induces whole-chromosome missegregation. *Cancer Discov* 4: 1281-1289

Bartkova J, Horejsi Z, Koed K, Kramer A, Tort F, Zieger K, Guldborg P, Sehested M, Nesland JM, Lukas C *et al* (2005) DNA damage response as a candidate anti-cancer barrier in early human tumorigenesis. *Nature* 434: 864-870

Bartkova J, Rezaei N, Liontos M, Karakaidos P, Kletsas D, Issaeva N, Vassiliou LV, Kolettas E, Niforou K, Zoumpourlis VC *et al* (2006) Oncogene-induced senescence is part of the tumorigenesis barrier imposed by DNA damage checkpoints. *Nature* 444: 633-637

Burrell RA, McClelland SE, Endesfelder D, Groth P, Weller MC, Shaikh N, Domingo E, Kanu N, Dewhurst SM, Gronroos E et al (2013) Replication stress links structural and numerical cancer chromosomal instability. *Nature* 494: 492-496

Caswell DR, Gui P, Mayekar MK, Law EK, Pich O, Bailey C, Boumelha J, Kerr DL, Blakely CM, Manabe T et al (2024) The role of APOBEC3B in lung tumor evolution and targeted cancer therapy resistance. *Nat Genet* 56: 60-73

Dharanipragada P, Zhang X, Liu S, Lomeli SH, Hong A, Wang Y, Yang Z, Lo KZ, Vega-Crespo A, Ribas A et al (2023) Blocking Genomic Instability Prevents Acquired Resistance to MAPK Inhibitor Therapy in Melanoma. *Cancer Discov* 13: 880-909

Joo YK, Black EM, Trier I, Haakma W, Zou L, Kabeche L (2023) ATR promotes clearance of damaged DNA and damaged cells by rupturing micronuclei. *Mol Cell* 83: 3642-3658 e3644

Lukas C, Savic V, Bekker-Jensen S, Doil C, Neumann B, Pedersen RS, Grofte M, Chan KL, Hickson ID, Bartek J et al (2011) 53BP1 nuclear bodies form around DNA lesions generated by mitotic transmission of chromosomes under replication stress. *Nat Cell Biol* 13: 243-253

Lukow DA, Sausville EL, Suri P, Chunduri NK, Wieland A, Leu J, Smith JC, Girish V, Kumar AA, Kendall J et al (2021) Chromosomal instability accelerates the evolution of resistance to anti-cancer therapies. *Dev Cell* 56: 2427-2439 e2424

Pfeifer M, Brummel JS, Price S, Pilling J, Bhavsar D, Farcas A, Bateson J, Sundarajan A, Miragaia RJ, Guan N et al (2024) Genome-wide CRISPR screens identify the YAP/TEAD axis as a driver of persister cells in EGFR mutant lung cancer. *Commun Biol* 7: 497

Spies J, Lukas C, Somyajit K, Rask MB, Lukas J, Neelsen KJ (2019) 53BP1 nuclear bodies enforce replication timing at under-replicated DNA to limit heritable DNA damage. *Nat Cell Biol* 21: 487-497

Sebastijan Hobor MAB, Crispin T. Hiley, Marcin Skrzypsk, Alexander M. Frankell, Bjorn Bakker, Thomas B. K. Watkins, Aleksandra Markovets, Jonathan R Dry, Nnennaya Kanu, Simone Zaccaria, Eva Grönroos and Charles Swanton (2024, In press. DOI:10.1038/s41467-024-47606-9) Mixed responses to targeted therapy driven by chromosomal instability through p53 dysfunction and genome doubling. *Nature communication* In press

Venkatesan S, Angelova M, Puttick C, Zhai H, Caswell DR, Lu WT, Dietzen M, Galanos P, Evangelou K, Bellelli R et al (2021) Induction of APOBEC3 Exacerbates DNA Replication Stress and Chromosomal Instability in Early Breast and Lung Cancer Evolution. *Cancer Discov* 11: 2456-2473

Zeng J, Hills SA, Ozono E, Diffley JFX (2023) Cyclin E-induced replicative stress drives p53-dependent whole-genome duplication. *Cell*

Also, different methods to alleviate FAT1 are being used, sometimes siRNA, sometimes CRISPR KO. It is not completely clear why this is, and while it is not necessary to redo all experiments for both siRNA and CRISPR knockout, it would strengthen the data if key experiments of the paper are done with both siRNA and CRISPR KO approaches including all necessary replicates (multiple knockout lines per gene, i.e independently engineered KO cell lines, ideally from different gRNAs and/or siRNAs).

We use RNAi to investigate the acute effect of FAT1 depletion, especially for experiments involving HRD and mitotic errors and CRISPR KO was used to mimic long-term loss in cancer. As detailed in the material and methods section, multiple siRNA, CRISPR guides, and KO clones were used to make the same point.

Again, we apologize if our cell line/methodology usage appears to be inconsistent. To dispel further misunderstandings, we have generated a table (**please see pages 22-23**) that summarises each take-home message in this manuscript. It should be clear that we aimed to use multiple cell lines and orthogonal methods for each of our take-home messages.

Table 1: A list of the key take-home messages of this manuscript vs cell line/methodology usage.

key take home results	readout	siRNA/sgRNA	cell line, source	figure	remarks
1. FAT1 loss resulting in HR loss post DNA DSB formation	HR reporter	siRNA (Ambion)	U2OS, bone	1E	
	NHEJ reporter	siRNA (Ambion)	U2OS, bone	S12A	
	Alt-EJ reporter	siRNA (Ambion)	U2OS, bone	S12B	
	ssDNA resection	siRNA (Ambion)	U2OS- AsiSI, bone	1F	used because this is the main system to assay ssDNA formation
	RAD51	siRNA(Ambion)	A549, lung, near triploid	1G,S4AB	
		siRNA (dharmacon)	H1944, lung, near diploid	S5A	alternate sirna used to eliminate possibility of siRNA off-target effect
		sgRNA, 2 guides	H1944, lung, near diploid	S9A	2 guides transfected together, polyclonal
	BRCA1	siRNA(Ambion)	A549, lung, near triploid	3A,S8D	individual guides, single cell cloning, with rescue experiments.
	CtIP	siRNA(Ambion)	A549, lung, near triploid	3A,S8C	
	2. FAT1 loss resulting in elevated 53BP1 bodies and micronuclei	53bp1 bodies HU pulse + recovery	siRNA(Ambion)	U2OS, bone	3J
53bp1 bodies, low dose aphidicolin pulse		siRNA(Ambion)	T2P, lung, untransformed	S13A	T2P chosen due to known ability to form 53bp1 bodies (Venkatesan Cancer Discovery 2021)
micronuclei HU pulse + recovery (also with CREST staining)		siRNA(Ambion)	U2OS, bone	4A	HU pulse and low dose aphidicolin are two known way to induce 53bp1 bodies and micronuclei (Feng and Jasin, Nat Comms 2017, Lucas et al NCB 2011)
		siRNA(Ambion)	U2OS, bone	4B	
micronuclei, low dose aphidicolin (also with CREST staining)		siRNA(Ambion)	T2P, lung, untransformed	S13B	* Difficult to generate stable KO in untransformed cells.
sgRNA, 2 guides		A549, lung, near triploid	S13C		
3. FAT1 loss resulting elevated replicative fork defect	DNA fibre assay, HU interphase FANCD2 foci, low dose aphidicolin	sgRNA, 1 guide	A549, lung, near triploid	3G	
		2 sgRNA, mixed guide	PC9-genome doubled, Lung	S14C	
		siRNA (Ambion)	PC9-genome doubled, Lung	S14D	
4. FAT1 loss resulting in elevated CIN	mitotic errors, HU	siRNA (Ambion)	U2OS, bone	4C	
	mitotic errors, aphidicolin pulse	siRNA (Ambion)	PC9, lung	S14A	*PC9 selected to probe the effect of FAT1 before and after WGD.
		siRNA (Ambion)	PC9, genome doubled, lung	S14A	
		sgRNA, 2guides	A549 lung, near triploid	7DE	
	Live cell imaging, Mitotic chromosomal bridges	siRNA (Ambion)	RPE1-FUCCI, retina untransformed	S16C	chosen due to H2B-mTurquoise system
	Metaphase spread, aphidicolin pulse	sgRNA, 2guides	A549 lung, near triploid	4E, S13F	
	metaphase spread	siRNA (Ambion)	H1944, lung, near diploid	4H, S13G	
	clonal FISH, traditional or FACS based	siRNA (Ambion)	H1944, lung, near diploid	4F,4G	
	translocation assay	siRNA (Ambion)	U2OS- AsiSI, bone	4D	selected because we know specific break site to probe for translocations by PCR
	5. FAT1 loss	EdU incorporation	sgRNA, mixed	PC9, lung	5AB

resulting in WGD through mitotic errors		guide, 2 clones			
		2 sgRNA, mixed guide	PC9, lung	S15B	
	MCM7 reloading	siRNA (Ambion)	RPE1, retina, untransformed	S15C	
	multinucleation immune-fluorescence HU	siRNA (Ambion)	U2OS	S17A	
	multinucleation immune-fluorescence aphidicolin	siRNA (Ambion)	RPE1, retina TP53 WT, untransformed	S17B	
			RPE1, TP53 KO retina, untransformed	S17B	
	live cell imaging	2 siRNA (Ambion)	RPE1-FUCCI, retina untransformed	5DE, S16BC, movies	chosen due to FUCCI-mTurquoise system
6. FAT1 loss leads of hippo pathway dysregulation. YAP1 hyperactivation is required for WGD but not HRD phenotypes	YAP1 nuclear localization (PFA or PTEMF fixation)	siRNA (Ambion)	RPE1-FUCCI, retina, untransformed T2P, lung, untransformed	6B S21A	NB, A549 and PC9 seems to have high basal YAP/TAZ so they are not suitable for YAP1 antibody staining
	TEAD reporter assay (YAP1 effector)	2 sgRNA, mixed guide	PC9 hexaploid, lung	6C, S22AB	PC9 Hexaploid because it contains multiple copies of YAP/TAZ to highlight the role of FAT1 in controlling YAP/TEAD transcription. Also completed with rescue experiments to elucidate mechanisms
	YAP1 + FAT1 double KD, 6Gy IR, RAD51	siRNA (Ambion)+ sgRNA	A549, lung, near triploid	7C	
	YAP1 + FAT1 double KD, aphidicolin pulse, mitotic error	siRNA (Ambion)+ sgRNA	A549, lung, near triploid	7DE, S24C	
	YAP1 + FAT1 double KD, HR reporter assay	siRNA (Ambion)	A549, lung, near triploid	7A	
	YAP1 + FAT1 double KD, ssDNA resection assay	siRNA (Ambion)	U2OS- AsiSi, bone	7B	
	YAP1+ FAT1 double KD, live cell imaging fail cytokinesis	siRNA (Ambion)	RPE1-FUCCI, retina, untransformed	7F	
	YAP15SA overexpression, EdU incorporation assay to visualize WGD population	siRNA (Ambion)+ sgRNA	PC9, lung, near triploid	7H	
		siRNA (Ambion)+ sgRNA	RPE1 TP53 WT/KO, untransformed	7G	RPE1 was used in Ganem et. al 2014 to test the role of YAP1 to sustain replication of 4N RPE1 cells

Regarding the quality of the imaging data (also see some specific points below), particularly the IF images, better quality images need to be provided for most panels. This can be done by providing zoomed in (higher res) images to simplify interpretation such that the images really support the

subsequent quantification.

We sincerely apologise for the low-quality image data – somehow the image quality was reduced during pdf uploading. As suggested by the reviewer in comment 8, we have now added zoomed-in insets for the immunofluorescence figures. Additionally, in case the same error occurs during pdf uploading, now we have also provided download links for a high-resolution pdf for our manuscript.

Finally, while the authors suggest a broader mechanism, the exact mechanism of the phenotype remains unresolved: the authors suggest that loss of FAT1 via loss of YAP1 nuclear localization leads to replication stress (and an HR defect), that leads to increased numbers of lagging chromosomes, leading to more micronuclei and increased cytokinesis defects. However, how exactly YAP1 mislocalization leads to replication stress is not clear, nor whether FAT1 also influences other pathways that might also lead to replication stress and downstream genomic instability.

We thank the reviewer for their comment. Since some of the phenotypes mentioned by the reviewer are *FAT1*-loss-dependent yet *YAP1*-independent (see also our revised Model, Fig S26) for a concise summary of the two sets of phenotypes observed upon *FAT1* loss, and the underlying causes attributed to distinct altered steps along the Hippo pathway. Thanks to the suggestions from both reviewers, we have performed a number of new experiments during the revision, we believe we now have a better idea of how *FAT1* loss can contribute to HRD and the mitotic defect phenotypes we observed. Here we list a summary of new observations:

1. *FAT1* loss leads to the formation of radial chromosomal structures and increases the incidence of chromosomal aberrations per metaphase, which are typically observed when HR genes are lost. **(Please see our response to reviewer 1 comment 4)**
2. *FAT1* loss reduces HR efficiency and alternative end-joining, but not the rate of classical end-joining **(Please refer to our response to reviewer 1 comment 1)**. This further enforced our observation that *FAT1* loss is detrimental to DSB end-resection.
3. Thanks to the insightful suggestions from both reviewers, we can now propose that a separation-of-functions appears to exist downstream of FAT1. The co-depletion of FAT1 and YAP1 rescues the cytokinesis failure/WGD but does not rescue HRD deficiency and the associated mitotic defects. This is observed using orthogonal experimental methods. **(please see our response to reviewer 1 comment 6 and reviewer 3 comment 25)**.
4. To further investigate potential separation-of-function downstream of FAT1, we used a constitutively active mScarlet-YAP1^{5SA} construct, enabling assessment of the WGD population through assaying EdU incorporation beyond the G2 cell cycle phase and monitoring the contribution of YAP1 hyperactivation towards WGD. We observed a significant and consistent increase in the WGD population in both isogenic *TP53* WT/KO RPE1 cells and *FAT1* WT/KO PC9 cells (see FigR1C6, panel GH), suggesting that constitutively active YAP1 is sufficient to promote WGD. This is in line with the observation made previously by Ganem et al (Ganem *et al*, 2014) where YAP1^{5SA} sustained replication in 4N RPE1 cells. This confirmed our hypothesis above that *FAT1* loss introduces two distinct phenotypes: HRD and structural CIN, which are YAP1 independent; and the induction of WGD, which is dependent on YAP1 dysregulation **(please see our response to reviewer 1 comment 6 for detail)**.
5. We have constructed novel fluorescence reporters and *FAT1* mutant constructs to assess TEAD transcriptional activity downstream of YAP1/TAZ. Using this construct, we have confirmed *FAT1* loss not only affects YAP1 nuclear localization but also attenuates the transcriptional activity of TEAD1-4

downstream of both YAP1 and TAZ. This is particularly important since YAP1 and TAZ are functionally redundant and are both frequently amplified in cancer (Fig 1C, Webster *et al*, Science 2022; FigS16, this manuscript). Our results suggest that *FAT1* loss is sufficient to alter the transcriptional activity of TEAD1-4 regardless of TAZ level (**please refer to our response to reviewer 3 comment 30 for details**).

6. Using our new TEAD reporter we performed rescue experiments to further functionally dissect the domains of *FAT1* that are required for proficient TEAD transcription (**please see figure R3C1 below**).

We have constructed novel neon green fluorescence reporters with 14x TEAD binding sites as minimal promoter regions (please refer to FigR3C1, panel A for cartoon detailing reporter and rescue constructs). This approach allows the delineation of TEAD transcriptional activity downstream of YAP1/TAZ with the single-cell resolution. This is an improvement compared to the existing TEAD luciferase reporter (Dupont *et al*, 2011) since we no longer need to rely on bulk luciferase assay quantification. Our immunofluorescence approach enables the identification of baseline TEAD activity by quantifying cells where successful TEAD reporter co-transfection has occurred by co-expression of an HA-tagged eIF4A1 construct, a highly expressed helicase in cancer cells (Meijer *et al*, 2013). In line with our previous results in Fig6B and FigS21A, we observed an increase in TEAD transcriptional activity in *FAT1* KO cells (Fig R3C1 panel B condition 1 vs condition 2).

TEAD transcriptional activity can be further de-repressed by overexpressing a constitutively active YAP1 mutant construct devoid of 5 LATS phosphorylation sites (HA-YAP1^{55A}, Fig R3C1 below, panel B, condition 1/2 vs condition 3). This observation confirms our analyses are well within the dynamic range of the assay.

With the system established, we performed rescue experiments attempting to reverse the TEAD transcriptional repression by re-expressing *FAT1* WT and a mutant construct. We next investigated which domains of the *FAT1* protein (total length 4588AA) are required to re-suppress TEAD transcriptional activity. To this end, in addition to the minimum C-terminal HA-*FAT1*^{ICD} fragment capable of rescuing RAD51 IRIF formation (Fig3BC), we also generated a *FAT1* construct encompassing the extracellular and transmembrane domain (mScarlet-HA-*FAT1*^{WT} in FigR3C1 panel A, consisting of 3735-4588AA of *FAT1*). In response to a recent report demonstrating that *FAT1* cooperates with the E3 ligase MIB2 to regulate YAP/TAZ signalling (Li *et al*, 2023), we also generated a *FAT1* mutant construct devoid of the MIB2 interaction domain (mScarlet-HA-*FAT1*^{MIBΔ} in FigR3C1 panel A). Consistent with our hypothesis, the mS-HA-*FAT1*^{WT} fragment successfully re-repressed TEAD transcription, however, the *FAT1* construct lacking the MIB2 interaction domain failed to do so (FigR3C1 panel B conditions 4 and 5, Fig S21A), suggesting that the interaction with the E3 ligase MIB2 is crucial for TEAD transcription modulation by *FAT1*. We further confirmed that our observation is not due to the localization of the HA-tag in our constructs (FigR3C1 panel C).

Surprisingly, the HA-*FAT1*^{ICD} construct, which lacks a transmembrane domain anchor, failed to re-repress TEAD transcription (FigR3C1 panel A, condition6), despite successfully rescuing IR-induced RAD51 foci formation in A549 cells (Fig3BC). Given the possible separation-of-function downstream of *FAT1* (See response to R1 comment 6 for details), we hypothesise that different domains of *FAT1* may be involved in distinct biological pathways downstream of *FAT1* along the Hippo pathway. Indeed, we observed distinct subcellular localizations of the *FAT1* constructs (Fig R3C1 below, panel D). We observed that the HA-*FAT1*^{ICD} fragment (4202-4588AA) localises to both the nucleus and cytoplasm. In contrast, a longer fragment containing a transmembrane domain (HA-*FAT1*^{WT}, 3735-4588AA) mainly displays cytoplasmic localization. Previous literature has reported that *FAT1* can be cleaved by furin and ADAM10, followed by the processing and internalisation of its C-terminal domain (Sadeqzadeh *et al*, 2011; Wojtalewicz *et al*, 2014). While it is beyond the scope of this manuscript, it is enticing to

hypothesise that distinct subcellular localizations of processed/cleaved FAT1 might introduce different binding partners and activate different downstream signalling pathways.

Taken together, we believe we have delineated the functional domains of FAT1 protein that are required to perform different functions which we believe will be of broad interest to the readership of Nature Cell Biology.

FigR3C1, A, Cartoon illustrating the predicted domains of the FAT1 protein, and respective regions cloned into a pCMV expression plasmid with an HA epitope tag. LAMG, Laminin-G like domain; TM, transmembrane domain;

MIB2, MindBomb2 interacting domain; NLS, nuclear localisation signal. TEAD activity in FAT1 KO PC9 WGD cells. Normalized TEAD activity is measured as an enrichment of the neonGreen signal over the untransfected background signal in each experiment. The HA signal is used to identify successful FAT1 rescue construct cotransfection at a single cell level. **B**, TEAD activity is elevated in FAT1 KO cells but can be further increased by overexpressing the constitutively active HA-YAP1^{55A} mutant. Overexpression of the FAT1 WT construct represses TEAD activity. However, overexpression of FAT1 mutants devoid of MIB2 binding region (mS-HA-FAT1^{MIBΔ}) and HA-FAT1^{ICD} does not repress TEAD activity. Dunn's Test, **p<0.01, ***p<0.001. **C**, HA-epitope tag was located either at the N-terminal or internally within the FAT1 construct. The localization of the HA-epitope tag does not affect the ability to repress TEAD reporter transcription. One-way ANOVA, for each condition >95 cells were scored across 3 biological repeats, **p<0.01, ***p<0.001. **D**, Representative immunofluorescence images illustrating the overexpression and subcellular localization of the HA-FAT1^{WT} construct versus the HA-FAT1^{ICD} construct.

We are very grateful for the insightful suggestions of both reviewers; we believe the revised manuscript is significantly improved and provides additional mechanistic insight into the FAT1-HIPPO pathway.

To reflect these new results, we have incorporated these data into Fig6C and S21AB.

In addition we have added a section starting on page7 line25 which now reads:

"FAT1 depletion causes dysregulation of the Hippo signalling pathway"

To determine the molecular mechanism by which FAT1 alterations lead to CIN, we investigated components of potential signalling pathways in which FAT1 has been associated. FAT1 loss has been implicated in the dysregulation of Hippo signalling, RAS-RAF-MEK-MAPK signalling and regulating the level of the YES tyrosine kinase, which mediates hybrid-epithelial-mesenchymal-transition⁴⁸⁻⁵⁰. Under our experimental conditions, we did not consistently observe FAT1 siRNA alters MEK-ERK phosphorylation or YES tyrosine kinase levels in two cell lines (Fig6A). However, upon FAT1 depletion, elevated nuclear localization of the key Hippo pathway activator YAP/TAZ was observed (Fig6B, S21A). In T2P and RPE1-FUCCI cells, despite marked alterations in YAP1 nuclear localization, no significant alteration in YAP1 phosphorylation could be consistently observed following LATS1 or FAT1 depletion (FigS21B).

As an orthogonal approach, we designed a neonGreen reporter with 14xTEAD binding sites to assess the ability of FAT1 to regulate the ultimate output of the Hippo pathway, TEAD transcription. This is particularly important because YAP1 and TAZ/WWTR1 are highly analogous and functionally overlapping⁵¹, and both are amplified in lung cancer^{3, 52} (Fig S21C). Using an HA-tag to control for cotransfection efficiency, we confirmed that FAT1 loss elevated TEAD transcriptional activity, despite the presence of multiple copies of YAP/TAZ in hexaploid WGD PC9 cells (Fig6C, conditions 1 vs 2). Notably, TEAD activity could be further stimulated in FAT1 KO cells by overexpressing the constitutively active YAP1 mutant construct devoid of 5 LATS phosphorylation sites⁵³ (HA-YAP1^{55A}, Fig6C condition 3).

We next investigated which domains of FAT1 (4588 amino acids in total) are required to re-suppress TEAD transcriptional activity. To this end, in addition to the minimum C-terminal HA-FAT1^{ICD} fragment capable of rescuing RAD51 IRIF formation (Fig3BC), we also generated a FAT1 construct encompassing the extracellular and transmembrane domain (mS-HA-FAT1^{WT} in Fig6C, consisting of 3735-4588 amino acids of FAT1). In response to a recent report demonstrating that FAT1 cooperates with the E3 ligase MIB2 to regulate YAP/TAZ signalling⁵⁴, we generated a FAT1 mutant construct devoid of the MIB2 interaction domain (mS-HA-FAT1^{MIBΔ} in Fig6C). Consistent with our hypothesis, the mS-HA-FAT1^{WT} fragment successfully re-repressed TEAD transcription, unlike the FAT1 construct lacking the MIB2 interaction domain (Fig6C conditions 4 and 5, FigS22A), suggesting the interaction with the E3 ligase MIB2 is crucial for TEAD transcription modulation by FAT1.

To our surprise, the HA-FAT1^{ICD} construct, which lacks a transmembrane domain anchor in comparison to HA-FAT1^{WT}, failed to re-repress TEAD transcription (Fig6C, condition6), despite successfully rescuing IR-induced RAD51 foci formation in A549 cells (Fig 3BC, FigS22B)."

References:

Dupont S, Morsut L, Aragona M, Enzo E, Giulitti S, Cordenonsi M, Zanconato F, Le Digabel J, Forcato M, Bicciato S et al (2011) Role of YAP/TAZ in mechanotransduction. Nature 474: 179-183

Ganem NJ, Cornils H, Chiu SY, O'Rourke KP, Arnaud J, Yimlamai D, Thery M, Camargo FD, Pellman D (2014) Cytokinesis failure triggers hippo tumor suppressor pathway activation. Cell 158: 833-848

Li R, Shao J, Jin YJ, Kawase H, Ong YT, Troidl K, Quan Q, Wang L, Bonnavion R, Wietelmann A et al (2023) Endothelial FAT1 inhibits angiogenesis by controlling YAP/TAZ protein degradation via E3 ligase MIB2. Nat Commun 14: 1980

Meijer HA, Kong YW, Lu WT, Wilczynska A, Spriggs RV, Robinson SW, Godfrey JD, Willis AE, Bushell M (2013) Translational repression and eIF4A2 activity are critical for microRNA-mediated gene regulation. Science 340: 82-85

Sadeqzadeh E, de Bock CE, Zhang XD, Shipman KL, Scott NM, Song C, Yeadon T, Oliveira CS, Jin B, Hersey P et al (2011) Dual processing of FAT1 cadherin protein by human melanoma cells generates distinct protein products. J Biol Chem 286: 28181-28191

Webster ALH, Sanders MA, Patel K, Dietrich R, Noonan RJ, Lach FP, White RR, Goldfarb A, Hadi K, Edwards MM et al (2022) Genomic signature of Fanconi anaemia DNA repair pathway deficiency in cancer. Nature 612: 495-502

Wojtalowicz N, Sadeqzadeh E, Weiss JV, Tehrani MM, Klein-Scory S, Hahn S, Schmiegel W, Warnken U, Schnolzer M, de Bock CE et al (2014) A soluble form of the giant cadherin Fat1 is released from pancreatic cancer cells by ADAM10 mediated ectodomain shedding. PLoS One 9: e90461

More specifically, a number of smaller and larger issues (presented in order of the manuscript) that the authors might want to address to improve the manuscript is copied below.

1. Page 2 line 7: "associated with, the irreversible accumulation". Comma should be removed: "associated with the irreversible accumulation".

We have altered the text as advised. We thank the reviewer for pointing this out.

2. Page 3, Fig S2: It is unclear what was analysed here and what the figure represents. The legend is very brief. This also makes it difficult to interpret the data towards the conclusion in line 7 that these genes are required to maintain chromosomal stability.

We are sorry for our lack of clarity. We have the legend in the revised manuscript.

The figure legends now reads: "*The HAC assay can assess the impact of gene depletion and mild replication stress on CIN, through modelling chromosome loss using an HT-1080 reporter system. Cells were processed and data analysed as described in Methods. Data are displayed as scatter plots of log fold changes (LFCs) and Strictly Standardised Mean Difference (SSMDs), highlighting negative and positive control genes, and genes that passed the hit threshold (green area)."*

3. Related to this point: it is not fully clear whether HU is required to see loss of the HAC when the target genes are knocked out. This is important as a required predisposition (like HU treatment) to observe a CIN phenotype could also imply that the targeted gene helps cells tolerate CIN (like loss of p53 does), but does not drive CIN per se. This should be explained better especially as the authors continue to show only for FAT1 that loss leads to increased CIN rates.

We apologise for the confusion. Indeed, FAT1 ablation scored in the HAC assay only after the HU challenge only.

We do agree with the point raised by the reviewer. However, whilst the HAC assay was a very useful tool for the initial screen, it is based on an *artificial, relatively small extrachromosomal element* that may not fully reflect the segregation efficiency of native chromosomes. It was merely used as an initial screen tool which formed a good starting point to identify initial gene “hits”, but the extent to which actual replication and CIN phenotypes are affected by these hits requires further validation in other cell lines. To this end, we performed multiple experiments (see Table 1, pages 22-23) highlighting the role of FAT1 in mitotic segregation in multiple cell lines, showing that *FAT1* loss can directly lead to increased CIN rates and may possess an additional role in tolerating CIN.

4. In general, the authors use the term CIN quite generically. While this might be a semantic issue, CIN is most commonly used to describe an increased rate of mitotic abnormalities. Furthermore, HR defects can lead to structural CIN while WGD is a form of numerical CIN. To increase readability, it would be best if authors more explicitly separate these types of CIN and also separate the upstream replication stress and HR defects from the resulting CIN as the latter occurs in mitosis, while the former takes place during S-phase. This is also relevant for the conclusion, as in my view loss of FAT1 leads to replication/HGR problems that then promote CIN as a secondary effect (also see last points below).

We thank the reviewer for the insightful comment. We initially adopted a broad definition of CIN and we regret that this decreased the readability of the manuscript.

We have now modified the text accordingly on the occasions where we can specify numerical or structural CIN, please see below:

Page 4 line 45 now reads *“FAT1 ablation leads to structural and numerical CIN”*

Page 5 lines 2-5 now reads *“Indeed, our analysis of the TCGA and Genomics England datasets revealed that FAT1 loss correlated with an increased Weighted Genome Instability Index (wGII) and total indel burden respectively, indicating an elevation of structural and numerical CIN (Fig3HI)”*

Page 5 sentence starting on line 31 now reads *“In the isogenic WGD PC9 cells, a reduction in interphase and mitotic FANCD2 foci formation was observed in FAT1-depleted cells, despite an increase in the rate of replication fork collapse, suggesting an inability to process replication stress, resulting in elevated structural CIN (FigS13ACDEF).”*

Page7 sentence starting on line 5 now reads *“Furthermore, live-cell imaging also revealed that FAT1-depleted cells exhibited an accelerated rate of entry into metaphase (FigS19), suggestive of a potential role for FAT1 in regulating mitotic timing. Taken together, these results demonstrate that FAT1 loss not only increases the rate of structural CIN but also exacerbates the mitotic defects in daughter cells following CIN, contributing to WGD.”*

Page 9 sentence starting line 11 now reads *“Based on these observations, conceivably FAT1 possesses a dual role. One outcome of FAT1 loss results in HRD and leads to unresolved recombination intermediates, replication stress, structural CIN and nuclear deformation (Fig3,4,7A-D) – these events appear to be YAP1 independent.”*

We agree with the reviewer that the HRD/replication defect phenotype was linked to its structural CIN phenotype. During the revision process, we set out to further dissect the relative contribution of FAT1 to structural and numerical CIN respectively. In response to the comments from both reviewers (R1

comment 6 and R3 comment 25) regarding the potential separation-of-function between FAT1 and TEAD/YAP1 dysregulation. Interestingly, we discovered that the relationship between the numerical CIN/WGD phenotype versus structural CIN/HRD phenotypes generated by *FAT1* loss may not be as direct as we originally anticipated (please see response to R1 comment 6 for a detailed description of experiments).

In short, we observed that co-depletion of *FAT1* and *YAP1* does not rescue HR deficiency phenotypes nor the mitotic errors attributed to *FAT1*, hence still contributing to structural CIN (Panel A-E, Fig R1C6). This is distinct from the role played by *FAT1* loss in the generation of a WGD population via cytokinesis failure, which constitutes numerical CIN and is dependent on dysregulated *YAP1* signalling (Panel F-H, Fig R1C6, please see the response to Reviewer 1 comment 6 for detail).

We have highlighted this finding in a revised section, starting on page 8 line 37:

“Co-depletion of *FAT1* and *YAP1* reverses WGD but not HR deficiency

*Given the involvement of *FAT1* in the Hippo signalling pathway (Fig6BC), we proceeded to systematically delineate whether both the WGD and CIN associated with *FAT1* loss might be reversed by co-depleting *YAP1*. We first investigated whether the HR deficiency might be reversed by *FAT1*/*YAP1* co-depletion. Surprisingly, despite *FAT1*/*YAP1* co-depletion reversing the cell-cycle arrest associated with *YAP1* single knockdown (FigS24AB), the co-depletion of *FAT1*/*YAP1* did not rescue HR activation defects after DSB formation (Fig7A-C). This result was observed using orthogonal methods including the DR-GFP reporter assay, ssDNA formation after endonuclease-induced DSBs, and *RAD51* foci formation after IR (Fig7A-C). Next, since unresolved recombination intermediates due to HR repair deficiency can cause mitotic errors (Chan et al., 2018; Daniels et al., 2004), we next quantified the rate of mitotic errors. *YAP1*/*FAT1* co-depletion failed to rescue the elevated mitotic error and chromosomal bridges observed in *FAT1* KO A549 cells (Fig7DE, FigS24C).*

*We next investigated whether *YAP1* co-depletion might rescue the CIN phenotypes associated with *FAT1* loss. Live-cell imaging experiments demonstrated that co-depletion of *FAT1* and *YAP1* reversed the cytokinesis failure phenotypes (Fig7F left, FigS24D). However, the nuclear envelope deformation observed following *FAT1* depletion was only partially rescued (Fig7F right).*

*Based on these observations, conceivably *FAT1* possesses a dual role. One outcome of *FAT1* loss is HRD and leads to unresolved recombination intermediates and nuclear deformation (Fig3,4,7A-D) – these events appear to be *YAP1* independent. In contrast, the elevated rate of cytokinesis failure and subsequent WGD appear to be dependent on *YAP1* activity (Fig7F). To test this hypothesis, we performed an EdU incorporation assay (Fig5AB) to visualize the impact of constitutively active *YAP1*^{55A} overexpression on WGD. Indeed, upon 48 hours of *YAP1*^{55A} overexpression, we observed an emerging WGD population in both *FAT1* WT and *FAT1* KO A549 cells, suggesting that constitutively active *YAP1* is sufficient to promote WGD (Fig7G). Taken together, these results suggest that *FAT1* might drive genome instability through two different routes; one through WGD, dependent on *YAP1*, and the second through HR deficiency, driving replication stress and structural CIN, in a *TEAD*/*YAP1*-independent manner.”*

We thank the reviewer for their suggestion which encouraged us to better define the mechanisms by which *FAT1* ablation contributes to structural and numerical CIN respectively.

5. Page 3, line 6 & Fig 1C: the effects of *FAT1* on DNA damage foci loss are most obvious following HU treatment and less so following irradiation. Why is this? Maybe because loss of *FAT1* leads to replication stress and HU synergizes with this? Also, this seems to be different for most of the other

candidate genes. Maybe these play a role in the DNA damage response? The authors might want to comment on this in the text/discussion.

We agree with all the statements above. We selected the 6 candidate genes in Fig1D-G due to their overlap in the HAC screen and successfully validated that 5 out of these genes impact the DNA damage response (specifically HR). Indeed, as suggested by the reviewer many target genes highlighted in the screen may play a role in the DNA damage response, but we specifically chose to study *FAT1* because it is highly mutated in cancer, and its involvement in the DNA damage response had not been reported previously.

While it's beyond the scope of this manuscript to validate all the hits in Fig1C, we agree with the reviewer that it is enticing to assume these play a role in the DDR. Apart from the 6 genes we selected, some of the candidates in Fig1C are well-characterised DDR genes, such as *WRN* and *FANCM* (Ciccia & Elledge, 2010). In addition, a number of these genes have been previously reported to be associated with the DNA damage response, such as *DICER1*, *SMARCA4/BRG1*, *ARID1B*, *ARID2*, *KDM5C*, *ATRX* (Aguilera & Lopez-Contreras, 2023; Davo-Martinez *et al*, 2023; Francia *et al*, 2012; Rondinelli *et al*, 2015; Schiavoni *et al*, 2022).

As advised, we have amended the text accordingly. Page 9 starting on line starting 38 now reads: “ *In addition, our DDR screen confirms previous reports characterizing the role of WRN, FANCM, DICER1, SMARCA4/BRG1, ARID1B, ARID2, KDM5C, and ATRX in the DDR⁶²⁻⁶⁷. Consistently, several chromatin remodellers have been recently reported to have roles in the maintenance of 3D genome organization and enabling DDR^{21, 62, 68}.* ”

References:

Aguilera P, Lopez-Contreras AJ (2023) *ATRX, a guardian of chromatin. Trends Genet* 39: 505-519

Ciccia A, Elledge SJ (2010) *The DNA damage response: making it safe to play with knives. Mol Cell* 40: 179-204

Davo-Martinez C, Helfricht A, Ribeiro-Silva C, Raams A, Tresini M, Uruci S, van Cappellen WA, Taneja N, Demmers JAA, Pines A *et al* (2023) *Different SWI/SNF complexes coordinately promote R-loop- and RAD52-dependent transcription-coupled homologous recombination. Nucleic Acids Res* 51: 9055-9074

Francia S, Michelini F, Saxena A, Tang D, de Hoon M, Anelli V, Mione M, Carninci P, d'Adda di Fagagna F (2012) *Site-specific DICER and DROSHA RNA products control the DNA-damage response. Nature* 488: 231-235

Rondinelli B, Schwerer H, Antonini E, Gaviraghi M, Lupi A, Frenquelli M, Cittaro D, Segalla S, Lemaitre JM, Tonon G (2015) *H3K4me3 demethylation by the histone demethylase KDM5C/JARID1C promotes DNA replication origin firing. Nucleic Acids Res* 43: 2560-2574

Schiavoni F, Zuazua-Villar P, Roumeliotis TI, Benstead-Hume G, Pardo M, Pearl FMG, Choudhary JS, Downs JA (2022) *Aneuploidy tolerance caused by BRG1 loss allows chromosome gains and recovery of fitness. Nat Commun* 13: 1731

6. Page 3, line 10: “The loss of HR proficiency”: while the authors explain their focus on DDR and CIN well, the focus on HR as one type of repair comes somewhat unexpected. Why focus on HR and e.g. not NHEJ or replication in general? The choice to focus on HR should be better introduced.

We are grateful to the reviewer for this helpful suggestion.

We initially focused on HR because:

1. Components of the HR pathway are involved in the protection of replication fork integrity and hence avoid replication stress, a very common feature of cancer leading to structural chromosomal instability (Bohly *et al*, 2022; Burrell *et al.*, 2013; Zeman & Cimprich, 2014).

2. HR defects are more common in cancer than defects of NHEJ.

3. The “error-free” repair of DSB by HR is crucial for genome integrity (Ciccina & Elledge, 2010), particularly for the accurate repair of the replication stress and clastogenic breaks investigated in our original DDR screen as shown in Fig1C. Additionally, unlike replication stress-inducing agents, the use of IR-induced DSB allowed a clear starting point for us to measure the early recruitment kinetics of DSB damage foci before we could re-direct our attention to other types of damage such as replication stress.

We wholeheartedly agree with the points raised by both reviewers regarding the necessity to elucidate a potential role played for FAT1 in NHEJ (please see Reviewer 1, comment 1 above for a more detailed response). Briefly, we found that the knockdown of FAT1 significantly impairs alternate end-joining. We did not observe a significant change in the distal end-joining rate when FAT1 was knocked down. Furthermore, to investigate potential evidence of microhomology arising from MMEJ in the genome, we utilized WGD data from Genomics England WGS sequencing data (since from TRACERx only WES data are available). Despite observing a significant increase in indel burden in *FAT1* mutated samples, we did not observe an enrichment of COSMIC mutational profiles ID6 and SBS3 (Fig R1C1 panels C-E), both of which are associated with indels as a result of end-joining (Alexandrov *et al.*, 2020).

In addition, we further strengthened our HR mutational signature data with a new analysis of TRACERx421 sequencing data. In line with the previous TCGA results, we observed a significant increase in HRD-related genomic scars such as Large Scale Transition (LST) and Telomeric Allelic Imbalance (TAI) rates in *FAT1* mutated tumour samples (see FigR1C1 panels F-H). In conclusion, utilizing experimental and bioinformatic data from TCGA, TRACERx and Genomics England, we propose that *FAT1* loss significantly affects HR repair efficiency.

We are grateful for the suggestions. We have now incorporated NHEJ and Alt-EJ reporter data, along with new bioinformatics analyses of end-joining mutational signatures as FigS12.

We also added a sentence starting on page 4 line 38, which now reads “Using established reporters for total and alternate end-joining activities²², we confirmed that transient *FAT1* siRNA depletion reduced alternative end-joining efficiency but did not significantly reduce distal end-joining rate (FigS12AB). Consistently, utilizing WGS data from Genomics England, no significant difference was observed in ID6 and SBS3 mutational profiles (FigS12CD), both of which are associated with indels due to NHEJ activity.”

References:

Alexandrov LB, Kim J, Haradhvala NJ, Huang MN, Tian Ng AW, Wu Y, Boot A, Covington KR, Gordenin DA, Bergstrom EN *et al* (2020) The repertoire of mutational signatures in human cancer. *Nature* 578: 94-101

Bohly N, Schmidt AK, Zhang X, Slusarenko BO, Hennecke M, Kschischo M, Bastians H (2022) Increased replication origin firing links replication stress to whole chromosomal instability in human cancer. *Cell Rep* 41: 111836

Burrell RA, McClelland SE, Endesfelder D, Groth P, Weller MC, Shaikh N, Domingo E, Kanu N, Dewhurst SM, Gronroos E *et al* (2013) Replication stress links structural and numerical cancer chromosomal instability. *Nature* 494: 492-496

Ciccina A, Elledge SJ (2010) The DNA damage response: making it safe to play with knives. *Mol Cell* 40: 179-204

Zeman MK, Cimprich KA (2014) Causes and consequences of replication stress. *Nat Cell Biol* 16: 2-9

7. Figure S3: the western blots, especially the one for FAT1, are not very convincing. Ideally they should be replaced.

We sincerely apologise for the low-quality image data due to reduced resolution during pdf uploading. Higher-quality images are included in the revised manuscript.

8. All figures showing IF images of various types of DNA foci (for instance Figs S4A, S5A, but many others): from the images it is not clear what is quantified: total signal or foci? I assume foci? Related to this, more zoomed images need to be shown to better appreciate the effects on the number of foci.

We apologise for the omission of this detail. We quantified foci per cell. We have now added insets of a zoomed-in nucleus. We have also added an external download link for high-resolution images in case the pdf resolution is compromised again during the uploading process.

We have now altered the y-axis labels of the relevant figures (Fig1G, 3AB, S4A, S5A and S9ABC) from “X foci” to “X foci per cell (N)”.

9. Page 3, line 26: “mutations which impair HR efficiency” should be “mutations that impair HR efficiency”

We thank the reviewer for the suggestion, we have now altered the text.

10. Sup Fig6C: the CtIP signalling is barely visible from the IF images. Please provide better (and zoomed) images so that the foci can be seen, also in a printed version.

Again, we apologise for the low-quality image.

We added better images with insets showing a zoomed-in nucleus. As a precautionary measure, we have also added an external download link for high-resolution images just in case the PDF resolution is compromised again during the uploading process.

Main figure pdf download link: <https://jmp.sh/gYfTY6dt>

Supplementary pdf figure download link: <https://jmp.sh/ZvU5vtfM>

Download link for Response to reviewer’s comment: <https://jmp.sh/aRKcnBwt>

11. The authors conclude from Sup Fig 7C that Rad51 foci are reduced in FAT1 siRNA cells, but this seems to be only the case for the 4 h time point; for the other time points the number of foci seems to be up (significant for 24 hours, and a trend for 30 mins and 1 h).

So I am not sure how strong this conclusion is.

We apologise for the misunderstanding. Revised FigS8C (corresponding previous Sup figS7C) depicts **53BP1 foci formation rate** in FAT1 siRNA cells. There is no significant change at the early time point of 30 min and 1 hr post-IR irradiation, but there appears to be a significant increase in the late time point at 24 hours after IR irradiation. Also, it is noteworthy that 24 hrs post IR, 53BP1 foci appear to be enlarged and resemble 53BP1 bodies first reported by the Jiri Lukas and Jiri Bartek labs (Lukas *et al.*, 2011). Taken together, this phenotype resembles the hallmark of HR pathway deficiency, where the early chromatin opening is not affected (marked by 53BP1 foci formation rate 1 hr within irradiation). However, these cells exhibit an increase in 53BP1 foci 24 hrs after IR, suggestive of persisting unrepaired damage.

Reference:

Lukas C, Savic V, Bekker-Jensen S, Doil C, Neumann B, Pedersen RS, Grofte M, Chan KL, Hickson ID, Bartek J et al (2011)
53BP1 nuclear bodies form around DNA lesions generated by mitotic transmission of chromosomes under replication stress.
Nat Cell Biol 13: 243-253

12. Page 4 – section “FAT1 ablation leads to CIN”. This section should be merged with the section “FAT1 ablation leads to mitotic errors” as both sections discuss the same type of findings, only using different approaches. For instance, reading this section, I missed time lapse imaging approaches, which in fact were done and support the phenotype, but which are only discussed later in the paper. This is, in my view, confusing.

We thank the reviewer for this suggestion, we have now restructured the manuscript accordingly. We have moved the data related to the TRACERx421 cohort up to new Fig 2, so that the HRD, CIN and mitotic error sections are in proximity (revised Figs 3, 4 and 5, respectively).

We thank the reviewer for their advice and agree that the readability of our manuscript is much improved after restructuring.

13. Page 4, line 19: the effect on number of micronuclei is quite weak. The authors should include a positive control for the induction of micronuclei, for instance by inhibiting Wee1 or Mps1 to compare their phenotype to.

We apologies for any misunderstanding. We did not quantify the “number of micronuclei per cell”, but we quantified the “proportion of cells with micronuclei” as per common practice. One benefit of using U2OS cells for micronuclei studies is that U2OS is so widely used in the field that one can compare the results with relative ease.

REDACTED

As highlighted in our response to reviewer 1 comment 3, in a recent publication, the Lee Zou lab also used U2OS cells to investigate the outcome of inhibiting the key upstream HR mediator ATR plays in micronuclei generation (Joo *et al.*, 2023). Moreover, Joo and coworkers also used an MPS1 inhibitor in U2OS cells so as mentioned by reviewer 3 we can compare the magnitude of our results. As shown above, both the basal level of micronuclei formation and the induction rates are very similar (see above, left: fig1A of Joo et al vs right: revised Fig 4A in the revised manuscript).

Joo and co-workers demonstrated that pharmacological inhibition of MPS1 increases micronuclei formation rate in U2OS cells by about 2 fold (left panel, *MPS1i* control lane), which is comparable to what we observe with *FAT1* depleted cells subjected to replication stress (right panel, 2 fold increase). Similarly, we have compared our results to a previous publications where Feng and Jasin reported that *BRCA2* deletion in MCF10A cells also yielded a 2-2.5 fold increase in the percentage of micronuclei positive U2OS rate (Fig S10C in Feng et. al) (Feng & Jasin, 2017). Our results therefore are typically in line with previous publications.

References:

Joo YK, Black EM, Trier I, Haakma W, Zou L, Kabeche L (2023) ATR promotes clearance of damaged DNA and damaged cells by rupturing micronuclei. *Mol Cell* 83: 3642-3658 e3644

Feng W, Jasin M (2017) *BRCA2* suppresses replication stress-induced mitotic and G1 abnormalities through homologous recombination. *Nat Commun* 8: 525

14. Page 4, line 40, ratio dN/dS: were these non-synonymous mutations single amino acid substitutions or frameshifts, i.e. loss of function or only partial loss of function or even gain of function mutations?

We are sorry we did not explain this concept more clearly. For the dN/dS analysis, the non-synonymous mutations were all single-base substitutions. However, dN/dS is agnostic to whether these mutations are gain/loss-of function mutations. The concept, which was coined by Martincorena et al, compares the mutation at the genomic level (Martincorena et al, 2017) and whether mutation will change the codon in a way that will translate to a different amino acid. It should be noted that these mutations are not annotated so we do not consider the specific functional outcomes of given mutations. Also, only base substitution mutations are analysed; frameshift mutations are not considered.

Essentially, this is a high-level estimate to account for the enrichment of “impactful” mutations (missense) over “neutral” mutations. This gives us an estimate of whether mutations in each gene(s) are favoured over the duration of cancer evolution.

Also, why does the ratio drop in WGD tumours?

As above, dN/dS is an estimate of the enrichment of non-synonymous mutations relative to an expected background. The null hypothesis is that the dN/dS estimate (+/- the 95% confidence interval) overlaps 1, and therefore there is no enrichment or depletion of non-synonymous mutations compared with the background rate.

An estimate significantly above 1 suggests an enrichment of non-synonymous mutations, and therefore potential positive selection. Conversely, an estimate significantly below 1 would suggest evidence for negative selection. In this case, we find evidence for positive selection in *FAT1* before WGD (dN/dS estimate and 95% confidence interval above 1), but after WGD there is no evidence for selection either way. In other words, this suggests that selection for *FAT1* mutations might be beneficial for the tumour before WGD (during early tumour evolution), but there is no evidence for selection (positive or negative) post-WGD.

We have added further explanations in Page 3, lines 40-42 to clarify the interpretation of dN/dS:

“Estimates of dN/dS significantly above 1 suggest positive selection, whereas estimates below 1 suggest negative selection. Estimates overlapping 1 imply that there isn’t evidence of selection.”

We also attempted to better explain dN/dS by adding a cartoon in Figure 3A.

References:

Martincorena I, Raine KM, Gerstung M, Dawson KJ, Haase K, Van Loo P, Davies H, Stratton MR, Campbell PJ (2017) Universal Patterns of Selection in Cancer and Somatic Tissues. Cell 171: 1029-1041 e1021

Do progressed tumours select against mutation of FAT1?

We would like to clarify that events that occur before WGD are likely to happen early in tumour evolution since WGD is a common clonal event in NSCLC development (Frankell *et al*, 2023). However, this does not mean that post-WGD events happen at progression. In fact, all tumours used for the TRACERx analyses are primary tumours extracted at the time of surgery. An event occurring early in tumour evolution can happen years before the tumour is even detected, and a large proportion of events occurring post-WGD will still be clonal (observed in all cancer cells in a given tumour, 28% of all mutations that could be timed in LUAD, 34% in LUSC). Hence mutations categorised as early or late will still be clonal regardless of their timing.

Regarding selection, based on the dN/dS ratios, we can conclude that there is no evidence for selection post-WGD in LUSC (see our response to the above point), but we cannot be sure whether there is genuinely no selection, or whether we don’t have the statistical power to detect weak selection.

References:

Frankell AM, Dietzen M, Al Bakir M, Lim EL, Karasaki T, Ward S, Veeriah S, Colliver E, Huebner A, Bunkum A et al (2023) The evolution of lung cancer and impact of subclonal selection in TRACERx. Nature 616: 525-533

15. Related to the above point: it would be good to also include absolute numbers of FAT1 mutations for both tumour types before and after WGD to put the ratios in perspective.

We thank the reviewer for their suggestion. The number of tumours with FAT1 mutations (of any kind) used in the dN/dS analysis are as follows: LUAD early, 12: LUAD late, 7: LUSC early, 17: LUSC late, 4.

These numbers are now included in the legend of the revised figure 2A, which reads as follows: “A, dN/dS ratio analysis in TRACERx421 LUSC cohort demonstrating FAT1 truncation mutations are selected early in tumor evolution. The number of tumours with a FAT1 mutation used in the dN/dS analysis are as follows: LUAD early, 12: LUAD late, 7: LUSC early, 17: LUSC late, 4.”

16. Page 4, line 43: “a significant enrichment of copy number deletion events was identified”: is this in tumour(s) (sections) with or without WGD, i.e. before or after WGD?

We are sorry for not clarifying the data better. To complement our dN/dS data in Fig 2A where we show selection for *FAT1* pre-whole genome doubling, we see an association between clonal (early) whole genome doubling and *FAT1* SCNA loss. Unfortunately, unlike mutations we are unable to reliably *time* copy number events relative to WGD. However, the suggestion by the reviewer encouraged us to further investigate the relationship between *FAT1* SCNA loss and the occurrence of WGD events.

As suggested by the reviewer, using ParallelGDDetect (Frankell *et al.*, 2023) we segregated TRACERx421 LUAD and LUSC patients into three groups:

(i) tumours that did not undergo genome doubling.

(ii) tumours that underwent clonal whole genome doubling (all samples within multiple regional sampling were whole genome doubled).

(iii) tumours with subclonal whole genome doubling, where heterogeneity of ploidies was observed across multiple sampling, indicating that the WGD event occurs late during cancer evolution.

As shown below in FigR3C16, using GISTIC analysis, we have observed a significant enrichment of SCNA loss at *FAT1* genomic loci 4q35.2 in LUAD and LUSC patients with clonal WGD (middle panels, 4q35.2 loci highlighted with yellow box), but not in patients with no WGD nor with subclonal WGD. Furthermore, we cross-referenced the SCNA loss loci with known common and rare chromosome fragile sites (Durkin & Glover, 2007; Kumar *et al*, 2019). *FAT1* genomic locus does not colocalise with common or rare chromosome-fragile sites.

Since we are unable to reliably *time* copy number events relative to WGD, we are not able to infer these *FAT1* SCNA loss events occurred before or after whole genome doubling. However, with our new analysis, we can now conclude that *FAT1* copy number loss events are associated with clonal WGD; but not associated with the patient group did not undergo WGD, nor the patient group with subclonal WGD only.

FigR3C16, GISTIC score analysis of LUAD and LUSC TRACERx cohort demonstrating SCNA loss at *FAT1* genomic locus (4q35.2, red with yellow highlight) is positively selected in patients with clonal whole genome doubling only, but not with patient groups that did not undergo WGD, nor with patients with subclonal WGD only. SCNA loci overlapping with common or rare chromosome fragile sites (Durkin and Glover 2007, Kumar et al 2019) were annotated (common fragile sites=blue, rare fragile sites=green).

We have incorporated these new results into the revised Fig2B and modified the text accordingly.

Sentences starting page3 line45 now reads "*Within the TRACERx421 lung cohort, a significant enrichment of copy number deletion events was identified around the FAT1 genomic locus 4q35.2, indicative of positive selection for loss of this region in patients with clonal WGD only (Fig2B, FigS7BC).*"

We would like to take this opportunity to thank the reviewer, as we believe their comments have significantly improved our manuscript.

References:

Durkin SG, Glover TW (2007) Chromosome fragile sites. *Annu Rev Genet* 41: 169-192

Frankell AM, Dietzen M, Al Bakir M, Lim EL, Karasaki T, Ward S, Veeriah S, Colliver E, Huebner A, Bunkum A et al (2023) The evolution of lung cancer and impact of subclonal selection in TRACERx. *Nature* 616: 525-533

Kumar R, Nagpal G, Kumar V, Usmani SS, Agrawal P, Raghava GPS (2019) HumCFS: a database of fragile sites in human chromosomes. *BMC Genomics* 19: 985

How does this compare to *FAT1* mutations? I.e is the wildtype copy lost? The decreased dN/dS ratio suggests that the wild-type copies are retained and even selected for after WGD.

There isn't necessarily an overlap between copy number loss and driver mutations in *FAT1*. These are analysed independently. We focus on loss of heterozygosity events (LOH) where at least one allele is completely lost, without necessarily having a driver mutation on *FAT1*. LOH is equivalent to having an inactivating mutation in one of the alleles. Still, of the 41 tumours with *FAT1* driver mutations (21 LUSC, 20 LUAD), 17 (14 LUSC, 3 LUAD) also had a *FAT1* LOH event, of which only in 10 cases (all in LUSC) could the mutation timing be determined relative to WGD. In most of these 10 cases the copy number of the *FAT1* mutation is similar (within 1 copy) of the remaining allele, which would suggest only the wild-type allele was lost (but this makes sense, since if the mutant allele was lost, we would not be able to detect it, and we have no way to quantify how often -if at all- this happens). In the rest of the cases, it is more difficult to determine the timing of the mutation relative to the LOH event.

As for the dN/dS ratio, similar to previous comments, we have no evidence to suggest that there is selection for the wild-type allele post-WGD. All we can say based on the dN/dS analysis is that there is no evidence for selection (either positive or negative) for *FAT1* mutations post-WGD.

17. Page 5, line 12: "mutations are selected prior to WGD in LUSC tumours": while this could well be true, it would be fair to add: "but selected against after WGD".

We are sorry for the misunderstanding. As explained above, the dN/dS analysis does not show evidence for negative selection post-WGD (this would be true if the dN/dS estimate was significantly below 1). What the analysis shows is that post-WGD we find no evidence for selection for *FAT1* mutations post-WGD (dN/dS overlap with 1), so we cannot alter the text as suggested.

We now clarify the above in the revised manuscript, page 3 line 37-42 now reads:" To quantify whether mutations in *FAT1* were under positive selection, we measured the enrichment of *FAT1* mutations before and after WGD using the ratio of the observed number of non-synonymous

substitutions per non-synonymous sites to the number of synonymous substitution per synonymous sites (dN/dS)(Martincorena *et al.*, 2017). Estimates of dN/dS significantly above 1 suggest positive selection, whereas estimates below 1 suggest negative selection. Estimates overlapping 1 imply that there isn't evidence of selection."

We also attempted to better explain dN/dS by adding a cartoon in Figure 3A.

18. Page 5, line 15: "likely confounded by the contribution of other HR-related gene alterations": this might be the case, alternatively the effect is not significant because the mutation ratio (and rate(?)) drops after WGD. If this is an alternative explanation, it should be included in the text/discussion.

We thank the reviewer for their text-change suggestion regarding "the mutation ratio (and rate(?)) drops after WGD". We did consider this possibility; however, we do not necessarily expect the mutation rate to drop after WGD. In fact, the opposite is likely to be true, since we have previously reported that the global mutation accumulation rate might actually increase post-WGD (Lopez *et al*, 2020).

References:

Lopez S, Lim EL, Horswell S, Haase K, Huebner A, Dietzen M, Mourikis TP, Watkins TBK, Rowan A, Dewhurst SM et al (2020) Interplay between whole-genome doubling and the accumulation of deleterious alterations in cancer evolution. Nat Genet 52: 283-293

19. Fig 3E,F but also in general: when using CRISPR, showing data for 2 sgRNAs is recommended. From Fig 3E it seems the data is generated from 1 sgRNA with 4 biological replicates: are these separate KO cell lines or 4 biological replicates from 1 cell line? If the latter, authors need to repeat this experiment for 2 more separate KO cell lines (polyclonal or single clones), ideally including a second sgRNA.

We apologise for not explaining the data clearly. The results in revised Figure 5AB (corresponding to the Fig3E mentioned above) was from 4 biological repeats from a single KO line. This line was derived from a single-cell clone transiently transfected with 2 CRISPR guides, both targeting the *FAT1* locus. We used a transient transfection strategy rather than transduction to minimize potential off-target effects introduced by CRISPR expression. We have now updated the results to include 2 different *FAT1* KO single-cell derived clones (both clones derived from 2 CRISPR guides to maximize targeting efficiency, see revised Figure 5B).

As mentioned previously, we have strived to validate the data using orthogonal methods and cell lines. Accordingly, this experiment is further supported using an MCM7 reloading assay (please see revised Figure S15B). We validated this result in the untransformed RPE1 cell line using *FAT1* RNAi (please refer to revised FigS15C). In summary, we observed that both transient *FAT1* knockdown and *FAT1* CRISPR knockout increase the rate of replication beyond G2 (determined through the MCM7 reloading assays and the EdU incorporation assay, Fig5B and Fig S14BC respectively).

Again, we apologise for our lack of clarity, we have now revised the material and methods section, on page22 line 27 which now reads "*CRISPR KO cell lines were generated via a single cell cloning approach. In order to minimize off-target effects, CRISPR plasmids and guides were transiently transfected using GeneJuice (EMD Milipore) and underwent puromycin selection for 5 days prior to single-cell sorting. Single-cell clones were subsequently cultured, validated and frozen. A549 clones were generated with two individual guides (sgFAT1#1 and sgFAT1#2, see material & methods section), while PC9 clones were generated with both guides.*"

20. Page 5, line 34: “at least in the initial cell cycle”: it is not completely clear to me where this conclusion comes from. Either remove or explain better.

We thank the reviewer for the suggestion, we have now altered the sentence as advised and toned down our narrative. Now the sentence starting on page 6 line 8 now reads “As an orthogonal approach, since PC9 cells carry a TP53 R248Q mutation, FAT1 was transiently depleted in untransformed RPE1-hTERT cells, which are TP53 WT, near diploid, and immortalized by hTERT overexpression. In RPE1-hTERT cells, we also observed that FAT1 depletion increased the rate of MCM7 loading beyond 4N, with or without aphidicolin-induced replication stress (FigS15C).”

21. Fig 3F: why are there up to 30% of cells in a tetraploid S-phase in nocodazole arrested cells: how long have the cells been treated with EdU? Did cells start leaking out of mitosis? It might be worthwhile to optimise the experimental conditions (e.g. 16h of noco arrest), which might lower the error bars for the noco-treated conditions.

We apologise for not explaining our data clearly, particularly for not including cell cycle plots for the nocodazole-treated samples, which we believe have contributed to the misunderstanding.

We believe nocodazole-treated cells did not leak out of mitosis *per se* – these populations did not complete cell division and did not revert to a G1 DNA content. As suggested by the reviewer, we performed the optimization experiment using the same PC9 cell line as in the revised FigS16A (corresponding to Fig 3F mentioned above) and did not see significant exit out of G2 (see Figure R3C21 below, panel A).

FigR3C21A,B, Histograms (A) and representative FACS plots (B) comparing the cell cycle profiles of asynchronous vs nocodazole arrested FAT1 WT PC9 cells (triploid) at various timepoints. A significant mitotic arrest could be observed from 16 hrs (or longer) of nocodazole treatment. Furthermore, a potential endoreplication population could be observed upon 24 hrs or 26 hrs nocodazole treatment. N=3, one-way ANOVA with Sidak corrections. *p<0.05, **p<0.01.

Instead, we believe after nocodazole treatment, cells underwent endoreplication, where DNA synthesis persists post-G2, despite mitotic arrest, resulting in polyploidy. This phenomenon has been observed repeatedly by others (Nair et al, Molecular Biology of Cells 2009, Fig 3C, Gandarillas, Davies, and Blanchard, Oncogene 2000, Fig 6, Zeng et al, Cell 2023, figure S2B)(Gandarillas *et al*, 2000; Nair *et al*, 2009; Zeng *et al.*, 2023). The population of endo-replicating cells can range from 20-40% depending on cell lines. In line with these published observations, we observed this phenomenon in the cell cycle histograms upon staining with (see Figure R3C21 above, panel B).

Taken together, our data suggests that in the absence of cell division (nocodazole arrest), FAT1 loss does not add further nascent endoreplication in our system. This is distinct from the WGD induced by cyclin E amplification as described recently (Zeng et al, Cell 2023, fig S2B).

To better reflect our data, we have now altered the sentence starting on page 6 line8, which now reads “As an orthogonal approach, since PC9 cells carry a TP53 R248Q mutation, FAT1 was transiently depleted in untransformed RPE1-hTERT cells, which are TP53 WT, near diploid, and immortalized by hTERT overexpression. In RPE1-hTERT cells, we also observed that FAT1 depletion increased the rate of MCM7 loading beyond 4N, with or without aphidicolin-induced replication stress (FigS15C).”

References:

Gandarillas A, Davies D, Blanchard JM (2000) Normal and c-Myc-promoted human keratinocyte differentiation both occur via a novel cell cycle involving cellular growth and endoreplication. *Oncogene* 19: 3278-3289

Nair JS, Ho AL, Tse AN, Coward J, Cheema H, Ambrosini G, Keen N, Schwartz GK (2009) Aurora B kinase regulates the postmitotic endoreduplication checkpoint via phosphorylation of the retinoblastoma protein at serine 780. *Mol Biol Cell* 20: 2218-2228

Zeng J, Hills SA, Ozono E, Diffley JFX (2023) Cyclin E-induced replicative stress drives p53-dependent whole-genome duplication. *Cell*

22. Fig 3G: the statistical test between HU treated control vs Fat1 KD cells is missing. It looks like that there are fewer cells bypassing mitosis in AMP-treated cells that are devoid of FAT1. Is this not the opposite of what you would expect? Please add the test and comment on this in the results.

We apologise for our omission. An unpaired T-test for revised Fig5C (corresponding to Fig 3G mentioned above) yielded an insignificant p-value of 0.08.

Regarding the term “mitotic bypass”, John Diffley and co-workers’ recent Cell paper demonstrates that under normal conditions, TP53WT cells with prolonged replication stress, skip mitotic chromosome condensation and round up altogether resulting in genome doubling (Zeng et al, Cell 2023). Based on our current data, FAT1 loss does not alter this event. This result does not contradict our findings since mitotic bypass (ie, when cells skip chromosome condensation and appear to round up) is just one of the several ways a cell uses to undergo WGD. Instead, we report that upon losing FAT1, cells DO indeed enter mitosis, however, it is the failure of cytokinesis that results in WGD. (For detailed discussion on this point please also refer to the response to reviewer 1 comment 6 and reviewer 3 comment 25).

We have now updated the statistical test (ns) for revised Fig5C as advised.

References:

Zeng J, Hills SA, Ozono E, Diffley JFX (2023) Cyclin E-induced replicative stress drives p53-dependent whole-genome duplication. *Cell*

23. Fig 13A: also for these type of images, zoomed in versions are required. How is multinucleation quantified?

We are sorry we did not explain our result clearly. Cell regions were defined using phalloidin staining. The nuclear envelope stain emerin was used to define the number of nuclei within a cell. Again, we apologise that the image resolution was inadvertently down-rasterised during the PDF generation process. We have now provided more detail in the methods section and main text. We have also provided zoomed-in images and a link to higher-resolution images.

We have now amended the figure legend for revised Fig S17A, which now reads “**Fig S17, A, Quantification of fixed microscopy imaging data demonstrating that FAT1 siRNA depletion increases the rate of multinucleation of U2OS cells, with or without replication stress induced with hydroxyurea. Multinucleation is quantified using phalloidin to mark cell boundaries and the nuclear envelope stain emerin is used to highlight the number of nuclei per cell.**”

24. Page 6, line 14: “blebbing”: I thought that ‘blebbing’ referred to abnormal dynamic behaviour of a cell during mitosis. I would call this ‘nuclear shape abnormalities’ or something along those lines. Looking at Sup Fig 14B, the blebbing looks like partial binucleation to me, i.e. failed/incomplete cytokinesis. This fits well with an increased rate of WGD due to tetraploidization in cases where cytokinesis fails altogether.

We thank the reviewer for their insightful suggestion.

We have altered the text accordingly. The paragraph starting on page 6 line 37 now reads, “*FAT1 depletion was associated with increased nuclear shape abnormalities in daughter cells after a normal mitosis (Fig5E, movie 2B). In parallel, we observed an increase in nuclear morphology alterations in WGD FAT1-depleted PC9 cells after replication stress (FigS18A).*”

Also, the sentence starting on page 10 line 21 now reads, “*Using live cell microscopy, we demonstrated the precise sequence of events whereby FAT1 depletion not only increases the occurrence of structural CIN in the form of mitotic bridges but also causes nuclear shape deformation.*”

Also, the relevant figure legends (Fig5E, Fig7F, FigS18BC) now include the term “nuclear shape deformations”.

25. Related to the above and the conclusion on page 6, lines 29-31: the conclusions suggest that FAT1 is involved in two separate processes related to replication/HR and chromosome segregation: Is this phenotype not simply a consequence of the HR-defect-induced replication stress that leads to lagging chromosomes and ultimately when not resolved by decatenation into cytokinesis failure? Also see related points below.

We thank the reviewer for their astute comment. This possibility was raised by both reviewers. Initially, our original text on page 6 lines 29-31 in the last manuscript only mentioned that FAT1 is involved in the mechanisms generating both phenotypes, without intending to segregate whether there was a direct linear relationship or separation of functions. However, the comments by both reviewers (R1 comment 6, R3 comment 25, 31 and 33) inspired us to investigate the relationship between HRD, chromosomal segregation error and cytokinesis failure.

As explained previously in the response to R1 comment 6 and R3 comment 4, we observed that co-depletion of FAT1 and YAP1 does not rescue the HR deficiency nor the mitotic error phenotypes (Panel A-E, Fig R1C6). This is distinct from the results reported in revised Fig7F, where FAT1/YAP1 co-depletion rescues failed cytokinesis. Given the role of FAT1 in modulating YAP1 transcriptional activity and localization, we further investigated a potential cause for cytokinesis failure by overexpressing a constitutively active YAP1^{55A} construct. Indeed, upon YAP1^{55A} overexpression, we observed a significant and consistent increase in the WGD population in both TP53 WT/KO RPE1 cells, and FAT1 WT/KO PC9 cells (FigR1C6, panel GH), suggesting that constitutively active YAP1 is sufficient to promote WGD. This suggests that there are two separate processes downstream of FAT1 loss, one responsible for HR repair deficiency and mitotic errors; and another responsible for cytokinesis failure and WGD.

Although the best-characterised function of YAP1 underpins its role in transcriptional activation, members of the HIPPO pathway have also been reported to be involved in the regulation of replication origin firing rate and replication fork maintenance (Melendez Garcia *et al*, 2022; Pefani *et al*, 2014; Pobbati *et al*, 2023). While it is beyond the scope of this manuscript, in the future it may be of scientific importance to establish the respective molecular mechanisms contributing to HR defects, mitotic errors and WGD.

We have highlighted this finding in a revised section, starting on page 8 line 37:

“Co-depletion of FAT1 and YAP1 reverses WGD but not HR deficiency

Given the involvement of FAT1 in the Hippo signalling pathway (Fig6BC), we attempted to systematically delineate whether both the WGD, numerical and structural CIN associated with FAT1 loss could be reversed by co-depleting YAP1. We first investigated whether the HR deficiency might be reversed by FAT1/YAP1 co-depletion. Surprisingly, despite FAT1/YAP1 co-depletion reversing the cell-cycle arrest associated with YAP1 single knockdown (FigS24AB), the co-depletion of FAT1/YAP1 did not rescue the HR activation defects after DSB formation (Fig7A-C). This result was observed using orthogonal methods including the DR-GFP reporter assay, ssDNA formation after endonuclease-induced DSBs, and RAD51 foci formation after IR (Fig7A-C). Next, since unresolved recombination intermediates due to HR repair deficiency can cause mitotic errors^{57, 58}, we next quantified the rate of mitotic errors. YAP1/FAT1 co-depletion failed to rescue the elevated mitotic errors and chromosomal bridges observed in FAT1 KO A549 cells (Fig7DE, FigS24C).

We next investigated whether YAP1 co-depletion might rescue the CIN phenotypes associated with FAT1 loss. Live-cell imaging experiments demonstrated that co-depletion of FAT1 and YAP1 reversed the cytokinesis failure phenotypes (Fig7F left, FigS24D). However, the nuclear envelope deformation observed following FAT1 depletion was only partially rescued (Fig7F right).

Based on these observations, conceivably FAT1 possesses a dual role. One outcome of FAT1 loss is HRD and leads to unresolved recombination intermediates, replication stress, structural CIN and nuclear deformation (Fig3,4,7A-D) – these events appear to be YAP1 independent. In contrast, the elevated rate of cytokinesis failure and subsequent WGD appear to be dependent on YAP1 activity (Fig7F). To test this hypothesis, we performed an EdU incorporation assay (Fig5AB) to visualize the impact of constitutively active mScarlet-YAP1^{55A} overexpression on WGD. Similar to a previous report⁵, we confirmed that mScarlet-YAP1^{55A} overexpression promotes WGD in RPE1 cells (Fig7G) which we observed to be independent of TP53 status. Concurrently, we observed an emerging WGD population in FAT1 WT and FAT1 KO PC9 cells (Fig7H). Taken together, these results suggest that constitutively active YAP1 is sufficient to promote WGD. We postulate that FAT1 loss might drive genome instability through two different routes, one through WGD, dependent on YAP1, and the second through HR deficiency, driving structural CIN, in a TEAD/YAP1-independent manner.”

References:

Melendez Garcia R, Haccard O, Chesneau A, Narassimprakash H, Roger J, Perron M, Marheineke K, Bronchain O (2022) A non-transcriptional function of Yap regulates the DNA replication program in Xenopus laevis. Elife 11

Pefani DE, Latusek R, Pires I, Grawenda AM, Yee KS, Hamilton G, van der Weyden L, Esashi F, Hammond EM, O'Neill E (2014) RASSF1A-LATS1 signalling stabilizes replication forks by restricting CDK2-mediated phosphorylation of BRCA2. Nat Cell Biol 16: 962-971, 961-968

Pobbati AV, Kumar R, Rubin BP, Hong W (2023) Therapeutic targeting of TEAD transcription factors in cancer. Trends Biochem Sci 48: 450-462

26. The section on heterogeneity in the next section (page 6 lines 37-page 7 line 5) shows similar data

as presented in the previous section and is a better fit there. This will simplify comparing the effects between cell lines at once.

We thank the reviewer for this excellent suggestion.

As suggested by the reviewer, we have now relocated the paragraph in the CIN section to serve as a further orthogonal validation (page 5 lines 25-35).

27. The section on evolution (page 7, lines 7-14) is quite shallow. To strengthen the point that FAT1 depletion drives cancer cell evolution, more experiments are required, possibly using other drugs, particularly drugs that benefit from copy number changes. Alternatively the authors could leave this out or deemphasize these experiments and include them into the CIN section as well.

We have de-emphasized these experiments and relocated them to the last paragraph of the previous section on page 7 (line 12-23).

Again, we thank the reviewer for their suggestion, we believe addressing both comments 26 and 27 has improved the readability of the revised manuscript.

28. Related to the above point: Can the authors rule out that Osimertinib resistance arose from EGFR gene mutations?

Indeed, we agree with the reviewer that *FAT1* KO-induced Osimertinib resistance can arise from *EGFR* mutations. However, in our recent studies, we have observed elevated Osimertinib or erlotinib resistance rates correlating with elevated *EGFR* mutation rates in PC9 cells that possess a *TP53* mutation or overexpress the APOBEC deaminase (Caswell *et al.*, 2024; Sebastijan Hobor, 2024, In press. DOI:10.1038/s41467-024-47606-9). Both studies report that Erlotinib and Osimertinib resistance can arise through means that are distinct from *EGFR* gain-of-function resistance.

Based on these studies, we attempted to address the impact of *FAT1* loss on TKI resistance. First, we confirmed that *FAT1* loss leads to more mitotic errors in genome-doubled PC9 cells (revised FigS14A). Thus, we hypothesised that knocking out *FAT1* in genome-doubled PC9 cells may achieve more EGFR inhibitor-resistant clones due to a heightened mitotic error rate. By transiently expressing CRISPR and 2 *sgFAT1* guides, we made *FAT1* KO genome doubled PC9 cells (revised FigS19). As hypothesised, we observed more acquired Osimertinib-resistant clones when *FAT1* was lost. Consistently, a recent report demonstrated that Osimertinib resistance may arise from the activity of the *YAP/TEAD* pathway (Pfeifer *et al.*, 2024).

However, as suggested by the reviewer in comments 27, we have now deemphasized these experiments.

We have de-emphasized these experiments and relocated them to the last paragraph of the previous section on page 7 (lines 12-23).

References:

Caswell DR, Gui P, Mayekar MK, Law EK, Pich O, Bailey C, Boumelha J, Kerr DL, Blakely CM, Manabe T *et al* (2024) The role of APOBEC3B in lung tumor evolution and targeted cancer therapy resistance. *Nat Genet* 56: 60-73

Sebastijan Hobor MAB, Crispin T. Hiley, Marcin Skrzypsk, Alexander M. Frankell, Bjorn Bakker, Thomas B. K. Watkins, Aleksandra Markovets, Jonathan R Dry, Nnennaya Kanu, Simone Zaccaria, Eva Grönroos and Charles Swanton (2024) Mixed responses to targeted therapy driven by chromosomal instability through p53 dysfunction and genome doubling. *Nature communication*

Pfeifer M, Brammeld JS, Price S, Pilling J, Bhavsar D, Farcas A, Bateson J, Sundarrajan A, Miragaia RJ, Guan N et al (2024) Genome-wide CRISPR screens identify the YAP/TEAD axis as a driver of persister cells in EGFR mutant lung cancer. *Commun Biol* 7: 497

29. Sup Fig 17E: From the figure it appears that the basal ploidy of PC9 cells is 6, but the text says these cells are triploid. Which of the two is correct?

We apologise for not explaining our results clearly.

Indeed, the basal ploidy of PC9 cells is triploid. We hypothesised that *FAT1* KO in genome-doubled PC9 cells may achieve more EGFR inhibitor-resistant clones due to a heightened mitotic error rate. Using the transient expression of CRISPR and 2 *sgFAT1* guides in a spontaneously genome-doubled PC9 cell (clone selected from Hobor et al, *Nat Comms* 2024) (Sebastijan Hobor, 2024, In press. DOI:10.1038/s41467-024-47606-9), we then made *FAT1* KO WGD PC9 cells – hence a basal ploidy of 6.

We have attempted to explain the system better in the revised manuscript, sentences starting on page 5 line 25 now reads, “*Since FAT1 mutations are both common in lung cancer and evolutionarily selected before WGD (Fig2A-C, FigS6C-E), we aimed to orthogonally validate the mitotic defect associated with FAT1 loss using the near-triploid PC9 lung adenocarcinoma cell line, PC9, and one isogenic hexaploid clone*¹⁴.”

30. Page 7, line 26: “elevated nuclear localization”: where does the YAP come from: from the images in Fig 4B there does not appear to be a cytoplasmic pool. In Sup Fig 18A there seems to be some cytoplasmic YAP, but these images look quite overexposed when looking at the nuclear pool. Are there additional ways to show translocation, e.g. by isolating cytoplasmic and nuclear fractions?

We apologise for the confusion. The apparent lack of cytosolic YAP1 in some images reflects the different protocols used to extract soluble proteins before fixation and staining. The impact of which is documented in the figure FigR3C30, panels A vs. B, below.

FigR3C30, A, Representative images showing YAP/TAZ staining following PTEMF treatment. Only the nuclear signal remains. **B**, Representative images showing YAP/TAZ staining following PFA fixation. The total signal per cell was measured using Fiji. The nuclear area was measured using DAPI. The nuclear YAP/TAZ signal was defined by YAP/TAZ signal overlapping with the DAPI area divided by the total signal in the cell. **C**, TEAD transcriptional activity downstream of the YAP/TAZ signalling pathway is elevated in FAT1 KO cells compared to FAT1 WT cells.

We aimed to use orthogonal methods to determine YAP1 localization in revised Fig6B and FigS21A (corresponding to Fig4B and sup Fig 18A mentioned above, respectively). In revised Fig6B, the RPE1-FUCCI cells are treated with PTEMF buffer (20mM PIPES pH6.8, 10mM EGTA, 0.2% TritonX-100, 1mM MgCl₂, 4%PFA), essentially stripping (extracting) the cytoplasmic content of proteins (including any YAP1 not bound to chromatin), to provide a cleaner signal/noise ratio. In these analyses we have made a direct comparison of the nuclear signal. For revised Fig S21A, we performed a direct PFA fixation and quantified total YAP1 signal vs nuclear YAP1 signal (using the DAPI channel colocalization to demarcate the nuclear region border using the 'watershed' function in FIJI/ImageJ).

In both cases, FIJI/image J was used to quantify YAP1 intensity, so that the analyses remained objective despite potential variability in the noise/exposure of the representative images. We have collated the images in Fig R3C30 above for side-by-side comparisons.

We acknowledge that quantification of YAP1 staining may not be completely informative since *YAP1* and *TAZ/WWTR1* are functionally redundant, share a high degree of homology, and are frequently duplicated in cancer cell lines. Therefore, we decided to design a further orthogonal method to investigate whether *FAT1* ablation not only disrupts YAP1 nuclear localization but also directly affects the transcriptional output of the YAP/TAZ binding partner, TEAD.

To this end, we constructed novel neon green fluorescence reporters with 14x TEAD binding sites as minimal promoter regions. This approach allows the delineation of TEAD transcriptional activity downstream of YAP1/TAZ with single-cell resolution. This is an improvement compared to the existing TEAD luciferase reporter (Dupont *et al.*, 2011), since we no longer need to rely on bulk luciferase assay quantification.

We decided to perform the experiment using the genome-doubled, hexaploid PC9 cell line. Due to its high ploidy number, we believe this is a good model to investigate whether the loss of *FAT1* is still able to control downstream TEAD transcriptional output, despite the presence of multiple copies of *YAP1* and *TAZ/WWTR1* in the genome. Since this is an immunofluorescence approach, we could identify the baseline TEAD activity by quantifying cells where successful TEAD reporter transfection has occurred by co-expression of an HA-tagged eIF4A1 construct, a highly expressed helicase in cancer cells (Meijer *et al.*, 2013). In line with our previous results in Fig5B and FigS21A, we observed an increase in TEAD transcriptional activity in *FAT1* KO cells (Fig R3C30 above, panel C)

Taken together, using 3 orthogonal approaches, we are confident that *FAT1* depletion affects YAP1 nuclear localization and its downstream transcriptional activity in conjunction with TEAD.

We have incorporated these new results into revised Fig 6C and Fig S22 and modified the text accordingly. We would like to take this opportunity to thank the reviewer, as we believe their comments have significantly improved our manuscript.

References:

Dupont S, Morsut L, Aragona M, Enzo E, Giulitti S, Cordenonsi M, Zanconato F, Le Digabel J, Forcato M, Bicciato S *et al* (2011) Role of YAP/TAZ in mechanotransduction. *Nature* 474: 179-183

Meijer HA, Kong YW, Lu WT, Wilczynska A, Spriggs RV, Robinson SW, Godfrey JD, Willis AE, Bushell M (2013) Translational repression and eIF4A2 activity are critical for microRNA-mediated gene regulation. *Science* 340: 82-85

31. Fig 4C-E; Sup Fig 18C: To see if signalling indeed goes through *FAT1* (alone), a DKO approach should be used: i.e. by combining *FAT1*KO with *LATS1* or *LATS2* KO to determine whether the effect is indeed epistatic. The final experiment with YAP1/*FAT1* DKO partially addresses the epistatic relation of *FAT1* in the HIPPO pathway, but not completely.

We thank the reviewer for their insightful comment. We avoided using the term “epistatic” in the main text for precisely the same reason.

Indeed, since there are multiple effector kinases in the HIPPO pathway such as *MST1/2*, *LATS1/2* and *AMOT1/2*, it is complicated to fully define epistatic relationships. However, our observation that co-depletion of *FAT1* and YAP1 does not rescue HR deficiency suggests that the epistatic relationship between *FAT1* and YAP1 is context-dependent and that a separation-of-functions may exist along the *FAT1*-HIPPO pathway (please see response to R1 comment 6 for detail). The co-depletion of *FAT1* and YAP1 does not rescue the HR deficiency phenotypes or mitotic errors (revised Fig7A-E). This is distinct from the results shown in revised Fig7FG, where *FAT1*/YAP1 co-depletion rescues the failed cytokinesis.

Thus, in line with the reviewer's astute prediction above, the epistatic relationship between FAT1 and YAP1 is not a complete one. There may be two separate processes downstream of FAT1, one responsible for HR repair deficiency and mitotic errors; and another responsible for cytokinesis failure and WGD.

We further toned-down the sentences starting on page 8 line 23 and replaced the word "phenocopies". The revised manuscript currently reads "Next, we investigated whether the depletion of LATS1/2, key proteins crucial for preventing YAP1 nuclear localization^{55,56}, might resemble the DDR and CIN phenotypes observed following FAT1 depletion."

32. Figure 4F only shows data for LATS1. The authors should also include relevant data for LATS2.

We agree with the reviewer's comment. We did attempt to include relevant mitotic error data for LATS2. However, LATS2 knockdown results in a significant reduction of mitotic cells, to an extent that precludes us from identifying enough mitotic cells to score mitotic errors (see quantification below FigR3C32).

Therefore, unlike LATS1 (revised fig6G), the LATS2 data were omitted from the manuscript.

FigR3C32, Histograms comparing the percentage of untreated mitotic U2OS cells with CTRL or LATS2 siRNA depletion. 10 fields of views per biological replicate was randomly selected at 60x magnification, and the ratio between mitotic and interphase cells was scored. N=3, one-sided paired T-test *p<0.05.

33. Page 9, lines 5-6: "that were associated with chromosomal instability (CIN)": related to several points raised before: from the data in this paper, it seems that the CIN resulting from FAT1 ablation is secondary to the replication/HR defects and in my view it would be better to present that data as such. Related to this it would help to add a summary figure of the proposed mechanism towards the end of the results or early in the discussion on page 9.

We thank the reviewer for their comment, which is in line with our earlier thoughts. The comments by both reviewers (R1 comment 6, R3 comment 25, 31 and 33) inspired us to further investigate whether a linear relationship might exist between HRD, chromosomal segregation error and WGD.

As explained earlier in this rebuttal, we were surprised to discover that co-depletion of FAT1 and YAP1 does not rescue the HR deficiency phenotypes (Panel A-E, fig R1C6). This is distinct from the results shown in Fig 7F, where FAT1/YAP1 co-depletion rescues the failed cytokinesis. Furthermore, ectopic expression of the constitutively active mScarlet-YAP1^{55A} construct is sufficient to induce WGD. These

observations suggest that there may be two separate processes downstream of FAT1 loss, one responsible for HR repair deficiency and another responsible for mitotic error.

Thus, considering the new data we have added a summary figure to clarify these observations (Fig S26, also see below).

FigS26, Proposed model showing potential interaction between FAT1 and the Hippo pathway.

Reference list

Aguilera P, Lopez-Contreras AJ (2023) ATRX, a guardian of chromatin. *Trends Genet* 39: 505-519

Alexandrov LB, Kim J, Haradhvala NJ, Huang MN, Tian Ng AW, Wu Y, Boot A, Covington KR, Gordenin DA, Bergstrom EN *et al* (2020) The repertoire of mutational signatures in human cancer. *Nature* 578: 94-101

Bakhom SF, Cantley LC (2018) The Multifaceted Role of Chromosomal Instability in Cancer and Its Microenvironment. *Cell* 174: 1347-1360

Bakhom SF, Kabeche L, Murnane JP, Zaki BI, Compton DA (2014) DNA-damage response during mitosis induces whole-chromosome missegregation. *Cancer Discov* 4: 1281-1289

Bartkova J, Horejsi Z, Koed K, Kramer A, Tort F, Zieger K, Guldborg P, Sehested M, Nesland JM, Lukas C *et al* (2005) DNA damage response as a candidate anti-cancer barrier in early human tumorigenesis. *Nature* 434: 864-870

Bartkova J, Rezaei N, Lontos M, Karakaidos P, Kletsas D, Issaeva N, Vassiliou LV, Kolettas E, Niforou K, Zoumpourlis VC *et al* (2006) Oncogene-induced senescence is part of the tumorigenesis barrier imposed by DNA damage checkpoints. *Nature* 444: 633-637

Bennardo N, Cheng A, Huang N, Stark JM (2008) Alternative-NHEJ is a mechanistically distinct pathway of mammalian chromosome break repair. *PLoS Genet* 4: e1000110

Bhargava R, Carson CR, Lee G, Stark JM (2017) Contribution of canonical nonhomologous end joining to chromosomal rearrangements is enhanced by ATM kinase deficiency. *Proc Natl Acad Sci U S A* 114: 728-733

Bohly N, Schmidt AK, Zhang X, Slusarenko BO, Hennecke M, Kschischo M, Bastians H (2022) Increased replication origin firing links replication stress to whole chromosomal instability in human cancer. *Cell Rep* 41: 111836

Burrell RA, McClelland SE, Endesfelder D, Groth P, Weller MC, Shaikh N, Domingo E, Kanu N, Dewhurst SM, Gronroos E *et al* (2013) Replication stress links structural and numerical cancer chromosomal instability. *Nature* 494: 492-496

Caswell DR, Gui P, Mayekar MK, Law EK, Pich O, Bailey C, Boumelha J, Kerr DL, Blakely CM, Manabe T *et al* (2024) The role of APOBEC3B in lung tumor evolution and targeted cancer therapy resistance. *Nat Genet* 56: 60-73

Chan YW, Fugger K, West SC (2018) Unresolved recombination intermediates lead to ultra-fine anaphase bridges, chromosome breaks and aberrations. *Nat Cell Biol* 20: 92-103

Ciccio A, Elledge SJ (2010) The DNA damage response: making it safe to play with knives. *Mol Cell* 40: 179-204

Daniels MJ, Wang Y, Lee M, Venkitaraman AR (2004) Abnormal cytokinesis in cells deficient in the breast cancer susceptibility protein BRCA2. *Science* 306: 876-879

Davo-Martinez C, Helfricht A, Ribeiro-Silva C, Raams A, Tresini M, Uruci S, van Cappellen WA, Taneja N, Demmers JAA, Pines A *et al* (2023) Different SWI/SNF complexes coordinately promote R-loop- and RAD52-dependent transcription-coupled homologous recombination. *Nucleic Acids Res* 51: 9055-9074

Dharanipragada P, Zhang X, Liu S, Lomeli SH, Hong A, Wang Y, Yang Z, Lo KZ, Vega-Crespo A, Ribas A *et al* (2023) Blocking Genomic Instability Prevents Acquired Resistance to MAPK Inhibitor Therapy in Melanoma. *Cancer Discov* 13: 880-909

Di Bona M, Bakhoun SF (2024) Micronuclei and Cancer. *Cancer Discov*: OF1-OF13

Dupont S, Morsut L, Aragona M, Enzo E, Giulitti S, Cordenonsi M, Zanconato F, Le Digabel J, Forcato M, Bicciato S *et al* (2011) Role of YAP/TAZ in mechanotransduction. *Nature* 474: 179-183

Durkin SG, Glover TW (2007) Chromosome fragile sites. *Annu Rev Genet* 41: 169-192

Feng W, Jasin M (2017) BRCA2 suppresses replication stress-induced mitotic and G1 abnormalities through homologous recombination. *Nat Commun* 8: 525

Francia S, Michellini F, Saxena A, Tang D, de Hoon M, Anelli V, Mione M, Carninci P, d'Adda di Fagagna F (2012) Site-specific DICER and DROSHA RNA products control the DNA-damage response. *Nature* 488: 231-235

Frankell AM, Dietzen M, Al Bakir M, Lim EL, Karasaki T, Ward S, Veeriah S, Colliver E, Huebner A, Bunkum A *et al* (2023) The evolution of lung cancer and impact of subclonal selection in TRACERx. *Nature* 616: 525-533

Gandarillas A, Davies D, Blanchard JM (2000) Normal and c-Myc-promoted human keratinocyte differentiation both occur via a novel cell cycle involving cellular growth and endoreplication. *Oncogene* 19: 3278-3289

Ganem NJ, Cornils H, Chiu SY, O'Rourke KP, Arnaud J, Yimlamai D, They M, Camargo FD, Pellman D (2014) Cytokinesis failure triggers hippo tumor suppressor pathway activation. *Cell* 158: 833-848

Iacovoni JS, Caron P, Lassadi I, Nicolas E, Massip L, Trouche D, Legube G (2010) High-resolution profiling of gammaH2AX around DNA double strand breaks in the mammalian genome. *EMBO J* 29: 1446-1457

Jamal-Hanjani M, Wilson GA, McGranahan N, Birkbak NJ, Watkins TBK, Veeriah S, Shafi S, Johnson DH, Mitter R, Rosenthal R *et al* (2017) Tracking the Evolution of Non-Small-Cell Lung Cancer. *N Engl J Med* 376: 2109-2121

Joo YK, Black EM, Trier I, Haakma W, Zou L, Kabeche L (2023) ATR promotes clearance of damaged DNA and damaged cells by rupturing micronuclei. *Mol Cell* 83: 3642-3658 e3644

Kumar R, Nagpal G, Kumar V, Usmani SS, Agrawal P, Raghava GPS (2019) HumCFS: a database of fragile sites in human chromosomes. *BMC Genomics* 19: 985

Li R, Shao J, Jin YJ, Kawase H, Ong YT, Troidl K, Quan Q, Wang L, Bonnavion R, Wietelmann A *et al* (2023) Endothelial FAT1 inhibits angiogenesis by controlling YAP/TAZ protein degradation via E3 ligase MIB2. *Nat Commun* 14: 1980

Lopez S, Lim EL, Horswell S, Haase K, Huebner A, Dietzen M, Mourikis TP, Watkins TBK, Rowan A, Dewhurst SM *et al* (2020) Interplay between whole-genome doubling and the accumulation of deleterious alterations in cancer evolution. *Nat Genet* 52: 283-293

Lukas C, Savic V, Bekker-Jensen S, Doil C, Neumann B, Pedersen RS, Grofte M, Chan KL, Hickson ID, Bartek J *et al* (2011) 53BP1 nuclear bodies form around DNA lesions generated by mitotic transmission of chromosomes under replication stress. *Nat Cell Biol* 13: 243-253

Lukow DA, Sausville EL, Suri P, Chunduri NK, Wieland A, Leu J, Smith JC, Girish V, Kumar AA, Kendall J *et al* (2021) Chromosomal instability accelerates the evolution of resistance to anti-cancer therapies. *Dev Cell* 56: 2427-2439 e2424

Martincorena I, Raine KM, Gerstung M, Dawson KJ, Haase K, Van Loo P, Davies H, Stratton MR, Campbell PJ (2017) Universal Patterns of Selection in Cancer and Somatic Tissues. *Cell* 171: 1029-1041 e1021

Martincorena I, Roshan A, Gerstung M, Ellis P, Van Loo P, McLaren S, Wedge DC, Fullam A, Alexandrov LB, Tubio JM *et al* (2015) Tumor evolution. High burden and pervasive positive selection of somatic mutations in normal human skin. *Science* 348: 880-886

Meijer HA, Kong YW, Lu WT, Wilczynska A, Spriggs RV, Robinson SW, Godfrey JD, Willis AE, Bushell M (2013) Translational repression and eIF4A2 activity are critical for microRNA-mediated gene regulation. *Science* 340: 82-85

Melendez Garcia R, Haccard O, Chesneau A, Narassimprakash H, Roger J, Perron M, Marheineke K, Bronchain O (2022) A non-transcriptional function of Yap regulates the DNA replication program in *Xenopus laevis*. *Elife* 11

Nair JS, Ho AL, Tse AN, Coward J, Cheema H, Ambrosini G, Keen N, Schwartz GK (2009) Aurora B kinase regulates the postmitotic endoreduplication checkpoint via phosphorylation of the retinoblastoma protein at serine 780. *Mol Biol Cell* 20: 2218-2228

Pefani DE, Latusek R, Pires I, Grawenda AM, Yee KS, Hamilton G, van der Weyden L, Esashi F, Hammond EM, O'Neill E (2014) RASSF1A-LATS1 signalling stabilizes replication forks by restricting CDK2-mediated phosphorylation of BRCA2. *Nat Cell Biol* 16: 962-971, 961-968

Pfeifer M, Brammell JS, Price S, Pilling J, Bhavsar D, Farcas A, Bateson J, Sundarajan A, Miragaia RJ, Guan N *et al* (2024) Genome-wide CRISPR screens identify the YAP/TEAD axis as a driver of persister cells in EGFR mutant lung cancer. *Commun Biol* 7: 497

Pobbati AV, Kumar R, Rubin BP, Hong W (2023) Therapeutic targeting of TEAD transcription factors in cancer. *Trends Biochem Sci* 48: 450-462

Rondinelli B, Schwerer H, Antonini E, Gaviraghi M, Lupi A, Frenquelli M, Cittaro D, Segalla S, Lemaitre JM, Tonon G (2015) H3K4me3 demethylation by the histone demethylase KDM5C/JARID1C promotes DNA replication origin firing. *Nucleic Acids Res* 43: 2560-2574

Sadeqzadeh E, de Bock CE, Zhang XD, Shipman KL, Scott NM, Song C, Yeadon T, Oliveira CS, Jin B, Hersey P *et al* (2011) Dual processing of FAT1 cadherin protein by human melanoma cells generates distinct protein products. *J Biol Chem* 286: 28181-28191

Schiavoni F, Zuazua-Villar P, Roumeliotis TI, Benstead-Hume G, Pardo M, Pearl FMG, Choudhary JS, Downs JA (2022) Aneuploidy tolerance caused by BRG1 loss allows chromosome gains and recovery of fitness. *Nat Commun* 13: 1731

Schoonen PM, Talens F, Stok C, Gogola E, Heijink AM, Bouwman P, Fojter F, Tarsounas M, Blatter S, Jonkers J *et al* (2017) Progression through mitosis promotes PARP inhibitor-induced cytotoxicity in homologous recombination-deficient cancer cells. *Nat Commun* 8: 15981

Sebastijan Hobor MA-B, Crispin T, Hiley, Marcin Skrzypsk, Alexander M, Frankell, Bjorn Bakker, Thomas B. K. Watkins, Aleksandra Markovets, Jonathan R Dry, Nnennaya Kanu, Simone Zaccaria, Eva Grönroos and Charles Swanton (2024, In press. DOI:10.1038/s41467-024-47606-9) Mixed responses to targeted therapy driven by chromosomal instability through p53 dysfunction and genome doubling. *Nature communication*

Seluanov A, Mao Z, Gorbunova V (2010) Analysis of DNA double-strand break (DSB) repair in mammalian cells. *J Vis Exp*

Spies J, Lukas C, Somyajit K, Rask MB, Lukas J, Neelsen KJ (2019) 53BP1 nuclear bodies enforce replication timing at under-replicated DNA to limit heritable DNA damage. *Nat Cell Biol* 21: 487-497

Steigemann P, Wurzenberger C, Schmitz MH, Held M, Guizetti J, Maar S, Gerlich DW (2009) Aurora B-mediated abscission checkpoint protects against tetraploidization. *Cell* 136: 473-484

Venkatesan S, Angelova M, Puttick C, Zhai H, Caswell DR, Lu WT, Dietzen M, Galanos P, Evangelou K, Bellelli R *et al* (2021) Induction of APOBEC3 Exacerbates DNA Replication Stress and Chromosomal Instability in Early Breast and Lung Cancer Evolution. *Cancer Discov* 11: 2456-2473

Webster ALH, Sanders MA, Patel K, Dietrich R, Noonan RJ, Lach FP, White RR, Goldfarb A, Hadi K, Edwards MM *et al* (2022) Genomic signature of Fanconi anaemia DNA repair pathway deficiency in cancer. *Nature* 612: 495-502

Wojtalewicz N, Sadeqzadeh E, Weiss JV, Tehrani MM, Klein-Scory S, Hahn S, Schmiegel W, Warnken U, Schnolzer M, de Bock CE *et al* (2014) A soluble form of the giant cadherin Fat1 is released from pancreatic cancer cells by ADAM10 mediated ectodomain shedding. *PLoS One* 9: e90461

Zeman MK, Cimprich KA (2014) Causes and consequences of replication stress. *Nat Cell Biol* 16: 2-9

Zeng J, Hills SA, Ozono E, Diffley JFX (2023) Cyclin E-induced replicative stress drives p53-dependent whole-genome duplication. *Cell*

Zhao B, Wei X, Li W, Udan RS, Yang Q, Kim J, Xie J, Ikenoue T, Yu J, Li L *et al* (2007) Inactivation of YAP oncoprotein by the Hippo pathway is involved in cell contact inhibition and tissue growth control. *Genes Dev* 21: 2747-2761